# Towards Resolving the Implicit Bias of Gradient Descent for Matrix Factorization: Greedy Low-Rank Learning

**Zhiyuan Li**,* **Yuping Luo**\*
Princeton University
{zhiyuanli,yupingl}@cs.princeton.edu

**Kaifeng Lyu**\*
Tsinghua University
vfleaking@gmail.com

## Abstract

Matrix factorization is a simple and natural test-bed to investigate the implicit regularization of gradient descent. Gunasekar et al. (2017) conjectured that Gradient Flow with infinitesimal initialization converges to the solution that minimizes the nuclear norm, but a series of recent papers argued that the language of norm minimization is not sufficient to give a full characterization for the implicit regularization. In this work, we provide theoretical and empirical evidence that for depth-2 matrix factorization, gradient flow with infinitesimal initialization is mathematically equivalent to a simple heuristic rank minimization algorithm, Greedy Low-Rank Learning, under some reasonable assumptions. This generalizes the rank minimization view from previous works to a much broader setting and enables us to construct counter-examples to refute the conjecture from Gunasekar et al. (2017). We also extend the results to the case where depth $\geq 3$, and we show that the benefit of being deeper is that the above convergence has a much weaker dependence over initialization magnitude so that this rank minimization is more likely to take effect for initialization with practical scale.

## 1 Introduction

There are usually far more learnable parameters in deep neural nets than the number of training data, but still deep learning works well on real-world tasks. Even with explicit regularization, the model complexity of state-of-the-art neural nets is so large that they can fit randomly labeled data easily (Zhang et al., 2017). Towards explaining the mystery of generalization, we must understand what kind of implicit regularization does Gradient Descent (GD) impose during training. Ideally, we are hoping for a nice mathematical characterization of how GD constrains the set of functions that can be expressed by a trained neural net.

As a direct analysis for deep neural nets could be quite hard, a line of works turned to study the implicit regularization on simpler problems to get inspirations, for example, low-rank matrix factorization, a fundamental problem in machine learning and information process. Given a set of observations about an unknown matrix $\boldsymbol{W}^* \in \mathbb{R}^{d \times d}$ of rank $r^* \ll d$, one needs to find a low-rank solution $\boldsymbol{W}$ that is compatible with the given observations. Examples include matrix sensing, matrix completion, phase retrieval, robust principal component analysis, just to name a few (see Chi et al. 2019 for a survey). When $\boldsymbol{W}^*$ is symmetric and positive semidefinite, one way to solve all these problems is to parameterize $\boldsymbol{W}$ as $\boldsymbol{W} = \boldsymbol{U}\boldsymbol{U}^\top$ for $\boldsymbol{U} \in \mathbb{R}^{d \times r}$ and optimize $\mathcal{L}(\boldsymbol{U}) := \frac{1}{2}f(\boldsymbol{U}\boldsymbol{U}^\top)$, where $f(\cdot)$ is some empirical risk function depending on the observations, and $r$ is the rank constraint. In theory, if the rank constraint is too loose, the solutions do not have to be low-rank and we may fail to recover $\boldsymbol{W}^*$. However, even in the case where the rank is unconstrained (i.e., $r = d$), GD with small initialization can still get good performance in practice. This empirical observation reveals that the implicit regularization of GD exists even in this simple matrix factorization problem, but its mechanism is still on debate. Gunasekar et al. (2017) proved that Gradient Flow (GD with infinitesimal step size, a.k.a., GF) with infinitesimal initialization finds the minimum nuclear norm solution in a special case of matrix sensing, and further conjectured this holds in general.

**Conjecture 1.1** (Gunasekar et al. 2017, informal). With sufficiently small initialization, GF converges to the minimum nuclear norm solution of matrix sensing.

---

*Alphabet ordering.

Subsequently, Arora et al. (2019a) challenged this view by arguing that a simple mathematical norm may not be a sufficient language for characterizing implicit regularization. One example illustrated in Arora et al. (2019a) is regarding matrix sensing with a single observation. They showed that GD with small initialization enhances the growth of large singular values of the solution and attenuates that of smaller ones. This enhancement/attenuation effect encourages low-rank, and it is further intensified with depth in deep matrix factorization (i.e., GD optimizes $f(\boldsymbol{U}_1 \cdots \boldsymbol{U}_L)$ for $L \geq 2$). However, these are not captured by the nuclear norm alone. Gidel et al. (2019); Gissin et al. (2020) further exploited this idea and showed in the special case of full-observation matrix sensing that GF learns solutions with gradually increasing rank. Razin and Cohen (2020) showed in a simple class of matrix completion problems that GF decreases the rank along the trajectory while any norm grows towards infinity. More aggressively, they conjectured that the implicit regularization can be explained by rank minimization rather than norm minimization.

**Our Contributions.**    In this paper, we move one further step towards resolving the implicit regularization in the matrix factorization problem. Our theoretical results show that GD performs rank minimization via a greedy process in a broader setting. Specifically, we provide theoretical evidence that GF with infinitesimal initialization is in general mathematically equivalent to another algorithm called *Greedy Low-Rank Learning* (GLRL). At a high level, GLRL is a greedy algorithm that performs rank-constrained optimization and relaxes the rank constraint by 1 whenever it fails to reach a global minimizer of $f(\cdot)$ with the current rank constraint. As a by-product, we refute Conjecture 1.1 by demonstrating an counterexample (Example 5.9).

We also extend our results to deep matrix factorization Section 6, where we prove that the trajectory of GF with infinitesimal identity initialization converges to a deep version of GLRL, at least in the early stage of the optimization. We also use this result to confirm the intuition achieved on toy models (Gissin et al., 2020), that benefits of depth in matrix factorization is to encourage rank minimization even for initialization with a relatively larger scale, and thus it is more likely to happen in practice. This shows that describing the implicit regularization using GLRL is more expressive than using the language of norm minimization. We validate all our results with experiments in Appendix E.

## 2    RELATED WORKS

**Norm Minimization.**    The view of norm minimization, or the closely related view of margin maximization, has been explored in different settings. Besides the nuclear norm minimization for matrix factorization (Gunasekar et al., 2017) discussed in the introduction, previous works have also studied the norm minimization/margin maximization for linear regression (Wilson et al., 2017; Soudry et al., 2018a;b; Nacson et al., 2019b;c; Ji and Telgarsky, 2019b), deep linear neural nets (Ji and Telgarsky, 2019a; Gunasekar et al., 2018), homogeneous neural nets (Nacson et al., 2019a; Lyu and Li, 2020), ultra-wide neural nets (Jacot et al., 2018; Arora et al., 2019b; Chizat and Bach, 2020).

**Small Initialization and Rank Minimization.**    The initialization scale can greatly influence the implicit regularization. A sufficiently large initialization can make the training dynamics fall into the lazy training regime defined by Chizat et al. (2019) and diminish test accuracy. Using small initialization is particularly important to bias gradient descent to low-rank solutions for matrix factorization, as empirically observed by Gunasekar et al. (2017). Arora et al. (2019a); Gidel et al. (2019); Gissin et al. (2020); Razin and Cohen (2020) studied how gradient flow with small initialization encourages low-rank in simple settings, as discussed in the introduction. Li et al. (2018) proved recovery guarantees for gradient flow solving matrix sensing under Restricted Isometry Property (RIP), but the proof cannot be generalized easily to the case without RIP. Belabbas (2020) made attempts to prove that gradient flow is approximately rank-1 in the very early phase of training, but it does not exclude the possibility that the approximation error explodes later and gradient flow is not converging to low-rank solutions. Compared to these works, the current paper studies how GF encourages low-rank in a much broader setting.

## 3    BACKGROUND
**Notations.**    For two matrices $\boldsymbol{A}, \boldsymbol{B}$, we define $\langle \boldsymbol{A}, \boldsymbol{B} \rangle := \mathrm{Tr}(\boldsymbol{A}\boldsymbol{B}^\top)$ as their inner product. We use $\|\boldsymbol{A}\|_\mathrm{F}, \|\boldsymbol{A}\|_*$ and $\|\boldsymbol{A}\|_2$ to denote the Frobenius norm, nuclear norm and the largest singular value of $\boldsymbol{A}$ respectively. For a matrix $\boldsymbol{A} \in \mathbb{R}^{d \times d}$, we use $\lambda_1(\boldsymbol{A}), \ldots, \lambda_d(\boldsymbol{A})$ to denote the eigenvalues of $\boldsymbol{A}$ in decreasing order (if they are all reals). We define $\mathbb{S}_d$ as the set of symmetric $d \times d$ matrices and $\mathbb{S}_d^+ \subseteq \mathbb{S}_d$ as the set of positive semidefinite (PSD) matrices. We write $\boldsymbol{A} \succeq \boldsymbol{B}$ or $\boldsymbol{B} \preceq \boldsymbol{A}$ iff $\boldsymbol{A} - \boldsymbol{B}$ is PSD. We use $\mathbb{S}_{d,r}^+, \mathbb{S}_{d,\leq r}^+$ to denote the set of $d \times d$ PSD matrices with rank $= r, \leq r$ respectively.

**Matrix Factorization.** Matrix factorization problem asks one to optimize $\mathcal{L}(\boldsymbol{U}, \boldsymbol{V}) := \frac{1}{2} f(\boldsymbol{U}\boldsymbol{V}^\top)$ among $\boldsymbol{U}, \boldsymbol{V} \in \mathbb{R}^{d \times r}$, where $f : \mathbb{R}^{d \times d} \to \mathbb{R}$ is a convex function and in this paper we assume $f$ is $\mathcal{C}^3$-smooth. A notable example is matrix sensing. There is an unknown rank-$r^*$ matrix $\boldsymbol{W}^* \in \mathbb{R}^{d \times d}$ with $r^* \ll d$. Given $m$ measurements $\boldsymbol{X}_1, \ldots, \boldsymbol{X}_m \in \mathbb{R}^{d \times d}$, one can observe $y_i := \langle \boldsymbol{X}_i, \boldsymbol{W}^* \rangle$ through each measurement. The goal of matrix sensing is to reconstruct $\boldsymbol{W}^*$ via minimizing $f(\boldsymbol{W}) := \frac{1}{2} \sum_{i=1}^m (\langle \boldsymbol{W}, \boldsymbol{X}_i \rangle - y_i)^2$. Matrix completion is a notable special case of matrix sensing in which every measurement has the form $\boldsymbol{X}_i = \boldsymbol{e}_{p_i} \boldsymbol{e}_{q_i}^\top$, where $\{\boldsymbol{e}_1, \cdots, \boldsymbol{e}_d\}$ stands for the standard basis (i.e., exactly one entry is observed through each measurement).

Note that matrix factorization in the general case can be reduced to this symmetric case: let $\boldsymbol{U}' = \left[ \begin{smallmatrix} \boldsymbol{U} \\ \boldsymbol{V} \end{smallmatrix} \right] \in \mathbb{R}^{2d \times r}$, $f'(\left[ \begin{smallmatrix} \boldsymbol{A} & \boldsymbol{B} \\ \boldsymbol{C} & \boldsymbol{D} \end{smallmatrix} \right]) = \frac{1}{2} f(\boldsymbol{B}) + \frac{1}{2} f(\boldsymbol{C})$, then $f(\boldsymbol{U}\boldsymbol{V}^\top) = f'(\boldsymbol{U}'\boldsymbol{U}'^\top)$. So in this paper we focus on the symmetric case as in previous works (Gunasekar et al., 2017), i.e., finding a low-rank solution for the convex optimization problem: $\min_{\boldsymbol{W} \succeq \boldsymbol{0}} f(\boldsymbol{W})$. For this, we parameterize $\boldsymbol{W}$ as $\boldsymbol{W} = \boldsymbol{U}\boldsymbol{U}^\top$ for $\boldsymbol{U} \in \mathbb{R}^{d \times r}$ and optimize $\mathcal{L}(\boldsymbol{U}) := \frac{1}{2} f(\boldsymbol{U}\boldsymbol{U}^\top)$. We assume WLOG throughout this paper that $f(\boldsymbol{W}) = f(\boldsymbol{W}^\top)$; otherwise, we can set $f'(\boldsymbol{W}) = \frac{1}{2} \left( f(\boldsymbol{W}) + f(\boldsymbol{W}^\top) \right)$ so that $f'(\boldsymbol{W}) = f'(\boldsymbol{W}^\top)$ while $\mathcal{L}(\boldsymbol{U}) = \frac{1}{2} f'(\boldsymbol{U}\boldsymbol{U}^\top)$ is unaffected. This assumption makes $\nabla f(\boldsymbol{W})$ symmetric for every symmetric $\boldsymbol{W}$.

**Gradient Flow.** In this paper, we analyze Gradient Flow (GF) for symmetric matrix factorization, defined as the solution of the following ODE for $\boldsymbol{U}(t) \in \mathbb{R}^{d \times r}$:
$$\frac{\mathrm{d}\boldsymbol{U}}{\mathrm{d}t} = -\nabla \mathcal{L}(\boldsymbol{U}) = -\nabla f(\boldsymbol{U}\boldsymbol{U}^\top)\boldsymbol{U}. \tag{1}$$
Let $\boldsymbol{W}(t) = \boldsymbol{U}(t)\boldsymbol{U}(t)^\top \in \mathbb{R}^{d \times d}$. Then the following end-to-end dynamics holds for $\boldsymbol{W}(t)$:
$$\frac{\mathrm{d}\boldsymbol{W}}{\mathrm{d}t} = -\boldsymbol{W}\nabla f(\boldsymbol{W}) - \nabla f(\boldsymbol{W})\boldsymbol{W} =: \boldsymbol{g}(\boldsymbol{W}). \tag{2}$$
We use $\phi(\boldsymbol{W}_0, t)$ to denote the matrix $\boldsymbol{W}(t)$ in (2) when $\boldsymbol{W}(0) = \boldsymbol{W}_0 \succeq \boldsymbol{0}$. Throughout this paper, we assume $\phi(\boldsymbol{W}_0, t)$ exists for all $t \in \mathbb{R}$, $\boldsymbol{W}_0 \succeq \boldsymbol{0}$. It is easy to prove that $\boldsymbol{U}$ is a stationary point of $\mathcal{L}(\cdot)$ (i.e., $\nabla \mathcal{L}(\boldsymbol{U}) = \boldsymbol{0}$) iff $\boldsymbol{W} = \boldsymbol{U}\boldsymbol{U}^\top$ is a critical point of (2) (i.e., $\boldsymbol{g}(\boldsymbol{W}) = \boldsymbol{0}$); see Lemma C.1 for a proof. If $\boldsymbol{W}$ is a minimizer of $f(\cdot)$ in $\mathbb{S}_d^+$, then $\boldsymbol{W}$ is a critical point of (2), but the reverse may not be true, e.g., $\boldsymbol{g}(\boldsymbol{0}) = \boldsymbol{0}$, but $\boldsymbol{0}$ is not necessarily a minimizer.

In this paper, we particularly focus on the overparameterized case, where $r = d$, to understand the implicit regularization of GF when there is no rank constraint for the matrix $\boldsymbol{W}$.

## 4 WARMUP EXAMPLES

First, we illustrate how GD performs greedy learning using two warmup examples.

**Linearization Around the Origin.** In general, for a loss function $\mathcal{L}(\boldsymbol{U}) = \frac{1}{2} f(\boldsymbol{U}\boldsymbol{U}^\top)$, we can always apply Taylor expansion $f(\boldsymbol{W}) \approx f(\boldsymbol{0}) + \langle \boldsymbol{W}, \nabla f(\boldsymbol{0}) \rangle$ around the origin to approximate it with a linear function. This motivates us to study the linear case: $f(\boldsymbol{W}) := f_0 - \langle \boldsymbol{W}, \boldsymbol{Q} \rangle$ for some symmetric matrix $\boldsymbol{Q}$. In this case, the matrix $\boldsymbol{U}$ follows the ODE, $\frac{\mathrm{d}\boldsymbol{U}}{\mathrm{d}t} = \boldsymbol{Q}\boldsymbol{U}$, which can be understood as a continuous version of the classical power iteration method for solving the top eigenvector. Let $\boldsymbol{Q} := \sum_{i=1}^d \mu_i \boldsymbol{v}_i \boldsymbol{v}_i^\top$ be the eigendecomposition of $\boldsymbol{Q}$, where $\mu_1 \geq \mu_2 \geq \cdots \geq \mu_d$ and $\boldsymbol{v}_1, \ldots, \boldsymbol{v}_d$ are orthogonal to each other. Then we can write the solution as:
$$\boldsymbol{U}(t) = e^{t\boldsymbol{Q}}\boldsymbol{U}(0) = \left( \sum_{i=1}^d e^{\mu_i t} \boldsymbol{v}_i \boldsymbol{v}_i^\top \right) \boldsymbol{U}(0). \tag{3}$$
When $\mu_1 > \mu_2$, the ratio between $e^{\mu_1 t}$ and $e^{\mu_i t}$ for $i \neq 1$ increases exponentially fast. As $t \to +\infty$, $\boldsymbol{U}(t)$ and $\boldsymbol{W}(t)$ become approximately rank-1 as long as $\boldsymbol{v}_i^\top \boldsymbol{U}(0) \neq \boldsymbol{0}$, i.e.,
$$\lim_{t \to \infty} e^{-\mu_1 t} \boldsymbol{U}(t) = \boldsymbol{v}_1 \boldsymbol{v}_1^\top \boldsymbol{U}(0), \qquad \lim_{t \to \infty} e^{-2\mu_1 t} \boldsymbol{W}(t) = (\boldsymbol{v}_1^\top \boldsymbol{W}(0) \boldsymbol{v}_1) \boldsymbol{v}_1 \boldsymbol{v}_1^\top. \tag{4}$$
The analysis for the simple linear case reveals that GD encourages low-rank through a process similar to power iteration. However, $f(\boldsymbol{W})$ is non-linear in general, and the linear approximation is close to $f(\boldsymbol{W})$ only if $\boldsymbol{W}$ is very small. With sufficiently small initialization, we can imagine that GD still resembles the above power iteration in the early phase of the optimization. But what if $\boldsymbol{W}(t)$ grows to be so large that the linear approximation is far from the actual $f(\boldsymbol{W})$?

**Full-observation Matrix Sensing.** To understand the dynamics of GD when the linearization fails, we now consider a well-studied special case (Gissin et al., 2020): $\mathcal{L}(\boldsymbol{U}) = \frac{1}{2} f(\boldsymbol{U}\boldsymbol{U}^\top)$, $f(\boldsymbol{W}) = \frac{1}{2} \|\boldsymbol{W} - \boldsymbol{W}^*\|_\mathrm{F}^2$ for some unknown PSD matrix $\boldsymbol{W}^*$. GF in this case can be written as:

$$\frac{\mathrm{d}\boldsymbol{U}}{\mathrm{d}t} = (\boldsymbol{W}^* - \boldsymbol{U}\boldsymbol{U}^\top)\boldsymbol{U}, \qquad \frac{\mathrm{d}\boldsymbol{W}}{\mathrm{d}t} = (\boldsymbol{W}^* - \boldsymbol{W})\boldsymbol{W} + \boldsymbol{W}(\boldsymbol{W}^* - \boldsymbol{W}). \tag{5}$$

Let $\boldsymbol{W}^* := \sum_{i=1}^{d} \mu_i \boldsymbol{v}_i \boldsymbol{v}_i^\top$ be the eigendecomposition of $\boldsymbol{W}^*$. Our previous analysis shows that the dynamics is approximately $\frac{\mathrm{d}\boldsymbol{U}}{\mathrm{d}t} = \boldsymbol{W}^*\boldsymbol{U}$ in the early phase and thus encourages low-rank.

To get a sense for the later phases, we simplify the setting by specifying $\boldsymbol{U}(0) = \sqrt{\alpha}\boldsymbol{I}$ for a small number $\alpha$. We can write $\boldsymbol{W}(0)$ and $\boldsymbol{W}^*$ as diagonal matrices $\boldsymbol{W}(0) = \mathrm{diag}(\alpha, \alpha, \cdots, \alpha)$, $\boldsymbol{W}^* = \mathrm{diag}(\mu_1, \mu_2, \cdots, \mu_d)$ with respect to the basis $\boldsymbol{v}_1, \ldots, \boldsymbol{v}_d$. It is easy to see that $\boldsymbol{W}(t)$ is always a diagonal matrix, since the time derivatives of non-diagonal coordinates stay 0 during training. Let $\boldsymbol{W}(t) = \mathrm{diag}(\sigma_1(t), \sigma_2(t), \cdots, \sigma_d(t))$, then $\sigma_i(t)$ satisfies the dynamical equation $\frac{\mathrm{d}}{\mathrm{d}t}\sigma_i(t) = 2\sigma_i(t)(\mu_i - \sigma_i(t))$, and thus $\sigma_i(t) = \frac{\alpha\mu_i}{\alpha + (\mu_i - \alpha)e^{-2\mu_i t}}$. This shows that every $\sigma_i(t)$ increases from $\alpha$ to $\mu_i$ over time. As $\alpha \to 0$, every $\sigma_i(t)$ has a sharp transition from near 0 to near $\mu_i$ at time roughly $(\frac{1}{2\mu_i} + o(1))\log\frac{1}{\alpha}$, which can be seen from the following limit:

$$\lim_{\alpha \to 0} \sigma_i\left(\left(\tfrac{1}{2\mu_i} + c\right)\log(1/\alpha)\right) = \lim_{\alpha \to 0} \frac{\alpha\mu_i}{\alpha + (\mu_i - \alpha)\alpha^{1+2c\mu_i}} = \begin{cases} 0 & c \in (-\tfrac{1}{2\mu_i}, 0), \\ \mu_i & c \in (0, +\infty). \end{cases}$$

This means for every $q \in (\frac{1}{2\mu_i}, \frac{1}{2\mu_{i+1}})$ for $i = 1, \ldots, d-1$ (or $q \in (\frac{1}{2\mu_i}, +\infty)$ for $i = d$), $\lim_{\alpha \to 0} \boldsymbol{W}(q\log(1/\alpha)) = \mathrm{diag}(\mu_1, \mu_2, \ldots, \mu_i, 0, 0, \cdots, 0)$. Therefore, when the initialization is sufficiently small, GF learns each component of $\boldsymbol{W}^*$ one by one, according to the relative order of eigenvalues. At a high level, this shows a greedy nature of GD: GD starts learning with simple models; whenever it underfits, it increases the model complexity (which is rank in our case). This is also called *sequential learning* or *incremental learning* (Gidel et al., 2019; Gissin et al., 2020).

However, it is unclear how and why this sequential learning/incremental learning can occur in general. Through the first warmup example, we may understand why GD learns a rank-1 matrix in the early phase, but does GD always learn solutions with rank $2, 3, 4, \ldots$ sequentially? If true, what is the mechanism behind this? The current paper answers the questions by providing both theoretical and empirical evidence that the greedy learning behavior does occur in general with a similar reason as for the first warmup example.

## 5  GREEDY LOW-RANK LEARNING (GLRL)

In this section, we present a trajectory-based analysis for the implicit bias of GF on matrix factorization. Our main result is that GF with infinitesimal initialization is generically the same as that of a simple greedy algorithm, *Greedy Low-Rank Learning* (GLRL, Algorithm 1). See Appendix A for a comparison with existing greedy algorithms for rank-constrained optimization.

The GLRL algorithm consists of several phases, numbered from 1. In phase $r$, GLRL increases the rank constraint to $r$ and optimizes $\mathcal{L}(\boldsymbol{U}_r) := \frac{1}{2}f(\boldsymbol{U}_r\boldsymbol{U}_r^\top)$ among $\boldsymbol{U}_r \in \mathbb{R}^{d \times r}$ via GD until it reaches a stationary point $\boldsymbol{U}_r(\infty)$, i.e., $\nabla\mathcal{L}(\boldsymbol{U}_r(\infty)) = \boldsymbol{0}$. At convergence, $\boldsymbol{W}_r := \boldsymbol{U}_r(\infty)\boldsymbol{U}_r^\top(\infty)$ is a critical point of (2), and we call it the *r-th critical point* of GLRL. If $\boldsymbol{W}_r$ is further a minimizer of $f(\cdot)$ in $\mathbb{S}_d^+$, or equivalently, $\lambda_1(-\nabla f(\boldsymbol{W}_r)) \leq 0$ (see Lemma C.2), GLRL returns $\boldsymbol{W}_r$; otherwise GLRL enters phase $r + 1$.

To set the initial point of GD in phase $r$, GLRL appends a small column vector $\boldsymbol{\delta}_r \in \mathbb{R}^d$ to the resulting stationary point $\boldsymbol{U}_{r-1}(\infty)$ from the last phase, i.e., $\boldsymbol{U}_r(0) \leftarrow [\boldsymbol{U}_{r-1}(\infty) \ \boldsymbol{\delta}_r] \in \mathbb{R}^{d \times r}$ (in the case of $r = 1$, $\boldsymbol{U}_1(0) \leftarrow [\boldsymbol{\delta}_1] \in \mathbb{R}^{d \times 1}$). In this way, $\boldsymbol{U}_r(0)\boldsymbol{U}_r^\top(0) = \boldsymbol{W}_{r-1} + \boldsymbol{\delta}_r\boldsymbol{\delta}_r^\top$ is perturbed away from the $(r-1)$-th critical point. In GLRL,

---

**Algorithm 1:** Greedy Low-Rank Learning

**parameter :** step size $\eta > 0$; small $\epsilon > 0$
$r \leftarrow 0, \boldsymbol{W}_0 \leftarrow \boldsymbol{0} \in \mathbb{R}^{d \times d}$, and
  $\boldsymbol{U}_0(\infty) \in \mathbb{R}^{d \times 0}$
**while** $\lambda_1(-\nabla f(\boldsymbol{W}_r)) > 0$ **do**
    $r \leftarrow r + 1$
    $\boldsymbol{u}_r \leftarrow$ unit top eigenvector of
      $-\nabla f(\boldsymbol{W}_{r-1})$
    $\boldsymbol{U}_r(0) \leftarrow [\boldsymbol{U}_{r-1}(\infty) \ \sqrt{\epsilon}\boldsymbol{u}_r] \in \mathbb{R}^{d \times r}$
    **for** $t = 0, 1, \ldots$ **do**
      $\lfloor\ \boldsymbol{U}_r(t+1) \leftarrow \boldsymbol{U}_r(t) - \eta\nabla\mathcal{L}(\boldsymbol{U}_r(t))$
    $\boldsymbol{W}_r \leftarrow \boldsymbol{U}_r(\infty)\boldsymbol{U}_r^\top(\infty)$ [a]
**return** $\boldsymbol{W}_r$

---

[a]In practice, we approximate the infinite time limit by running sufficiently many steps.

---

we set $\boldsymbol{\delta}_r = \sqrt{\epsilon}\boldsymbol{u}_r$, where $\boldsymbol{u}_r$ is the top eigenvector of $-\nabla f(\boldsymbol{W}_r)$ with unit norm $\|\boldsymbol{u}_r\|_2 = 1$, and $\epsilon > 0$ is a parameter controlling the magnitude of perturbation (preferably very small). Note that it is guaranteed that $\lambda_1(-\nabla f(\boldsymbol{W}_{r-1})) > 0$; otherwise $\boldsymbol{W}_{r-1}$ is a minimizer of the convex function $f(\cdot)$ in $\mathbb{S}_d^+$ and GLRL exits before phase $r$.

**Trajectory of GLRL.** We define the (limiting) trajectory of GLRL by taking the learning rate $\eta \to 0$. The goal is to show that the trajectory of GLRL is close to that of GF with infinitesimal initialization. Recall that $\phi(\boldsymbol{W}_0, t)$ stands for the solution $\boldsymbol{W}(t)$ in (2) when $\boldsymbol{W}(0) = \boldsymbol{W}_0$.

**Definition 5.1** (Trajectory of GLRL). Let $\overline{\boldsymbol{W}}_{0,\epsilon} := \boldsymbol{0}$ be the 0th critical point of GLRL. For every $r \geq 1$, if the $(r-1)$-th critical point $\overline{\boldsymbol{W}}_{r-1,\epsilon}$ exists and is not a minimizer of $f(\,\cdot\,)$ in $\mathbb{S}_d^+$, we define $\boldsymbol{W}_{r,\epsilon}^{\mathrm{G}}(t) := \phi(\overline{\boldsymbol{W}}_{r-1,\epsilon} + \epsilon \boldsymbol{u}_{r,\epsilon} \boldsymbol{u}_{r,\epsilon}^\top, t)$, where $\boldsymbol{u}_{r,\epsilon}$ is a top eigenvector of $\nabla f(\overline{\boldsymbol{W}}_{r-1,\epsilon})$ with unit norm, $\|\boldsymbol{u}_{r,\epsilon}\|_2 = 1$. We define $\overline{\boldsymbol{W}}_{r,\epsilon} := \lim_{t \to +\infty} \boldsymbol{W}_{r,\epsilon}^{\mathrm{G}}(t)$ to be the $r$-th critical point of GLRL if the limit exists.

Throughout this paper, we always focus on the case where the top eigenvalue of every $\nabla f(\overline{\boldsymbol{W}}_{r-1,\epsilon})$ is unique. In this case, the trajectory of GLRL is unique for every $\epsilon > 0$, since the normalized top eigenvectors can only be $\pm \boldsymbol{u}_{r,\epsilon}$, and both of them lead to the same $\boldsymbol{W}_{r,\epsilon}^{\mathrm{G}}(t)$.

## 5.1 The Limiting Trajectory: A General Theorem for Dynamical System

To prove the equivalence between GF and GLRL, we first introduce our high-level idea by analyzing the behavior of a more general dynamical system around its critical point, say $\boldsymbol{0}$. A specific example is (2) if we set $\boldsymbol{\theta}$ to be the vectorization of $\boldsymbol{W}$.

$$\frac{\mathrm{d}\boldsymbol{\theta}}{\mathrm{d}t} = \boldsymbol{g}(\boldsymbol{\theta}), \quad \text{where} \quad \boldsymbol{g}(\boldsymbol{0}) = \boldsymbol{0}. \tag{6}$$

We use $\phi(\boldsymbol{\theta}_0, t)$ to denote the value of $\boldsymbol{\theta}(t)$ in the case of $\boldsymbol{\theta}(0) = \boldsymbol{\theta}_0$. We assume that $\boldsymbol{g}(\boldsymbol{\theta})$ is $\mathcal{C}^2$-smooth with $\boldsymbol{J}(\boldsymbol{\theta})$ being the Jacobian matrix and $\phi(\boldsymbol{\theta}_0, t)$ exists for all $\boldsymbol{\theta}_0$ and $t$. For ease of presentation, in the main text we assume $\boldsymbol{J}(\boldsymbol{0})$ is diagonalizable over $\mathbb{R}$ and defer the same result for the general case into Appendix G.3. Let $\boldsymbol{J}(\boldsymbol{0}) = \tilde{\boldsymbol{V}} \tilde{\boldsymbol{D}} \tilde{\boldsymbol{V}}^{-1}$ be the eigendecomposition, where $\tilde{\boldsymbol{V}}$ is an invertible matrix and $\tilde{\boldsymbol{D}} = \mathrm{diag}(\tilde{\mu}_1, \ldots, \tilde{\mu}_d)$ is the diagonal matrix consisting of the eigenvalues $\tilde{\mu}_1 \geq \tilde{\mu}_2 \geq \cdots \geq \tilde{\mu}_d$. Let $\tilde{\boldsymbol{V}} = (\tilde{\boldsymbol{v}}_1, \ldots, \tilde{\boldsymbol{v}}_d)$ and $\tilde{\boldsymbol{V}}^{-1} = (\tilde{\boldsymbol{u}}_1, \ldots, \tilde{\boldsymbol{u}}_d)^\top$, then $\tilde{\boldsymbol{u}}_i, \tilde{\boldsymbol{v}}_i$ are left and right eigenvectors associated with $\tilde{\mu}_i$ and $\tilde{\boldsymbol{u}}_i^\top \tilde{\boldsymbol{v}}_j = \delta_{ij}$. We can rewrite the eigendecomposition as $\boldsymbol{J}(\boldsymbol{0}) = \sum_{i=1}^d \tilde{\mu}_i \tilde{\boldsymbol{v}}_i \tilde{\boldsymbol{u}}_i^\top$. We also assume the top eigenvalue $\tilde{\mu}_1$ is positive and unique. Note $\tilde{\mu}_1 > 0$ means the critical point $\boldsymbol{\theta} = \boldsymbol{0}$ is unstable, and in matrix factorization it means $\boldsymbol{0}$ is a strict saddle point of $\mathcal{L}(\,\cdot\,)$. The key observation is that if the initialization is infinitesimal, the trajectory is almost uniquely determined. To be more precise, we need the following definition:

**Definition 5.2.** For any $\boldsymbol{\theta}_0 \in \mathbb{R}^d$ and $\boldsymbol{u} \in \mathbb{R}^d$, we say that $\{\boldsymbol{\theta}_\alpha\}_{\alpha \in (0,1)}$ converges to $\boldsymbol{\theta}_0$ with positive alignment with $\boldsymbol{u}$ if $\lim_{\alpha \to 0} \boldsymbol{\theta}_\alpha = \boldsymbol{\theta}_0$ and $\liminf_{\alpha \to 0} \left\langle \frac{\boldsymbol{\theta}_\alpha - \boldsymbol{\theta}_0}{\|\boldsymbol{\theta}_\alpha - \boldsymbol{\theta}_0\|_2}, \boldsymbol{u} \right\rangle > 0$.

A special case is that the direction of $\boldsymbol{\theta}_\alpha - \boldsymbol{\theta}_0$ converges, i.e., $\bar{\boldsymbol{\theta}} := \lim_{\alpha \to 0} \frac{\boldsymbol{\theta}_\alpha - \boldsymbol{\theta}_0}{\|\boldsymbol{\theta}_\alpha - \boldsymbol{\theta}_0\|_2}$ exists. In this case, $\{\boldsymbol{\theta}_\alpha\}$ has positive alignment with either $\boldsymbol{u}$ or $-\boldsymbol{u}$ except for a zero-measure subset of $\bar{\boldsymbol{\theta}}$. This means any convergent sequence generically falls into either of these two categories.

The following theorem shows that if the initial point $\boldsymbol{\theta}_\alpha$ converges to $\boldsymbol{0}$ with positive alignment with $\tilde{\boldsymbol{u}}_1$ as $\alpha \to 0$, the trajectory starting with $\boldsymbol{\theta}_\alpha$ converges to a unique trajectory $\boldsymbol{z}(t) := \phi(\alpha \tilde{\boldsymbol{v}}_1, t + \frac{1}{\tilde{\mu}_1} \log \frac{1}{\alpha})$. By symmetry, there is another unique trajectory for sequences $\{\boldsymbol{\theta}_\alpha\}$ with positive alignment to $-\tilde{\boldsymbol{u}}_1$, which is $\boldsymbol{z}'(t) := \phi(-\alpha \tilde{\boldsymbol{v}}_1, t + \frac{1}{\tilde{\mu}_1} \log \frac{1}{\alpha})$. This is somewhat surprising: different initial points should lead to very different trajectories, but our analysis shows that generically there are only two limiting trajectories for infinitesimal initialization. We will soon see how this theorem helps in our analysis for matrix factorization in Sections 5.2 and 5.3.

**Theorem 5.3.** Let $\boldsymbol{z}_\alpha(t) := \phi(\alpha \tilde{\boldsymbol{v}}_1, t + \frac{1}{\tilde{\mu}_1} \log \frac{1}{\alpha})$ for every $\alpha > 0$, then $\boldsymbol{z}(t) := \lim_{\alpha \to 0} \boldsymbol{z}_\alpha(t)$ exists and is also a solution of (6), i.e., $\boldsymbol{z}(t) = \phi(\boldsymbol{z}(0), t)$. If $\boldsymbol{\delta}_\alpha$ converges to $\boldsymbol{0}$ with positive alignment with $\tilde{\boldsymbol{u}}_1$ as $\alpha \to 0$, then $\forall t \in \mathbb{R}$, there is a constant $C > 0$ such that

$$\left\| \phi\left(\boldsymbol{\delta}_\alpha, t + \frac{1}{\tilde{\mu}_1} \log \frac{1}{\langle \boldsymbol{\delta}_\alpha, \tilde{\boldsymbol{u}}_1 \rangle}\right) - \boldsymbol{z}(t) \right\|_2 \leq C \cdot \|\boldsymbol{\delta}_\alpha\|_2^{\frac{\tilde{\gamma}}{\tilde{\mu}_1 + \tilde{\gamma}}}, \tag{7}$$

for every sufficiently small $\alpha$, where $\tilde{\gamma} := \tilde{\mu}_1 - \tilde{\mu}_2 > 0$ is the eigenvalue gap.

*Proof sketch.* The main idea is to linearize the dynamics near origin as we have done for the first warmup example. For sufficiently small $\boldsymbol{\theta}$, by Taylor expansion of $\boldsymbol{g}(\boldsymbol{\theta})$, the dynamics is approximately $\frac{\mathrm{d}\boldsymbol{\theta}}{\mathrm{d}t} \approx \boldsymbol{J}(\boldsymbol{0})\boldsymbol{\theta}$, which can be understood as a continuous version of power iteration. If the linear approximation is exact, then $\boldsymbol{\theta}(t) = e^{t\boldsymbol{J}(\boldsymbol{0})}\boldsymbol{\theta}(0)$. For large enough $t_0$,

$e^{t_0 \boldsymbol{J}(0)} = \sum_{i=1}^{d} e^{\tilde{\mu}_i t_0} \tilde{\boldsymbol{v}}_i \tilde{\boldsymbol{u}}_i^\top = e^{\tilde{\mu}_1 t_0} \tilde{\boldsymbol{v}}_1 \tilde{\boldsymbol{u}}_1^\top + O(e^{\tilde{\mu}_2 t_0})$. Therefore, as long as the initial point $\boldsymbol{\theta}(0)$ has a positive inner product with $\tilde{\boldsymbol{u}}_1$, $\boldsymbol{\theta}(t_0)$ should be very close to $\epsilon \tilde{\boldsymbol{v}}_1$ for some $\epsilon > 0$, and the rest of the trajectory after $t_0$ should be close to the trajectory starting from $\epsilon \tilde{\boldsymbol{v}}_1$. However, here is a tradeoff: we should choose $t_0$ to be large enough so that the power iteration takes effect; but if $t_0$ is so large that the norm of $\boldsymbol{\theta}(t_0)$ reaches a constant scale, then the linearization fails unavoidably. Nevertheless, if the initialization scale is sufficiently small, we show via a careful error analysis that there is always a suitable choice of $t_0$ such that $\boldsymbol{\theta}(t_0)$ is well approximated by $\epsilon \tilde{\boldsymbol{v}}_1$ and the difference between $\boldsymbol{\theta}(t_0 + t)$ and $\phi(\epsilon \tilde{\boldsymbol{v}}_1, t)$ is bounded as well. We defer the details to Appendix G. □

## 5.2 EQUIVALENCE BETWEEN GD AND GLRL: RANK-ONE CASE

Now we establish the equivalence between GF and GLRL in the first phase. The main idea is to apply Theorem 5.3 on (2). For this, we need the following lemma on the eigenvalues and eigenvectors.

**Lemma 5.4.** *Let $\boldsymbol{g}(\boldsymbol{W}) := -\boldsymbol{W}\nabla f(\boldsymbol{W}) - \nabla f(\boldsymbol{W})\boldsymbol{W}$ and $\boldsymbol{J}(\boldsymbol{W})$ be its Jacobian. Then $\boldsymbol{J}(0)$ is symmetric and thus diagonalizable. Let $-\nabla f(0) = \sum_{i=1}^{d} \mu_i \boldsymbol{u}_{1[i]} \boldsymbol{u}_{1[i]}^\top$ be the eigendecomposition of the symmetric matrix $-\nabla f(0)$, where $\mu_1 \geq \mu_2 \geq \cdots \geq \mu_d$. Then $\boldsymbol{J}(0)$ has the form:*

$$\boldsymbol{J}(0)[\boldsymbol{\Delta}] = \sum_{i=1}^{d} \sum_{j=1}^{d} (\mu_i + \mu_j) \left\langle \boldsymbol{\Delta}, \boldsymbol{u}_{1[i]} \boldsymbol{u}_{1[j]}^\top \right\rangle \boldsymbol{u}_{1[i]} \boldsymbol{u}_{1[j]}^\top, \tag{8}$$

*where $\boldsymbol{J}(0)[\boldsymbol{\Delta}]$ stands for the resulting matrix produced by left-multiplying $\boldsymbol{J}(0)$ to the vectorization of $\boldsymbol{\Delta}$. For every pair of $1 \leq i \leq j \leq d$, $\mu_i + \mu_j$ is an eigenvalue of $\boldsymbol{J}(0)$ and $\boldsymbol{u}_{1[i]} \boldsymbol{u}_{1[j]}^\top + \boldsymbol{u}_{1[j]} \boldsymbol{u}_{1[i]}^\top$ is a corresponding eigenvector. All the other eigenvalues are $0$.*

We simplify the notation by letting $\boldsymbol{u}_1 := \boldsymbol{u}_{1[1]}$. A direct corollary of Lemma 5.4 is that $\boldsymbol{u}_1 \boldsymbol{u}_1^\top$ is the top eigenvector of $\boldsymbol{J}(0)$. According to Theorem 5.3, now there are only two types of trajectories, which correspond to infinitesimal initialization $\boldsymbol{W}_\alpha \to 0$ with positive alignment with $\boldsymbol{u}_1 \boldsymbol{u}_1^\top$ or $-\boldsymbol{u}_1 \boldsymbol{u}_1^\top$. As the initialization must be PSD, $\boldsymbol{W}_\alpha \to 0$ cannot have positive alignment with $-\boldsymbol{u}_1 \boldsymbol{u}_1^\top$. For the former case, Theorem 5.6 below states that, for every fixed time $t$, the GF solution $\phi(\boldsymbol{W}_\alpha, T(\boldsymbol{W}_\alpha) + t)$ after shifting by a time offset $T(\boldsymbol{W}_\alpha) := \frac{1}{2\mu_1} \log(\langle \boldsymbol{W}_\alpha, \boldsymbol{u}_1 \boldsymbol{u}_1^\top \rangle^{-1})$ converges to the GLRL solution $\boldsymbol{W}_1^{\mathrm{G}}(t)$ as $\boldsymbol{W}_\alpha \to 0$. The only assumption for this result is that $0$ is not a minimizer of $f(\cdot)$ in $\mathbb{S}_d^+$ (which is equivalent to $\lambda_1(-\nabla f(0)) > 0$) and $-\nabla f(0)$ has an eigenvalue gap. In the full observation case, this assumption is satisfied easily if the ground-truth matrix has a unique top eigenvalue. The proof for Theorem 5.6 is deferred to Appendix I.1.

**Assumption 5.5.** $\mu_1 > \max\{\mu_2, 0\}$, *where $\mu_1 := \lambda_1(-\nabla f(0))$, $\mu_2 := \lambda_2(-\nabla f(0))$.*

**Theorem 5.6.** *Under Assumption 5.5, the following limit $\boldsymbol{W}_1^{\mathrm{G}}(t)$ exists and is a solution of (2).*

$$\boldsymbol{W}_1^{\mathrm{G}}(t) := \lim_{\epsilon \to 0} \boldsymbol{W}_{1,\epsilon}^{\mathrm{G}} \left( \frac{1}{2\mu_1} \log \frac{1}{\epsilon} + t \right) = \lim_{\epsilon \to 0} \phi \left( \epsilon \boldsymbol{u}_1 \boldsymbol{u}_1^\top, \frac{1}{2\mu_1} \log \frac{1}{\epsilon} + t \right). \tag{9}$$

*Let $\{\boldsymbol{W}_\alpha\} \subseteq \mathbb{S}_d^+$ be PSD matrices converging to $0$ with positive alignment with $\boldsymbol{u}_1 \boldsymbol{u}_1^\top$ as $\alpha \to 0$, that is, $\lim_{\alpha \to 0} \boldsymbol{W}_\alpha = 0$ and $\exists \alpha_0, q > 0$ such that $\langle \boldsymbol{W}_\alpha, \boldsymbol{u}_1 \boldsymbol{u}_1^\top \rangle \geq q \|\boldsymbol{W}_\alpha\|_{\mathrm{F}}$ for all $\alpha < \alpha_0$. Then $\forall t \in \mathbb{R}$, there is a constant $C > 0$ such that*

$$\left\| \phi \left( \boldsymbol{W}_\alpha, \frac{1}{2\mu_1} \log \frac{1}{\langle \boldsymbol{W}_\alpha, \boldsymbol{u}_1 \boldsymbol{u}_1^\top \rangle} + t \right) - \boldsymbol{W}_1^{\mathrm{G}}(t) \right\|_{\mathrm{F}} \leq C \|\boldsymbol{W}_\alpha\|_{\mathrm{F}}^{\frac{\tilde{\gamma}}{2\mu_1 + \tilde{\gamma}}} \tag{10}$$

*for every sufficiently small $\alpha$, where $\tilde{\gamma} := 2\mu_1 - (\mu_1 + \mu_2) = \mu_1 - \mu_2$.*

It is worth to note that $\boldsymbol{W}_1^{\mathrm{G}}(t)$ has rank $\leq 1$ for any $t \in \mathbb{R}$, since every $\boldsymbol{W}_{1,\epsilon}^{\mathrm{G}}(t)$ has rank $\leq 1$ and the set $\mathbb{S}_{d,\leq 1}^+$ is closed. This matches with the first warmup example: GD does start learning with rank-1 solutions. Interestingly, in the case where the limit $\overline{\boldsymbol{W}}_1 := \lim_{t \to +\infty} \boldsymbol{W}_1^{\mathrm{G}}(t)$ happens to be a minimizer of $f(\cdot)$ in $\mathbb{S}_d^+$, GLRL should exit with the rank-1 solution $\overline{\boldsymbol{W}}_1$ after the first phase, and the following theorem shows that this is also the solution found by GF.

**Assumption 5.7.** $f(\boldsymbol{W})$ *is locally analytic at each point.*

**Theorem 5.8.** *Under Assumptions 5.5 and 5.7, if $\|\boldsymbol{W}_1^{\mathrm{G}}(t)\|_{\mathrm{F}}$ is bounded for all $t \geq 0$, then the limit $\overline{\boldsymbol{W}}_1 := \lim_{t \to +\infty} \boldsymbol{W}_1^{\mathrm{G}}(t)$ exists. Further, if $\overline{\boldsymbol{W}}_1$ is a minimizer of $f(\cdot)$ in $\mathbb{S}_d^+$, then for PSD matrices $\{\boldsymbol{W}_\alpha\} \subseteq \mathbb{S}_d^+$ converging to $0$ with positive alignment with $\boldsymbol{u}_1 \boldsymbol{u}_1^\top$ as $\alpha \to 0$, it holds that $\lim_{\alpha \to 0} \lim_{t \to +\infty} \phi(\boldsymbol{W}_\alpha, t) = \overline{\boldsymbol{W}}_1$.*

Assumption 5.7 is a natural assumption, since $f(\cdot)$ in most cases of matrix factorization is a quadratic or polynomial function (e.g., matrix sensing, matrix completion). In general, it is unlikely for a gradient-based optimization process to get stuck at saddle points (Lee et al., 2017; Panageas et al., 2019). Thus, we should expect to see in general that GLRL finds the rank-1 solution if the problem is feasible with rank-1 matrices. This means at least for this subclass of problems, the implicit regularization of GD is rather unrelated to norm minimization. Below is a concrete example:

**Example 5.9** (Counter-example of Conjecture 1.1, Gunasekar et al. 2017)**.** Theorem 5.8 enables us to construct counterexamples of the *implicit nuclear norm regularization* conjecture in (Gunasekar et al., 2017). The idea is to construct a loss $\mathcal{L} : \mathbb{R}^{d \times d} \to \mathbb{R}$ where every rank-1 stationary point of $\mathcal{L}(U)$ attains the global minimum but none of them is minimizing the nuclear norm. Below we give a concrete matrix completion problem that meets the above requirement. Let $M$ be a partially observed matrix to be recovered, where the entries in $\Omega = \{(1,3),(1,4),(2,3),(3,1),(3,2),(4,1)\}$ are observed and the others (marked with "?") are unobserved. The optimization problem is defined formally by $\mathcal{L}(U) = \frac{1}{2}f(UU^\top), f(W) = \frac{1}{2}\sum_{(i,j)\in\Omega}(W_{ij} - M_{ij})^2$.

$$M = \begin{bmatrix} ? & ? & 1 & R \\ ? & ? & R & ? \\ 1 & R & ? & ? \\ R & ? & ? & ? \end{bmatrix}, M_{\text{norm}} = \begin{bmatrix} R & 1 & 1 & R \\ 1 & R & R & 1 \\ 1 & R & R & 1 \\ R & 1 & 1 & R \end{bmatrix}, M_{\text{rank}} = \begin{bmatrix} 1 & R & 1 & R \\ R & R^2 & R & R^2 \\ 1 & R & 1 & R \\ R & R^2 & R & R^2 \end{bmatrix}.$$

Here $R > 1$ is a large constant, e.g., $R = 100$. The minimum nuclear norm solution is the rank-2 matrix $M_{\text{norm}}$, which has $\|M_{\text{norm}}\|_* = 4R$ (which is 400 when $R = 100$). $M_{\text{rank}}$ is a rank-1 solution with much larger nuclear norm, $\|M_{\text{norm}}\|_* = 2R^2 + 2$ (which is 20002 when $R = 100$). We can verify that $f(\cdot)$ satisfies Assumptions 5.5 and 5.7 and $W_1^{\text{G}}(t)$ converges to the rank-1 solution $M_{\text{rank}}$. Therefore, GF with infinitesimal initialization converges to $M_{\text{rank}}$ rather than $M_{\text{norm}}$, which refutes the conjecture in (Gunasekar et al., 2017). See Appendix D for a formal statement.

## 5.3 EQUIVALENCE BETWEEN GD AND GLRL: GENERAL CASE

Theorem 5.6 shows that for any *fixed* time $t$, the trajectory of GLRL in the first phase approximates GF with infinitesimal initialization, i.e., $W_1^{\text{G}}(t) = \lim_{\alpha\to 0} \widehat{W}_\alpha(t)$, where $\widehat{W}_\alpha(t) := \phi(W_\alpha, \frac{1}{2\mu_1}\log(\langle W_\alpha, u_1 u_1^\top \rangle^{-1}) + t)$. However, $W_1^{\text{G}}(\infty) \neq \lim_{\alpha\to 0} \widehat{W}_\alpha(\infty)$ does not hold in general, unless the prerequisite in Theorem 5.8 is satisfied, i.e., unless $\overline{W}_1 = W_1^{\text{G}}(\infty)$ is a minimizer of $f(\cdot)$ in $\mathbb{S}_d^+$. This is because of the well-known result that GD converges to local minimizers (Lee et al., 2016; 2017). We adapt Theorem 2 of Lee et al. (2017) to the setting of GF (Theorem I.5) and obtain the following result (Theorem 5.10); see Appendix I.4 for the proof.

**Theorem 5.10.** *Let $f : \mathbb{R}^{d \times d} \to \mathbb{R}$ be a convex $\mathcal{C}^2$-smooth function. (1). All stationary points of $\mathcal{L} : \mathbb{R}^{d \times d} \to \mathbb{R}, \mathcal{L}(U) = \frac{1}{2}f(UU^\top)$ are either strict saddles or global minimizers; (2). For any random initialization, GF (1) converges to strict saddles of $\mathcal{L}(U)$ with probability $0$.*

Therefore, for convex $f(\cdot)$ such as matrix sensing and completion, suppose $f(\cdot)$ has no rank-1 PSD minimizer, then no matter how small $\alpha$ is, $\widehat{W}_\alpha(\infty)$ (if exists) is a minimizer of $f(\cdot)$ with a higher rank and thus away from the rank-1 matrix $\overline{W}_1$. In other words, $W_1^{\text{G}}(t)$ only describes the limiting trajectory of GF in the first phase, i.e., when GF goes from near $0$ to near $\overline{W}_1$. After a sufficiently long time (depending on $\alpha$), GF escapes the critical point $\overline{W}_1$, but this is not described by $W_1^{\text{G}}(t)$.

To understand how GF escapes $\overline{W}_1$, a priori, we need to know how GF approaches $\overline{W}_1$. Using a similar argument for Theorem 5.3, Theorem 5.11 shows that generically GF only escapes in the direction of $v_1 v_1^\top$, where $v_1$ is the (unique) top eigenvector of $-\nabla f(\overline{W}_1)$, and thus the limiting trajectory exactly matches with that of GLRL in the second phase until GF gets close to another critical point $\overline{W}_2 \in \mathbb{S}_{d,\leq 2}^+$. If $\overline{W}_2$ is still not a minimizer of $f(\cdot)$ in $\mathbb{S}_d^+$ (but it is a local minimizer in $\mathbb{S}_{d,\leq 2}^+$ generically), then GF escapes $\overline{W}_2$ and the above process repeats until $\overline{W}_K$ is a minimizer in $\mathbb{S}_d^+$ for some $K$. Here by "generically" we hide some technical assumptions and we elaborate on them in Appendix J. See Figure 1 and Figure 2 for experimental verification of the equivalence between GD and GLRL. We end this section with the following characterization of GF:

**Theorem 5.11** (Theorem I.2, informal)**.** *Let $\overline{W}$ be a critical point of (2) satisfying that $\overline{W}$ is a local minimizer of $f(\cdot)$ in $\mathbb{S}_{d,\leq r}^+$ for some $r \geq 1$ but not a minimizer in $\mathbb{S}_d^+$. Let $-\nabla f(\overline{W}) =$*

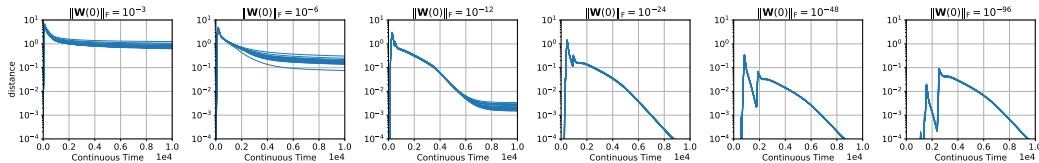

**Figure 1:** The trajectory of depth-2 GD, $\boldsymbol{W}_{\mathrm{GD}}(t)$, converges to the trajectory of GLRL, $\boldsymbol{W}_{\mathrm{GLRL}}(t)$, as the initialization scale goes to 0. We plot $\mathrm{dist}(t) = \min_{t' \in \mathcal{T}} \|\boldsymbol{W}_{\mathrm{GD}}(t) - \boldsymbol{W}_{\mathrm{GLRL}}(t')\|_{\mathrm{F}}$ for different initialization scale $\|\boldsymbol{W}(0)\|_{\mathrm{F}}$, where $\mathcal{T}$ is a discrete subset of $\mathbb{R}$ that $\delta$-covers the entire trajectory of GLRL: $\max_t \min_{t' \in \mathcal{T}} \|\boldsymbol{W}_{\mathrm{GLRL}}(t) - \boldsymbol{W}_{\mathrm{GLRL}}(t')\|_{\mathrm{F}} \le \delta$ for $\delta \approx 0.00042$. For each $\|\boldsymbol{W}(0)\|_{\mathrm{F}}$, we run 20 random seeds and plot them separately. The ground truth $\boldsymbol{W}^* \in \mathbb{R}^{20 \times 20}$ is a randomly generated rank-3 matrix with $\|\boldsymbol{W}^*\|_{\mathrm{F}} = 20$. 30% entries are observed. See more in Appendix E.1.

$\sum_{i=1}^d \mu_i \boldsymbol{v}_i \boldsymbol{v}_i^\top$ be the eigendecomposition of $-\nabla f(\overline{\boldsymbol{W}})$. If $\mu_1 > \mu_2$ and if there exists time $T_\alpha \in \mathbb{R}$ for every $\alpha$ so that $\phi(\boldsymbol{W}_\alpha, T_\alpha)$ converges to $\overline{\boldsymbol{W}}$ with positive alignment with the top principal component $\boldsymbol{v}_1 \boldsymbol{v}_1^\top$ as $\alpha \to 0$, then for every fixed $t$, $\lim_{\alpha \to 0} \phi(\boldsymbol{W}_\alpha, T_\alpha + \frac{1}{2\mu_1} \log \frac{1}{\langle \phi(\boldsymbol{W}_\alpha, T_\alpha), \boldsymbol{v}_1 \boldsymbol{v}_1^\top \rangle} + t)$ exists and is equal to $\boldsymbol{W}^{\mathrm{G}}(t) := \lim_{\epsilon \to 0} \phi(\overline{\boldsymbol{W}} + \epsilon \boldsymbol{v}_1 \boldsymbol{v}_1^\top, \frac{1}{2\mu_1} \log \frac{1}{\epsilon} + t)$.

**Characterization of the trajectory of GF.** Generically, the trajectory of GF with small initialization can be split into $K$ phases by $K+1$ critical points of (2), $\{\overline{\boldsymbol{W}}_r\}_{r=0}^K$ ($\overline{\boldsymbol{W}}_0 = \boldsymbol{0}$), where in phase $r$ GF escapes from $\overline{\boldsymbol{W}}_{r-1}$ in the direction of the top principal component of $-\nabla f(\overline{\boldsymbol{W}}_{r-1})$ and gets close to $\overline{\boldsymbol{W}}_r$. Each $\overline{\boldsymbol{W}}_r$ is a local minimizer of $f(\cdot)$ in $\mathbb{S}_{d, \le r}^+$, but none of them is a minimizer of $f(\cdot)$ in $\mathbb{S}_d^+$ except $\overline{\boldsymbol{W}}_K$. The smaller the initialization is, the longer GF stays around each $\overline{\boldsymbol{W}}_r$. Moreover, $\{\overline{\boldsymbol{W}}_r\}_{r=0}^K$ corresponds to $\{\overline{\boldsymbol{W}}_{r,\epsilon}\}_{r=0}^K$ in Definition 5.1 with infinitesimal $\epsilon > 0$.

# 6 BENEFITS OF DEPTH: A VIEW FROM GLRL

In this section, we consider matrix factorization problems with depth $L \ge 3$. Our goal is to understand the effect of the depth-$L$ parametrization $\boldsymbol{W} = \boldsymbol{U}_1 \boldsymbol{U}_2 \cdots \boldsymbol{U}_L$ on the implicit bias — how does depth encourage GF to find low rank solutions? We take the standard assumption in existing analysis for the end-to-end dynamics that the weight matrices have a balanced initialization, i.e. $\boldsymbol{U}_i^\top(0) \boldsymbol{U}_i(0) = \boldsymbol{U}_{i+1}(0) \boldsymbol{U}_{i+1}^\top(0)$, $\forall 1 \le i \le L-1$. Arora et al. (2018) showed that if $\{\boldsymbol{U}_i\}_{i=1}^L$ is balanced at initialization, then we have the following end-to-end dynamics. Similar to the depth-2 case, we use $\phi(\boldsymbol{W}(0), t)$ to denote $\boldsymbol{W}(t)$, where

$$\frac{\mathrm{d}\boldsymbol{W}}{\mathrm{d}t} = -\sum_{i=0}^{L-1} (\boldsymbol{W}\boldsymbol{W}^\top)^{\frac{i}{L}} \nabla f(\boldsymbol{W}) (\boldsymbol{W}^\top \boldsymbol{W})^{1 - \frac{i+1}{L}}. \tag{11}$$

The lemma below is the foundation of our analysis for the deep case, which greatly simplifies (11). Due to the space limit, we defer its derivations and applications into Appendix K.

**Lemma 6.1.** For $\boldsymbol{M}(t) := \boldsymbol{W}(t)^{2/L}$, we have $\frac{\mathrm{d}\boldsymbol{M}}{\mathrm{d}t} = -\nabla f(\boldsymbol{M}^{L/2}) \boldsymbol{M}^{L/2} - \boldsymbol{M}^{L/2} \nabla f(\boldsymbol{M}^{L/2})$.

Our main result, Theorem 6.2, gives a characterization of the limiting trajectory for deep matrix factorization with infinitesimal identity initialization. Here $\overline{\boldsymbol{W}}(t) := \lim_{\alpha \to 0} \boldsymbol{W}_\alpha^{\mathrm{G}}(t)$ is the trajectory of deep GLRL, where $\boldsymbol{W}_\alpha^{\mathrm{G}}(t) := \phi(\alpha \boldsymbol{e}_1 \boldsymbol{e}_1^\top, \frac{\alpha^{-(1-1/P)}}{2\mu_1(P-1)} + t)$ (see Algorithm 2). The dynamics for general initialization is more complicated. Please see discussions in Appendix L.

**Theorem 6.2.** Let $P = \frac{L}{2}$, $L \ge 3$. Suppose $\|\nabla f(\boldsymbol{0})\|_2 = \lambda_1(-\nabla f(\boldsymbol{0})) > \max\{\lambda_2(-\nabla f(\boldsymbol{0})), 0\}$,[1]

$$\text{for every fixed } t \in \mathbb{R}, \quad \left\| \phi\left(\alpha \boldsymbol{I}, \frac{\alpha^{-(1-1/P)}}{2\mu_1(P-1)} + t\right) - \overline{\boldsymbol{W}}(t) \right\|_{\mathrm{F}} = O(\alpha^{\frac{1}{P(P+1)}}), \tag{12}$$

and for any $2 \le k \le d$,

$$\text{for every fixed } t \in \mathbb{R}, \quad \lambda_k\left(\phi\left(\alpha \boldsymbol{I}, \frac{\alpha^{-(1-1/P)}}{2\mu_1(P-1)} + t\right)\right) = O(\alpha). \tag{13}$$

**So how does depth encourage GF to find low-rank solutions?** When the ground truth is low-rank, say rank-$k$, our experiments (Figure 2) suggest that GF with small initialization finds solutions with smaller $k$-low-rankness compared to the depth-2 case, thus achieving better generalization. At first glance, this is contradictory to what Theorem 6.2 suggests, i.e., the convergence rate of deep GLRL at a constant time gets slower as the depth increases. However, it turns out the uniform upper bound for the distance between GF and GLRL is not the ideal metric for the eventual $k$-low-rankness of learned solution. Below we will illustrate why *the $r$-low-rankness of GF within each phase $r$ is a better metric* and how they are different.

**Definition 6.3** ($r$-low-rankness). For matrix $\boldsymbol{M} \in \mathbb{R}^{d \times d}$, we define the *$r$-low-rankness* of $\boldsymbol{M}$ as $\sqrt{\sum_{i=r+1}^d \sigma_i^2(\boldsymbol{M})}$, where $\sigma_i(\boldsymbol{M})$ is the $i$-th largest singular value of $\boldsymbol{M}$.

---

[1]We believe assumption $\|\nabla f(\boldsymbol{0})\|_2 = \lambda_1(-\nabla f(\boldsymbol{0}))$ could be removed with a more refined analysis.

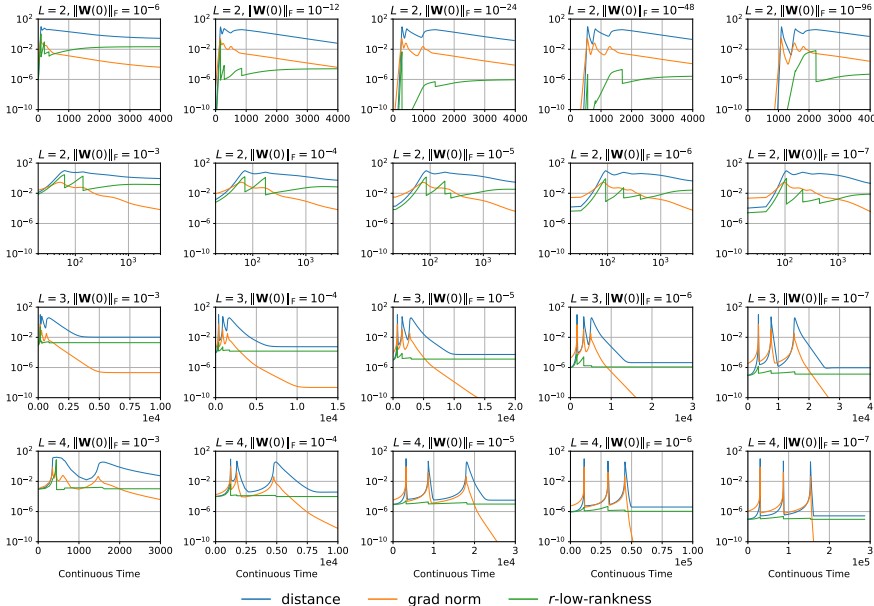

Figure 2: GD passes by the same set of critical points as GLRL when the initialization scale is small, and gets much closer to the critical points when $L \geq 3$. Depth-2 GD requires a much smaller initialization scale to maintain small low-rankness. Here the ground truth matrix $\boldsymbol{W}^* \in \mathbb{R}^{20 \times 20}$ is of rank 3 as stated in Appendix E.1. In this case, GLRL has 3 phases and 4 critical points $\{\overline{\boldsymbol{W}}_r\}_{r=0}^3$, where $\overline{\boldsymbol{W}}_0 = \boldsymbol{0}$ and $\overline{\boldsymbol{W}}_3 = \boldsymbol{W}^*$. For each depth $L$ and initialization scale $\|\boldsymbol{W}(0)\|_F$, we plot the distance between the current step of GD and the closest critical point of GLRL, $\|\boldsymbol{W}_{GD}(t) - \overline{\boldsymbol{W}}_r\|_F$, the norm of full gradient, $\|\nabla_{\boldsymbol{U}_{1:L}} \mathcal{L}(\boldsymbol{U}_{1:L})\|_F$ and the $(r+1)$-low-rankness of $\boldsymbol{W}_{GD}(t)$ with $r := \arg\min_{0 \leq i \leq 3} \|\boldsymbol{W}_{GD}(t) - \overline{\boldsymbol{W}}_i\|_F$.

Suppose $f(\cdot)$ admits a unique minimizer $\boldsymbol{W}_0$ in $\mathbb{S}_{d,1}^+$, and we run GF from $\alpha\boldsymbol{I}$ for both depth-2 and depth-$L$ cases. Intuitively, the 1-low-rankness of the depth-2 solution is $\Omega(\alpha^{1-\mu_2/\mu_1})$, which can be seen from the second warmup example in Section 4. For the depth-$L$ solution, though it may diverge from the trajectory of deep GLRL more than the depth-2 solution does, its 1-low-rankness is only $O(\alpha)$, as shown in Theorem 6.4. The key idea is to show that there is a basin in the manifold of rank-1 matrices around $\boldsymbol{W}_0$ such that any GF starting within the basin converges to $\boldsymbol{W}_0$. Based on this, we can prove that starting from any matrix $O(\alpha)$-close to the basin, GF converges to a solution $O(\alpha)$-close to $\boldsymbol{W}_0$. See Appendix M for more details.

**Theorem 6.4.** *In the same settings as Theorem 6.2, if $\overline{\boldsymbol{W}}(\infty)$ exists and is a minimizer of $f(\cdot)$ in $\mathbb{S}_{d,\leq 1}^+$, under regularity assumption M.1, we have $\inf_{t\in\mathbb{R}} \left\|\phi(\alpha\boldsymbol{I},t) - \overline{\boldsymbol{W}}(\infty)\right\|_F = O(\alpha)$.*

**Interpretation for the advantage of depth with multiple phases.** For depth-2 GLRL, the low-rankness is raised to some power less then 1 per phase (depending on the eigengap). For deep GLRL, we show the low-rankness is only multiplied by some constant for the first phase and speculate it to be true for later phases. This conjecture is supported by our experiments; see Figure 2. Interestingly, our theory and experiments (Figure 5) suggest that while **being deep is good for generalization, being much deeper may not be much better**: once $L \geq 3$, increasing the depth does not improve the order of low-rankness significantly. While this theoretical result is only for identity initialization, Theorem F.1 and Corollary F.2 further show that the dynamics of GF (11) with any initialization pointwise converges as $L \to \infty$, under a suitable time rescaling. See Figure 6 for experimental verification.

## 7 CONCLUSION AND FUTURE DIRECTIONS

In this work, we connect gradient descent to Greedy Low-Rank Learning (GLRL) to explain the success of using gradient descent to find low-rank solutions in the matrix factorization problem. This enables us to construct counterexamples to the implicit nuclear norm conjecture in (Gunasekar et al., 2017). Taking the view of GLRL can also help us understand the benefits of depth.

### ACKNOWLEDGMENTS

The authors thank Sanjeev Arora and Jason D. Lee for helpful discussions. The authors also thank Runzhe Wang for useful suggestions on writing. ZL and YL acknowledge support from NSF, ONR, Simons Foundation, Schmidt Foundation, Mozilla Research, Amazon Research, DARPA and SRC. ZL is also supported by Microsoft PhD Fellowship.

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

## A  COMPARISON TO EXISTING GREEDY ALGORITHMS FOR RANK-CONSTRAINED OPTIMIZATION

The most related one to GLRL (Algorithm 1) is probably *Rank-1 Matrix Pursuit* (R1MP) proposed by Wang et al. (2014) for matrix completion, which was later generalized to general convex loss in (Yao and Kwok, 2016). R1MP maintains a set of rank-1 matrices as the basis, and in phase $r$, R1MP adds the same $\boldsymbol{u}_r \boldsymbol{u}_r^\top$ as defined in Algorithm 1 into its basis and solve $\min_{\boldsymbol{\alpha}} f(\sum_{i=1}^r \alpha_i \boldsymbol{u}_i \boldsymbol{u}_i^\top)$ for rank-$r$ estimation. The main difference between R1MP and GLRL is that the optimization in each phase of R1MP is performed on the coefficients $\boldsymbol{\alpha}$, while the entire $\boldsymbol{U}_r$ evolves with GD in each phase of GLRL. In Figure 3, we provide empirical evidence that GLRL generalizes better than R1MP when ground truth is low-rank, although GLRL may have a higher computational cost depending on $\eta, \epsilon$.

Similar to R1MP, Greedy Efficient Component Optimization (GECO, Shalev-Shwartz and Singer 2010) also chooses the $r$-th component of its basis as the top eigenvector of $-\nabla f(\boldsymbol{W}_r)$, while it solves $\min_{\boldsymbol{\beta}} f(\sum_{1 \le i,j \le r} \beta_{ij} \boldsymbol{u}_i \boldsymbol{u}_j^\top)$ for the rank-$r$ estimation. Khanna et al. (2017) provided convergence guarantee for GECO assuming strong convexity. Haeffele and Vidal (2019) proposed a local-descent meta algorithm, of which GLRL can be viewed as a specific realization.

## B  DEEP GLRL ALGORITHM

---

**Algorithm 2:** Deep Greedy Low-Rank Learning (Deep GLRL)

---

**parameter :** step size $\eta > 0$; small $\epsilon > 0$

$\epsilon' \leftarrow \epsilon^{1/L}$, $\mathcal{L}(\boldsymbol{U}_1, \cdots, \boldsymbol{U}_L) := f(\boldsymbol{W}_1 \cdots \boldsymbol{W}_L)$.

$\boldsymbol{W}_0 \leftarrow \boldsymbol{0} \in \mathbb{R}^{d \times d}$, and $\boldsymbol{U}_{0,1}(\infty), \ldots, \boldsymbol{U}_{0,L}(\infty) \in \mathbb{R}^{d \times 0}$ are empty matrices

**while** $\lambda_1(-\nabla f(\boldsymbol{W}_r)) > 0$ **do**

    $r \leftarrow r + 1$

    let $\boldsymbol{u}_r$ be a top (unit) eigenvector of $-\nabla f(\boldsymbol{W}_{r-1})$

    $\boldsymbol{U}_{r,1}(0) \leftarrow [\boldsymbol{U}_{r-1,1}(\infty) \quad \epsilon' \boldsymbol{u}_r] \in \mathbb{R}^{d \times r}$

    $\boldsymbol{U}_{r,k}(0) \leftarrow \begin{bmatrix} \boldsymbol{U}_{r-1,k}(\infty) & \boldsymbol{0} \\ \boldsymbol{0} & \epsilon' \end{bmatrix} \in \mathbb{R}^{r \times r}$ for all $2 \le k \le L-1$

    $\boldsymbol{U}_{r,L}(0) \leftarrow \begin{bmatrix} \boldsymbol{U}_{r-1,L}(\infty) \\ \epsilon' \boldsymbol{u}_r^\top \end{bmatrix} \in \mathbb{R}^{r \times d}$

    **for** $t = 0, 1, \ldots$ **do**

        $\boldsymbol{U}_{r,i}(t+1) \leftarrow \boldsymbol{U}_{r,i}(t) - \eta \nabla_{\boldsymbol{U}_i} \mathcal{L}(\boldsymbol{U}_{r,1}(t), \cdots, \boldsymbol{U}_{r,L}(t)), \forall 1 \le i \le L.$

    $\boldsymbol{W}_r \leftarrow \boldsymbol{U}_{r,1}(\infty) \cdots \boldsymbol{U}_{r,L}(\infty)$

**return** $\boldsymbol{W}_r$

---

## C  PRELIMINARY LEMMAS

**Lemma C.1.** *For $\boldsymbol{U}_0 \in \mathbb{R}^{d \times r}$ and $\boldsymbol{W}_0 := \boldsymbol{U}_0 \boldsymbol{U}_0^\top$, the following statements are equivalent:*

    *(1). $\boldsymbol{U}_0$ is a stationary point of $\mathcal{L}(\boldsymbol{U}) = \frac{1}{2} f(\boldsymbol{U} \boldsymbol{U}^\top)$;*

    *(2). $\nabla f(\boldsymbol{W}_0) \boldsymbol{W}_0 = \boldsymbol{0}$;*

    *(3). $\boldsymbol{W}_0 := \boldsymbol{U}_0 \boldsymbol{U}_0^\top$ is a critical point of (2).*

*Proof.* (2) $\Rightarrow$ (3) is trivial. We only prove (1) $\Rightarrow$ (2), (3) $\Rightarrow$ (1).

**Proof for (1) $\Rightarrow$ (2).** If $\boldsymbol{U}_0$ is a stationary point, then $\boldsymbol{0} = \nabla \mathcal{L}(\boldsymbol{U}_0) = \nabla f(\boldsymbol{W}_0) \boldsymbol{U}_0$. So

$$\nabla f(\boldsymbol{W}_0) \boldsymbol{W}_0 = (\nabla f(\boldsymbol{W}_0) \boldsymbol{U}_0) \boldsymbol{U}_0^\top = \boldsymbol{0}.$$

**Proof for (3) $\Rightarrow$ (1).** If $\boldsymbol{W}_0$ is a critical point, then

$$0 = \langle \boldsymbol{g}(\boldsymbol{W}_0), \nabla f(\boldsymbol{W}_0) \rangle = -2 \operatorname{Tr}(\nabla f(\boldsymbol{W}_0) \boldsymbol{W}_0 \nabla f(\boldsymbol{W}_0)) = -2 \|\nabla f(\boldsymbol{W}_0) \boldsymbol{U}_0\|_{\mathrm{F}}^2,$$

which implies $\nabla \mathcal{L}(\boldsymbol{U}_0) = \boldsymbol{0}$. $\qquad\square$

**Lemma C.2.** *For a stationary point $U_0 \in \mathbb{R}^{d \times r}$ of $\mathcal{L}(U) = \frac{1}{2} f(UU^\top)$ where $f(\cdot)$ is convex, $W_0 := U_0 U_0^\top$ attains the global minimum of $f(\cdot)$ in $\mathbb{S}_d^+ := \{W : W \succeq 0\}$ iff $\nabla f(W_0) \succeq 0$.*

*Proof.* Since $f(W)$ is a convex function and $\mathbb{S}_d^+$ is convex, we know that $W_0$ is a global minimizer of $f(W)$ in $\mathbb{S}_d^+$ iff

$$\langle \nabla f(W_0), W - W_0 \rangle \geq 0, \qquad \forall W \succeq 0. \tag{14}$$

Note that $\langle \nabla f(W_0), W_0 \rangle = \text{Tr}(\nabla f(W_0) W_0)$. By Lemma C.1, $\langle \nabla f(W_0), W_0 \rangle = 0$. Combining this with (14), we know that $W_0$ is a global minimizer iff

$$\langle \nabla f(W_0), W \rangle \geq 0, \qquad \forall W \succeq 0. \tag{15}$$

It is easy to check that this condition is equivalent to $\nabla f(W_0) \succeq 0$. $\qquad \square$

# D    PROOFS FOR COUNTER-EXAMPLE

**Conjecture D.1** (Formal Statement, Gunasekar et al. 2017). Suppose $f : \mathbb{R}^{d \times d} \to \mathbb{R}$ is a quadratic function and $\min_{W \succeq 0} f(W) = 0$. Then for any $W_{\text{init}} \succ 0$ if $\overline{W}_1 = \lim_{\alpha \to 0} \lim_{t \to +\infty} \phi(\alpha W_{\text{init}}, t)$ exists and $f(\overline{W}_1) = 0$, then $\|\overline{W}_1\|_* = \min_{W \succeq 0} \|W\|_*$ s.t. $f(W) = 0$.

**Propsition D.2** (Formal Statement for Example 5.9). For constant $R > 1$, let

$$M = \begin{bmatrix} ? & ? & 1 & R \\ ? & ? & R & ? \\ 1 & R & ? & ? \\ R & ? & ? & ? \end{bmatrix}, M_{\text{norm}} = \begin{bmatrix} R & 1 & 1 & R \\ 1 & R & R & 1 \\ 1 & R & R & 1 \\ R & 1 & 1 & R \end{bmatrix}, \text{ and } M_{\text{rank}} = \begin{bmatrix} 1 & R & 1 & R \\ R & R^2 & R & R^2 \\ 1 & R & 1 & R \\ R & R^2 & R & R^2 \end{bmatrix}.$$

and

$$\mathcal{L}(U) = \frac{1}{2} f(UU^\top), \quad f(W) = \frac{1}{2} \sum_{(i,j) \in \Omega} (W_{ij} - M_{ij})^2$$

where $\Omega = \{(1,3), (1,4), (2,3), (3,1), (3,2), (4,1)\}$.

Then for any $W_{\text{init}} \succeq 0$, s.t. $u_1^\top W_{\text{init}} u_1 > 0$,

$$\lim_{\alpha \to 0} \lim_{t \to +\infty} \phi(\alpha W_{\text{init}}, t) = M_{\text{rank}}.$$

Moreover, we have

$$\|M_{\text{rank}}\|_* = 2R^2 + 2 > 4R = \|M_{\text{norm}}\|_* = \min_{W \succeq 0, f(W) = 0} \|W\|_*.$$

*Proof.* We define $W_{1,\epsilon}^{\text{G}}(t), W_1^{\text{G}}(t)$ in the same way as in Definition 5.1, Theorem 5.6.

$$W_{1,\epsilon}^{\text{G}}(t) := \phi\left(\epsilon u_1 u_1^\top, t\right),$$
$$W_1^{\text{G}}(t) := \lim_{\epsilon \to 0} W_{1,\epsilon}^{\text{G}}\left(\frac{1}{2\mu_1} \log \frac{1}{\epsilon} + t\right).$$

Below we will show

1. Assumption 5.7 and Assumption 5.5 are satisfied.

2. $\left\|W_1^{\text{G}}(t)\right\|_{\text{F}}$ bounded for $t \geq 0$;

3. $\lim_{t \to +\infty} W_1^{\text{G}}(t) = M_{\text{rank}}$;

4. $M_{\text{norm}} = \arg\min_{W \succeq 0, f(W) = 0} \|W\|_*$.

Thus Since $\boldsymbol{M}_{\mathrm{rank}}$ is a global minimizer of $f(\,\cdot\,)$, applying Theorem 5.8 finishes the proof.

**Proof for Item 1.** Let $\boldsymbol{M}_0 := \nabla f(\mathbf{0})$, then

$$\boldsymbol{M}_0 = \begin{bmatrix} 0 & 0 & 1 & R \\ 0 & 0 & R & 0 \\ 1 & R & 0 & 0 \\ R & 0 & 0 & 0 \end{bmatrix}.$$

Let $\boldsymbol{A} := \left[\begin{smallmatrix} 1 & R \\ R & 0 \end{smallmatrix}\right]$, then we have $\lambda_1(\boldsymbol{A}) = \frac{1+\sqrt{1+R^2}}{2}$, $\lambda_2(\boldsymbol{A}) = \frac{1-\sqrt{1+R^2}}{2}$, thus $\lambda_1(\boldsymbol{A}) > |\lambda_2(\boldsymbol{A})| > 0 > \lambda_2(\boldsymbol{A})$. As a result, $\lambda_1(\boldsymbol{A}) = \|\boldsymbol{A}\|_2$. Let $\boldsymbol{v}_1 \in \mathbb{R}^2$ be the top eigenvector of $\boldsymbol{A}$. We claim that $\boldsymbol{u}_1 = \left[\begin{smallmatrix} \boldsymbol{v}_1 \\ \boldsymbol{v}_1 \end{smallmatrix}\right] \in \mathbb{R}^4$ is the top eigenvector of $\nabla f(\mathbf{0})$. First by definition it is easy to check that $\boldsymbol{M}_0\boldsymbol{u}_1 = \lambda_1(\boldsymbol{A})\boldsymbol{u}_1$. Further noticing that $\boldsymbol{M}_0^2 = \left[\begin{smallmatrix} \boldsymbol{A}^2 & \mathbf{0} \\ \mathbf{0} & \boldsymbol{A}^2 \end{smallmatrix}\right]$, we know $\lambda_i^2(\boldsymbol{M}_0) \in \{\lambda_1^2(\boldsymbol{A}), \lambda_2^2(\boldsymbol{A})\}$ for all eigenvalues $\lambda_i(\boldsymbol{M}_0)$. That is, $\lambda_1(\boldsymbol{M}_0) = \lambda_1(\boldsymbol{A})$, $\lambda_2(\boldsymbol{M}_0) = -\lambda_2(\boldsymbol{A})$, $\lambda_3(\boldsymbol{M}_0) = \lambda_2(\boldsymbol{A})$, and $\lambda_4(\boldsymbol{M}_0) = -\lambda_1(\boldsymbol{A})$. Thus Assumption 5.5 is satisfied. Also note that $f$ is quadratic, thus analytic, i.e., Assumption 5.7 is also satisfied.

**Proof for Item 2.** Let $(x_\epsilon(t), y_\epsilon(t)) \in \mathbb{R}^2$ be the gradient flow of $g(x, y) = \frac{1}{2}(x^2 - 1)^2 + (xy - R)^2$ starting from $(x_\epsilon(0), y_\epsilon(0)) = \sqrt{\epsilon}\boldsymbol{v}_1$.

$$\begin{aligned} \frac{\mathrm{d}x(t)}{\mathrm{d}t} &= (1 - x(t)^2)x(t) - 2y(t)(x(t)y(t) - R) \\ \frac{\mathrm{d}y(t)}{\mathrm{d}t} &= -2x(t)(x(t)y(t) - R) \end{aligned} \tag{16}$$

Let $\boldsymbol{W}_\epsilon(t)$ be the following matrix:

$$\boldsymbol{W}_\epsilon(t) := \begin{bmatrix} x_\epsilon(t) \\ y_\epsilon(t) \\ x_\epsilon(t) \\ y_\epsilon(t) \end{bmatrix} \begin{bmatrix} x_\epsilon(t) & y_\epsilon(t) & x_\epsilon(t) & y_\epsilon(t) \end{bmatrix}.$$

Then it is easy to verify that $\boldsymbol{W}_\epsilon(0) = \boldsymbol{W}_{1,\epsilon}^{\mathrm{G}}(0)$ and $\boldsymbol{W}_\epsilon(t)$ satisfies (2). Thus by the existence and uniqueness theorem, we have $\boldsymbol{W}_\epsilon(t) = \boldsymbol{W}_{1,\epsilon}^{\mathrm{G}}(t)$ for all $t$. Taking the limit $\epsilon \to 0$, we know that $\boldsymbol{W}_1^{\mathrm{G}}(t)$ can also be written in the following form:

$$\boldsymbol{W}_1^{\mathrm{G}}(t) = \begin{bmatrix} x(t) \\ y(t) \\ x(t) \\ y(t) \end{bmatrix} \begin{bmatrix} x(t) & y(t) & x(t) & y(t) \end{bmatrix},$$

and $(x_\epsilon(t), y_\epsilon(t)) \in \mathbb{R}^2$ is a gradient flow of $g(x, y) = \frac{1}{2}(x^2 - 1)^2 + (xy - R)^2$.

Since $g(x(t), y(t))$ is non-increasing overtime, and $\lim_{t \to -\infty} g(x(-t), y(-t)) = g(x(-\infty), y(-\infty)) = g(0, 0) = R^2 + 0.5$, we know $|x(t)y(t)| \leq 3R$ for all $t$. So whenever $y^2(t) - x^2(t) \geq 9R^2$, we have $x^2(t) \leq \frac{9R^2}{y^2(t)} \leq \frac{9R^2}{y^2(t) - x^2(t)} \leq 1$. In this case, $\frac{\mathrm{d}(y^2(t) - x^2(t))}{\mathrm{d}t} = 2x^2(t)(x^2(t) - 1) \leq 0$. Combining this with $y(-\infty)^2 - x(-\infty)^2 = 0 \leq 9R^2$, we have $y^2(t) - x^2(t) \leq 9R^2$ for all $t$, which also implies that $y(t)$ is bounded. Noticing that $9R^2 \geq g(x(t), y(t)) \geq (x^2(t) - 1)^2$, we know $x^2(t)$ is also bounded. Therefore, $\boldsymbol{W}_1^G(t)$ is bounded.

**Proof for Item 3.** Note that $(x(\infty), y(\infty))$ is a stationary point of $g(x, y)$. It is clear that $g(x, y)$ only has 3 stationary points — $(0, 0)$, $(1, R)$ and $(-1, -R)$. Thus $\overline{\boldsymbol{W}}_1$ can only be $\mathbf{0}$ or $\boldsymbol{M}_{\mathrm{rank}}$. However, since for all $t$, $f(\boldsymbol{W}_1^{\mathrm{G}}(t)) < f(\mathbf{0})$, $\overline{\boldsymbol{W}}_1 = \lim_{t \to \infty} \boldsymbol{W}_1^{\mathrm{G}}(t)$ cannot be $\mathbf{0}$. So $\overline{\boldsymbol{W}}_1$ must be $\boldsymbol{M}_{\mathrm{rank}}$.

**Proof for Item 4.** Let $m_{ij}$ be $(i, j)$th element of $\boldsymbol{M}$. Suppose $\boldsymbol{M} \succeq \mathbf{0}$, we have

$$\begin{aligned} (\boldsymbol{e}_1 - \boldsymbol{e}_4)^\top \boldsymbol{M}(\boldsymbol{e}_1 - \boldsymbol{e}_4) \geq 0 &\implies m_{11} + m_{44} \geq m_{14} + m_{41} = 2R \\ (\boldsymbol{e}_2 - \boldsymbol{e}_3)^\top \boldsymbol{M}(\boldsymbol{e}_2 - \boldsymbol{e}_3) \geq 0 &\implies m_{22} + m_{33} \geq m_{23} + m_{32} = 2R \end{aligned}$$

| Depth ($L$) | Simulation method |
|:---:|:---:|
| 2 | Constant LR, $\eta = 10^{-3}$ for $10^6$ iterations |
| 3 | Adaptive LR, $\eta = 2 \times 10^{-5}$ and $\varepsilon = 10^{-4}$ for $10^6$ iterations |
| 4 | Adaptive LR, $\eta = 3 \times 10^{-4}$ and $\varepsilon = 10^{-3}$ for $10^6$ iterations |

Table 1: Choice of hyperparameters for simulating gradient flow. For $L = 2$, gradient descent escapes saddles in $O(\log \frac{1}{\epsilon})$ time, where $\epsilon$ is the distance between the initialization and the saddle.

Thus $4R = \min_{\boldsymbol{W} \succeq \boldsymbol{0}, f(\boldsymbol{W})=0} \|\boldsymbol{W}\|_*$, where the equality is only attained at $m_{ii} = R, i = 1, 2, 3, 4$. Otherwise, either $\begin{bmatrix} m_{11} & m_{14} \\ m_{41} & m_{44} \end{bmatrix}$ or $\begin{bmatrix} m_{22} & m_{23} \\ m_{32} & m_{33} \end{bmatrix}$ will have negative eigenvalues. Contradiction to that $\boldsymbol{M} \succeq \boldsymbol{0}$.

Below we will show the rest unknown off-diagonal entries must be 1. Let $\boldsymbol{V} = \begin{bmatrix} 1 & -1 & 0 & 0 \\ 0 & 0 & 1 & 0 \\ 0 & 0 & 0 & 1 \end{bmatrix}$, then

$$\boldsymbol{M} \succeq \boldsymbol{0} \Longrightarrow \boldsymbol{V}\boldsymbol{M}\boldsymbol{V}^\top \succeq \boldsymbol{0} \Longrightarrow \begin{bmatrix} 0 & m_{13} - m_{23} & m_{14} - m_{24} \\ m_{31} - m_{32} & R & R \\ m_{41} - m_{42} & R & R \end{bmatrix} \succeq \boldsymbol{0},$$

which implies $m_{13} = m_{23}, m_{14} = m_{24}$.

With the same argument for $\boldsymbol{V} = \begin{bmatrix} 1 & 0 & 0 & 0 \\ 0 & 1 & 0 & 0 \\ 0 & 0 & 1 & -1 \end{bmatrix}$, we have $m_{13} = m_{14}, m_{23} = m_{24}$. Also note $\boldsymbol{M}$ is symmetric and $m_{13} = 1$, thus $m_{ij} = m_{ji} = 1, \forall i = 1, 2, j = 3, 4$. Thus $\boldsymbol{M}_{\mathrm{norm}} = \arg\min_{\boldsymbol{W} \succeq \boldsymbol{0}, f(\boldsymbol{W})=0} \|\boldsymbol{W}\|_*$, which is unique. □

# E    EXPERIMENTS

## E.1    GENERAL SETUP

The code is written in Julia (Bezanson et al., 2012) and PyTorch (Paszke et al., 2019).

The ground-truth matrix $\boldsymbol{W}_*$ is low-rank by construction: we sample a random orthogonal matrix $\boldsymbol{U}$, a diagonal matrix $\boldsymbol{S}$ with Frobenius norm $\|\boldsymbol{S}\|_\mathrm{F} = 1$ and set $\boldsymbol{W}_* = \boldsymbol{U}\boldsymbol{S}\boldsymbol{U}^\top$. Each measurement $\boldsymbol{X}$ in $\boldsymbol{X}_1, \ldots, \boldsymbol{X}_m$ is generated by sampling two one-hot vectors $\boldsymbol{u}$ and $\boldsymbol{v}$ uniformly and setting $\boldsymbol{X} = \frac{1}{2}\boldsymbol{u}\boldsymbol{v}^\top + \frac{1}{2}\boldsymbol{v}\boldsymbol{u}^\top$.

In Figures 1, 2, 3 to 5 and 7, the ground truth matrix $\boldsymbol{W}_*$ has shape $20 \times 20$ and rank 3, where $\|\boldsymbol{W}^*\|_\mathrm{F} = 20$, $\lambda_1(\boldsymbol{W}^*) = 17.41$, $\lambda_2(\boldsymbol{W}^*) = 8.85$, $\lambda_3(\boldsymbol{W}^*) = 4.31$ and $\lambda_1(-\nabla f(\boldsymbol{0})) = 6.23, \lambda_2(-\nabla f(\boldsymbol{0})) = 5.41$. $p = 0.3$ is used for generating measurements, except $p = 0.25$ in Figure 3, i.e., each pair of entries of $\boldsymbol{W}^*_{ij}$ and $\boldsymbol{W}^*_{ji}$ is observed with probability $p$.

**Gradient Descent.**    Let $\tilde{\epsilon} > 0$ be the Frobenius norm of the target random initialization. For the depth-2 case, we sample 2 orthogonal matrices $\boldsymbol{V}_1, \boldsymbol{V}_2$ and a diagonal matrix $\boldsymbol{D}$ with Frobenius norm $\tilde{\epsilon}$, and we set $\boldsymbol{U} = \boldsymbol{V}_1 \boldsymbol{D}^{1/2} \boldsymbol{V}_2^\top$; for the depth-$L$ case with $L \geq 3$, we sample $L$ orthogonal matrices $\boldsymbol{V}_1, \ldots, \boldsymbol{V}_L$ and a diagonal matrix $\boldsymbol{D}$ with Frobenius norm $\tilde{\epsilon}$, and we set $\boldsymbol{U}_i := \boldsymbol{V}_i \boldsymbol{D}^{1/L} \boldsymbol{V}_{i+1}^\top$ ($\boldsymbol{V}_{L+1} = \boldsymbol{V}_1$). In this way, we can guarantee that the end-to-end matrix $\boldsymbol{W} = \boldsymbol{U}_1 \cdots \boldsymbol{U}_L$ is symmetric and the initialization is balanced for $L \geq 3$.

We discretize the time to simulate gradient flow. When $L > 2$, gradient flow stays around saddle points for most of the time, therefore we use full-batch GD with adaptive learning rate $\tilde{\eta}_t$, inspired by

RMSprop (Tieleman and Hinton, 2012), for faster convergence:

$$v_{t+1} = \alpha v_t + (1 - \alpha) \left\| \nabla \mathcal{L}(\boldsymbol{\theta}_t) \right\|_2^2,$$

$$\tilde{\eta}_t = \frac{\eta}{\sqrt{\frac{v_{t+1}}{1 - \alpha^{t+1}} + \varepsilon}},$$

$$\boldsymbol{\theta}_{t+1} = \boldsymbol{\theta}_t - \tilde{\eta}_t \nabla \mathcal{L}(\boldsymbol{\theta}_t),$$

where $\alpha = 0.99$, $\eta$ is the (unadjusted) learning rate. The choices of hyperparameters are summarized in Table 1. The continuous time for $\boldsymbol{\theta}_t$ is measured as $\sum_{i=0}^{t-1} \tilde{\eta}_i$.

**GLRL.** In Figures 1, 2, 3 and 4, the GLRL's trajectory is obtained by running Algorithm 1 with $\epsilon = 10^{-7}$ and $\eta = 10^{-3}$. The stopping criterion is that if the loop has been iterated for $10^7$ times.

### E.2 EXPERIMENTAL EQUIVALENCE BETWEEN GLRL AND GRADIENT DESCENT

Here we provide experimental evidence supporting our theoretical claims about the equivalence between GLRL and GF for both cases, $L = 2$ and $L \geq 3$.

In Figure 1, we show the distance from every point on GF (simulated by GD) from random initialization is close to the trajectory of GLRL. In Figure 2, we first run GLRL and obtain the critical points $\{\overline{\boldsymbol{W}}_r\}_{r=0}^3$ passed by GLRL. We also define the distance of a matrix $\boldsymbol{W}$ to the critical points to be $\min_{0 \leq r \leq 3} \left\| \boldsymbol{W} - \overline{\boldsymbol{W}}_r \right\|_{\mathrm{F}}$.

### E.3 HOW WELL DOES GLRL WORK?

We compare GLRL with gradient descent (with not-so-small initialization), nuclear norm minimization and R1MP (Wang et al., 2014). We use CVXPY (Diamond and Boyd, 2016; Agrawal et al., 2018) for finding the nuclear norm solution. The results are shown in Figure 3. GLRL can fully recover the ground truth, while others have difficulty doing so.

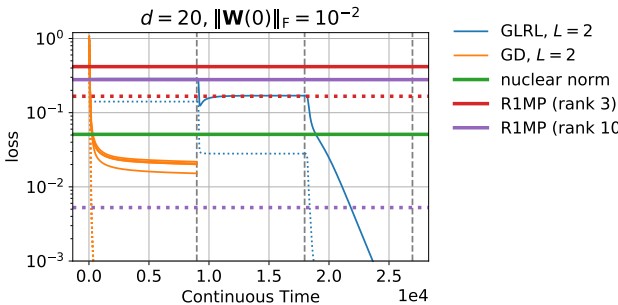

Figure 3: GD with small initialization outperforms R1MP and minimal nuclear norm solution on synthetic data with low-rank ground truth. Solid (dotted) curves correspond to test (training) loss. Here the loss $f(\boldsymbol{W}) := \frac{1}{d^2} \left\| \boldsymbol{W} - \boldsymbol{W}^* \right\|_{\mathrm{F}}^2$ and $f(\boldsymbol{0}) = 1$. We run 10 random seeds for GD and plot them separately (most of them overlap).

### E.4 HOW DOES INITIALIZATION AFFECT THE CONVERGENCE RATE TO THE RANK-1 GLRL TRAJECTORY?

We use the general setting in Appendix E.1. In these experiments, we use the constant learning rate $10^{-5}$ for $4 \times 10^7$ iterations. The reference matrix $\boldsymbol{W}_{\mathrm{ref}}$ is obtained by running the first stage of GLRL with $\left\| \boldsymbol{W}(0) \right\|_{\mathrm{F}} = 10^{-48}$ and we pick one matrix in the trajectory with $\left\| \boldsymbol{W}_{\mathrm{ref}} \right\|_{\mathrm{F}}$ about 0.6.

For every $\epsilon = 10^i$, $i \in \{-1, -2, -3, -4, -5\}$, we run both gradient descent and the first phase of GLRL with $\left\| \boldsymbol{W}(0) \right\|_{\mathrm{F}} = \epsilon$. For gradient descent, we use random initialization so $\left\| \boldsymbol{W}(0) \right\|_{\mathrm{F}}$ is full rank w.p. 1. The distance of a trajectory to $\boldsymbol{W}_{\mathrm{ref}}$ is defined as $\min_{t \geq 0} \left\| \boldsymbol{W}(t) - \boldsymbol{W}_{\mathrm{ref}} \right\|_{\mathrm{F}}$. In practice,

as we discretized time to simulate gradient flow, we check every $t$ during simulation to compute the distance. As a result, the estimation might be inaccurate when a trajectory is really close to $\boldsymbol{W}_{\text{ref}}$.

The result is shown at Figure 4. We observe that GLRL trajectories are closer to the reference matrix $\boldsymbol{W}_{\text{ref}}$ by magnitudes. Thus the take home message here is that GLRL is in general a more computational efficient method to simulate the trajectory of GF (GD) with infinitesimal initialization, as one can start GLRL with a much larger initialization, while still maintaining high precision.

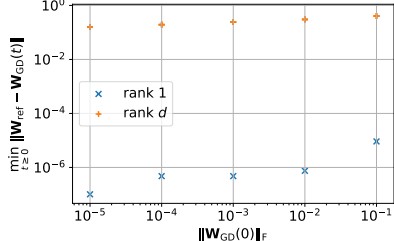

Figure 4: Using $\epsilon \boldsymbol{v}_1 \boldsymbol{v}_1^\top$ (denoted by "rank 1") as initialization makes GD much closer to GLRL compared to using random initialization (denoted by "rank $d$"), where $\boldsymbol{v}_1$ is the top eigenvector of $-\nabla f(\boldsymbol{0})$. We take a fixed reference matrix on the trajectory of GLRL with constant norm and plot the distance of GD with each initialization to it respectively..

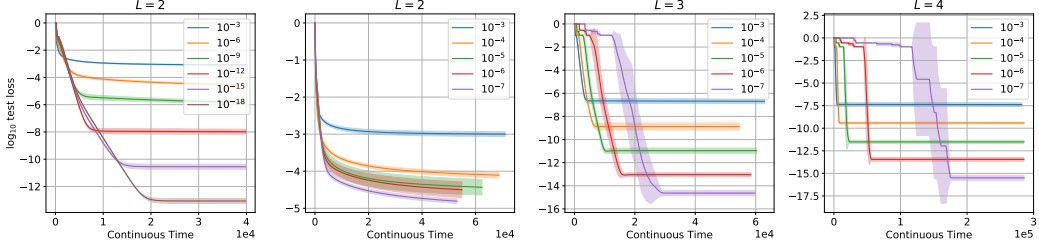

Figure 5: Deep matrix factorization encourages GF to find low rank solutions at a much practical initialization scale, e.g. $10^{-3}$. Here the ground truth is rank-3. For each setting, we run 5 different random seeds. The solid curves are the mean and the shaded area indicates one standard deviation. We observe that performance of GD is quite robust to its initialization. Note that for $L > 2$, the shaded area with initialization scale $10^{-7}$ is large, as the sudden decrement of loss occurs at quite different continuous times for different random seeds in this case.

### E.5 BENEFIT OF DEPTH: POLYNOMIAL VS EXPONENTIAL DEPENDENCE ON INITIALIZATION

To verify the our theory in Section 6, we run gradient descent with different depth and initialization. The results are shown in Figure 5. We can see that as the initialization becomes smaller, the final solution gets closer to the ground truth. However, a depth-2 model requires exponentially small initialization, while deeper models require polynomial small initialization, though it takes much longer to converge.

## F THE MARGINAL VALUE OF BEING DEEPER

Theorem F.1 shows that the end-to-end dynamics (17) converges point-wise while $L \to \infty$ if the product of learning rate and depth, $\eta L$, is fixed as constant. Interestingly, (17) also allows us to simulate the dynamics of $\boldsymbol{W}(t)$ for all depths $L$ while the computation time is independent of $L$. In Figure 6, we compare the effect of depth while fixing the initialization and $\eta L$. We can see that deeper models converge faster. The difference between $L = 1, 2$, and 4 is large, while difference among $L \geq 16$ is marginal.

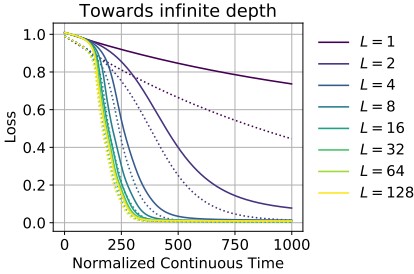

Figure 6: The marginal value of being deeper. The trajectory of GD converges when depth goes to infinity. Solid (dotted) curves correspond to test (train) loss. The $x$-axis stands for the normalized continuous time $t$ (multiplied by $L$).

**Theorem F.1.** *Suppose* $\boldsymbol{W} = \tilde{\boldsymbol{U}}\tilde{\boldsymbol{\Sigma}}\tilde{\boldsymbol{V}}^\top$ *is the SVD decomposition of* $\boldsymbol{W}$*, where* $\tilde{\boldsymbol{\Sigma}} = \mathrm{diag}(\sigma_1, \ldots, \sigma_d)$*. The dynamics of $L$-layer linear net is the following, $\circ$ denotes the entry-wise multiplication:*

$$\frac{\mathrm{d}\boldsymbol{W}}{\mathrm{d}t} = -L\tilde{\boldsymbol{U}}\left(\left(\tilde{\boldsymbol{U}}^\top \nabla f(\boldsymbol{W})\tilde{\boldsymbol{V}}\right) \circ \boldsymbol{K}^{(L)}\right)\tilde{\boldsymbol{V}}^\top, \tag{17}$$

*where* $K_{i,i}^{(L)} = \sigma_i^{2-2/L}$*,* $K_{i,j}^{(L)} = \frac{\sigma_i^2 - \sigma_j^2}{L\sigma_i^{2/L} - L\sigma_j^{2/L}}$ *for* $i \neq j$*.*

*Proof.* We start from (11):

$$\frac{\mathrm{d}\boldsymbol{W}}{\mathrm{d}t} = -\sum_{l=0}^{L-1} (\boldsymbol{W}\boldsymbol{W}^\top)^{\frac{l}{L}} \nabla f(\boldsymbol{W}) (\boldsymbol{W}^\top \boldsymbol{W})^{\frac{L-1-l}{L}}$$

$$= -\sum_{l=0}^{L-1} \tilde{\boldsymbol{U}}\tilde{\boldsymbol{\Sigma}}^{\frac{2l}{L}}\tilde{\boldsymbol{U}}^\top \nabla f(\boldsymbol{W})\tilde{\boldsymbol{V}}\tilde{\boldsymbol{\Sigma}}^{\frac{2(L-1-l)}{L}}\tilde{\boldsymbol{V}}$$

$$= -L\tilde{\boldsymbol{U}}\left[L^{-1}\sum_{l=0}^{L-1} \tilde{\boldsymbol{\Sigma}}^{\frac{2l}{L}}(\tilde{\boldsymbol{U}}^\top \nabla f(\boldsymbol{W})\tilde{\boldsymbol{V}})\tilde{\boldsymbol{\Sigma}}^{\frac{2(L-1-l)}{L}}\right]\tilde{\boldsymbol{V}}.$$

Note that $\tilde{\boldsymbol{\Sigma}}$ is diagonal, so

$$\tilde{\boldsymbol{\Sigma}}^{\frac{2l}{L}}(\tilde{\boldsymbol{U}}^\top \nabla f(\boldsymbol{W})\tilde{\boldsymbol{V}})\tilde{\boldsymbol{\Sigma}}^{\frac{2(L-1-l)}{L}} = (\tilde{\boldsymbol{U}}^\top \nabla f(\boldsymbol{W})\tilde{\boldsymbol{V}}) \circ \boldsymbol{H}^{(l)},$$

where $\boldsymbol{H}_{i,j}^{(l)} = \sigma_i^{\frac{2l}{L}}\sigma_j^{\frac{2(L-1-l)}{L}}$. Therefore,

$$L^{-1}\sum_{l=0}^{L-1} \tilde{\boldsymbol{\Sigma}}^{\frac{2l}{L}}(\tilde{\boldsymbol{U}}^\top \nabla f(\boldsymbol{W})\tilde{\boldsymbol{V}})\tilde{\boldsymbol{\Sigma}}^{\frac{2(L-1-l)}{L}} = L^{-1}\sum_{l=0}^{L-1} (\tilde{\boldsymbol{U}}^\top \nabla f(\boldsymbol{W})\tilde{\boldsymbol{V}}) \circ \boldsymbol{H}^{(l)}$$

$$= (\tilde{\boldsymbol{U}}^\top \nabla f(\boldsymbol{W})\tilde{\boldsymbol{V}}) \circ \boldsymbol{K}^{(L)},$$

where $\boldsymbol{K}^{(L)} = L^{-1}\sum_{l=0}^{L-1} \boldsymbol{H}^{(l)}$. Hence,

$$\frac{\mathrm{d}\boldsymbol{W}}{\mathrm{d}t} = -L\tilde{\boldsymbol{U}}\left[(\tilde{\boldsymbol{U}}^\top \nabla f(\boldsymbol{W})\tilde{\boldsymbol{V}}) \circ \boldsymbol{K}^{(L)}\right]\tilde{\boldsymbol{V}}.$$

The entries of $\boldsymbol{K}^{(L)}$ can be directly calculated by

$$K_{i,j}^{(L)} = L^{-1}\sum_{l=0}^{L-1} \sigma_i^{\frac{2l}{L}}\sigma_j^{\frac{2(L-1-l)}{L}} = \begin{cases} \sigma_i^{2-2/L}, & i = j, \\ \frac{\sigma_i^2 - \sigma_j^2}{L\sigma_i^{2/L} - L\sigma_j^{2/L}}, & i \neq j. \end{cases}$$

$\square$

**Corollary F.2.** *As* $L \to \infty$*,* $\boldsymbol{K}^{(L)}$ *converges to* $\boldsymbol{K}^*$*, where* $K_{i,i}^* = \sigma_i^2$*,* $K_{i,j}^* = \frac{\sigma_i^2 - \sigma_j^2}{\ln \sigma_i^2 - \ln \sigma_j^2}$ *for* $i \neq j$*.*

**Experiment details.** We follow the general setting in Appendix E.1. The ground truth $\boldsymbol{W}^*$ is different but is generated in the same manner and has the same shape of $20 \times 20$ and $p = 0.3$ is used for observation generation. We directly apply (17), in which we compute $\tilde{\boldsymbol{V}}$ and $\tilde{\boldsymbol{U}}$ through SVD, to simulate the trajectory together with a constant learning rate of $\frac{10^{-3}}{L}$ for depth $L$. $\boldsymbol{W}(0)$ is sampled from $10^{-3} \times \mathcal{N}(0, \boldsymbol{I}_d)$.

## G  PROOFS FOR DYNAMICAL SYSTEM

In this section, we prove Theorem 5.3 in Section 5.1. In Appendix G.1, we show how to reduce Theorem 5.3 to the case where $\boldsymbol{J}(\boldsymbol{0})$ is exactly a diagonal matrix, then we prove this diagonal case in Appendix G.2. Finally, in Appendix G.3, we discuss how to extend it to the case where $\boldsymbol{J}(\boldsymbol{0})$ is non-diagonalizable.

### G.1  REDUCTION TO THE DIAGONAL CASE

**Theorem G.1.** *If $\boldsymbol{J}(\boldsymbol{0}) = \mathrm{diag}(\tilde{\mu}_1, \ldots, \tilde{\mu}_d)$ is diagonal, then the statement in Theorem 5.3 holds.*

*Proof for Theorem 5.3.* We show how to prove Theorem 5.3 based on Theorem G.1. Let $\frac{\mathrm{d}\boldsymbol{\theta}}{\mathrm{d}t} = \boldsymbol{g}(\boldsymbol{\theta})$ be the dynamical system in Theorem 5.3. Let $\boldsymbol{J}(\boldsymbol{0}) = \tilde{\boldsymbol{V}}\tilde{\boldsymbol{D}}\tilde{\boldsymbol{V}}^{-1}$ be the eigendecomposition, where $\tilde{\boldsymbol{V}}$ is an invertible matrix and $\tilde{\boldsymbol{D}} = \mathrm{diag}(\tilde{\mu}_1, \ldots, \tilde{\mu}_d)$. Now we define the following new dynamics by changing the basis:

$$\hat{\boldsymbol{\theta}}(t) = \tilde{\boldsymbol{V}}^{-1}\boldsymbol{\theta}(t).$$

Then $\frac{\mathrm{d}\hat{\boldsymbol{\theta}}(t)}{\mathrm{d}t} = \hat{\boldsymbol{g}}(\hat{\boldsymbol{\theta}})$ for $\hat{\boldsymbol{g}}(\hat{\boldsymbol{\theta}}) := \tilde{\boldsymbol{V}}^{-1}\boldsymbol{g}(\tilde{\boldsymbol{V}}\hat{\boldsymbol{\theta}})$, and the associated Jacobian matrix is $\hat{\boldsymbol{J}}(\hat{\boldsymbol{\theta}}) := \tilde{\boldsymbol{V}}^{-1}\boldsymbol{J}(\tilde{\boldsymbol{V}}\hat{\boldsymbol{\theta}})\tilde{\boldsymbol{V}}$, and thus $\hat{\boldsymbol{J}}(\boldsymbol{0}) = \mathrm{diag}(\tilde{\mu}_1, \ldots, \tilde{\mu}_d)$.

Now we apply Theorem G.1 to $\hat{\boldsymbol{\theta}}(t)$. Then $\hat{\boldsymbol{z}}_\alpha(t) := \tilde{\boldsymbol{V}}^{-1}\boldsymbol{z}_\alpha(t)$ converges to the limit $\hat{\boldsymbol{z}}(t) := \lim_{\alpha \to 0} \hat{\boldsymbol{z}}_\alpha(t)$. This shows that the limit $\boldsymbol{z}(t) = \tilde{\boldsymbol{V}}\hat{\boldsymbol{z}}(t)$ exists in Theorem 5.3. We can also verify that $\boldsymbol{z}(t)$ is a solution of (6).

Given $\boldsymbol{\delta}_\alpha$ converging to $\boldsymbol{0}$ with positive alignment with $\tilde{\boldsymbol{u}}_1$ as $\alpha \to 0$, we can define $\hat{\boldsymbol{\delta}}_\alpha := \tilde{\boldsymbol{V}}^{-1}\boldsymbol{\delta}_\alpha$, then $\hat{\boldsymbol{\delta}}_\alpha$ converges to $\boldsymbol{0}$ with positive alignment with $\boldsymbol{e}_1$, where $\boldsymbol{e}_1$ is the first vector in the standard basis and is also the top eigenvector of $\hat{\boldsymbol{J}}(\boldsymbol{0})$. Therefore, for every $t \in (-\infty, +\infty)$, there is a constant $C > 0$ such that

$$\left\| \tilde{\boldsymbol{V}}^{-1}\phi\left(\boldsymbol{\delta}_\alpha, t + \frac{1}{\tilde{\mu}_1}\log\frac{1}{\langle\boldsymbol{\delta}_\alpha, \tilde{\boldsymbol{u}}_1\rangle}\right) - \hat{\boldsymbol{z}}(t) \right\|_2 \leq C \cdot \|\hat{\boldsymbol{\delta}}_\alpha\|_2^{\frac{\tilde{\gamma}}{\tilde{\mu}_1 + \tilde{\gamma}}} \tag{18}$$

for every sufficiently small $\alpha$. As $\tilde{\boldsymbol{V}}$ are invertible, this directly implies (7). $\square$

### G.2  PROOF FOR THE DIAGONAL CASE

Now we only need to prove Theorem G.1. Let $\boldsymbol{e}_1, \ldots, \boldsymbol{e}_d$ be the standard basis. Then $\tilde{\boldsymbol{u}}_1 = \tilde{\boldsymbol{v}}_1 = \boldsymbol{e}_1$ in this diagonal case. We only use $\boldsymbol{e}_1$ to stand for $\tilde{\boldsymbol{u}}_1$ and $\tilde{\boldsymbol{v}}_1$ in the rest of our analysis.

Let $R > 0$. Since $\boldsymbol{g}(\boldsymbol{\theta})$ is $\mathcal{C}^2$-smooth, there exists $\beta > 0$ such that

$$\|\boldsymbol{J}(\boldsymbol{\theta}) - \boldsymbol{J}(\boldsymbol{\theta} + \boldsymbol{h})\|_2 \leq \beta\|\boldsymbol{h}\|_2 \tag{19}$$

for all $\|\boldsymbol{\theta}\|_2, \|\boldsymbol{\theta} + \boldsymbol{h}\|_2 \leq R$. Then the following can be proved by integration:

$$\boldsymbol{g}(\boldsymbol{\theta} + \boldsymbol{h}) - \boldsymbol{g}(\boldsymbol{\theta}) = \left(\int_0^1 \boldsymbol{J}(\boldsymbol{\theta} + \xi\boldsymbol{h})\mathrm{d}\xi\right)\boldsymbol{h}, \tag{20}$$

$$\|\boldsymbol{g}(\boldsymbol{\theta} + \boldsymbol{h}) - \boldsymbol{g}(\boldsymbol{\theta}) - \boldsymbol{J}(\boldsymbol{\theta})\boldsymbol{h}\|_2 \leq \beta\|\boldsymbol{h}\|_2^2. \tag{21}$$

By (21), we also have

$$\|\boldsymbol{g}(\boldsymbol{\theta}) - \boldsymbol{J}(\boldsymbol{0})\boldsymbol{\theta}\|_2 = \|\boldsymbol{g}(\boldsymbol{\theta}) - \boldsymbol{g}(\boldsymbol{0}) - \boldsymbol{J}(\boldsymbol{0})\boldsymbol{\theta}\|_2 \leq \beta\|\boldsymbol{\theta}\|_2^2. \tag{22}$$

Let $\kappa := \beta/\tilde{\mu}_1$. We assume WLOG that $R \leq 1/\kappa$. Let $F(x) = \log x - \log(1 + \kappa x)$. It is easy to see that $F'(x) = \frac{1}{x + \kappa x^2}$ and $F(x)$ is an increasing function with range $(-\infty, \log(1/\kappa))$. We use $F^{-1}(y)$ to denote the inverse function of $F(x)$. Define $T_\alpha(r) := \frac{1}{\tilde{\mu}_1}(F(r) - F(\alpha)) = \frac{1}{\tilde{\mu}_1}\left(\log \frac{r}{\alpha} - \log \frac{1+\kappa r}{1+\kappa\alpha}\right)$.

Our proof only relies on the following properties of $\boldsymbol{J}(\mathbf{0})$ (besides that $\tilde{\mu}_1, \boldsymbol{e}_1$ are the top eigenvalue and eigenvector of $\boldsymbol{J}(\mathbf{0})$):

**Lemma G.2.** *For $\boldsymbol{J}(\mathbf{0}) := \operatorname{diag}(\tilde{\mu}_1, \ldots, \tilde{\mu}_d)$, we have*

1. *For any $\boldsymbol{h} \in \mathbb{R}^d$, $\boldsymbol{h}^\top \boldsymbol{J}(\mathbf{0})\boldsymbol{h} \leq \tilde{\mu}_1 \|\boldsymbol{h}\|_2^2$;*

2. *For any $t \geq 0$, $\left\|e^{t\boldsymbol{J}(\mathbf{0})} - e^{\tilde{\mu}_1 t}\boldsymbol{e}_1\boldsymbol{e}_1^\top\right\|_2 = e^{\tilde{\mu}_2 t}$.*

*Proof.* For Item 1, $\boldsymbol{h}^\top \boldsymbol{J}(\mathbf{0})\boldsymbol{h} = \sum_{i=1}^d \tilde{\mu}_i h_i^2 \leq \tilde{\mu}_1 \|\boldsymbol{h}\|_2^2$. For Item 2, $\left\|e^{t\boldsymbol{J}(\mathbf{0})} - e^{\tilde{\mu}_1 t}\boldsymbol{e}_1\boldsymbol{e}_1^\top\right\|_2 = \left\|\operatorname{diag}(0, e^{\tilde{\mu}_2 t}, \ldots, e^{\tilde{\mu}_d t})\right\|_2 = e^{\tilde{\mu}_2 t}$. $\square$

**Lemma G.3.** *For $\boldsymbol{\theta}(t) = \phi(\boldsymbol{\theta}_0, t)$ with $\|\boldsymbol{\theta}_0\|_2 \leq \alpha$ and $t \leq T_\alpha(r)$,*
$$\|\boldsymbol{\theta}(t)\|_2 \leq \frac{1+\kappa r}{1+\kappa\alpha}\alpha \cdot e^{\tilde{\mu}_1 t} \leq r.$$

*Proof.* By (22) and Lemma G.2, we have
$$\frac{1}{2}\frac{\mathrm{d}\|\boldsymbol{\theta}(t)\|_2^2}{\mathrm{d}t} = \langle \boldsymbol{\theta}(t), \boldsymbol{g}(\boldsymbol{\theta}(t))\rangle \leq \langle \boldsymbol{\theta}(t), \boldsymbol{J}(\mathbf{0})\boldsymbol{\theta}(t)\rangle + \beta\|\boldsymbol{\theta}(t)\|_2^3 \leq \tilde{\mu}_1\|\boldsymbol{\theta}(t)\|_2^2 + \beta\|\boldsymbol{\theta}(t)\|_2^3.$$
This implies $\frac{\mathrm{d}\|\boldsymbol{\theta}(t)\|_2}{\mathrm{d}t} \leq \tilde{\mu}_1(\|\boldsymbol{\theta}(t)\|_2 + \kappa\|\boldsymbol{\theta}(t)\|_2^2)$. Since $F'(x) = \frac{1}{x+\kappa x^2}$, we further have
$$\frac{\mathrm{d}}{\mathrm{d}t}F(\|\boldsymbol{\theta}(t)\|_2) \leq \tilde{\mu}_1.$$
So $F(\|\boldsymbol{\theta}(t)\|_2) \leq F(\alpha) + \tilde{\mu}_1 t$. By definition of $T_\alpha(r)$, we then know that $\|\boldsymbol{\theta}(t)\|_2 \leq r$ for all $t \leq T_\alpha(r)$. So
$$\log\|\boldsymbol{\theta}(t)\|_2 \leq F(\|\boldsymbol{\theta}(t)\|_2) + \log(1+\kappa r) \leq F(\alpha) + \tilde{\mu}_1 t + \log(1+\kappa r).$$
Expending $F(\alpha)$ proves the lemma. $\square$

**Lemma G.4.** *For $\boldsymbol{\theta}(t) = \phi(\boldsymbol{\theta}_0, t)$ with $\|\boldsymbol{\theta}_0\|_2 \leq \alpha$ and $t \leq T_\alpha(r)$, we have*
$$\boldsymbol{\theta}(t) = e^{t\boldsymbol{J}(\mathbf{0})}\boldsymbol{\theta}_0 + O(r^2).$$

*Proof.* Let $\hat{\boldsymbol{\theta}}(t) = e^{t\boldsymbol{J}(\mathbf{0})}\boldsymbol{\theta}_0$. Then we have
$$\frac{1}{2}\frac{\mathrm{d}}{\mathrm{d}t}\|\boldsymbol{\theta}(t) - \hat{\boldsymbol{\theta}}(t)\|_2^2 \leq \left\langle \boldsymbol{g}(\boldsymbol{\theta}(t)) - \boldsymbol{J}(\mathbf{0})\hat{\boldsymbol{\theta}}(t), \boldsymbol{\theta}(t) - \hat{\boldsymbol{\theta}}(t)\right\rangle$$
$$= \left\langle \boldsymbol{g}(\boldsymbol{\theta}(t)) - \boldsymbol{J}(\mathbf{0})\boldsymbol{\theta}(t), \boldsymbol{\theta}(t) - \hat{\boldsymbol{\theta}}(t)\right\rangle + (\boldsymbol{\theta}(t) - \hat{\boldsymbol{\theta}}(t))^\top \boldsymbol{J}(\mathbf{0})(\boldsymbol{\theta}(t) - \hat{\boldsymbol{\theta}}(t))$$
$$\leq \|\boldsymbol{g}(\boldsymbol{\theta}(t)) - \boldsymbol{J}(\mathbf{0})\boldsymbol{\theta}(t)\|_2 \cdot \|\boldsymbol{\theta}(t) - \hat{\boldsymbol{\theta}}(t)\|_2 + \tilde{\mu}_1\|\boldsymbol{\theta}(t) - \hat{\boldsymbol{\theta}}(t)\|_2^2,$$
where the last inequality is due to Lemma G.2. By (22) and Lemma G.3, we have
$$\|\boldsymbol{g}(\boldsymbol{\theta}(t)) - \boldsymbol{J}(\mathbf{0})\boldsymbol{\theta}(t)\|_2 \leq \beta\|\boldsymbol{\theta}(t)\|_2^2 \leq \beta\left(\frac{1+\kappa r}{1+\kappa\alpha}\alpha\right)^2 \cdot e^{2\tilde{\mu}_1 t}.$$

So we have $\frac{\mathrm{d}}{\mathrm{d}t}\|\boldsymbol{\theta}(t) - \hat{\boldsymbol{\theta}}(t)\|_2 \leq \beta\left(\frac{1+\kappa r}{1+\kappa\alpha}\alpha\right)^2 \cdot e^{2\tilde{\mu}_1 t} + \tilde{\mu}_1\|\boldsymbol{\theta}(t) - \hat{\boldsymbol{\theta}}(t)\|_2$. By Grönwall's inequality,
$$\|\boldsymbol{\theta}(t) - \hat{\boldsymbol{\theta}}(t)\|_2 \leq \int_0^t \beta\left(\frac{1+\kappa r}{1+\kappa\alpha}\alpha\right)^2 \cdot e^{2\tilde{\mu}_1 \tau}e^{\tilde{\mu}_1(t-\tau)}d\tau.$$
Evaluating the integral gives
$$\|\boldsymbol{\theta}(t) - \hat{\boldsymbol{\theta}}(t)\|_2 \leq \beta\left(\frac{1+\kappa r}{1+\kappa\alpha}\alpha\right)^2 e^{\tilde{\mu}_1 t} \cdot \frac{e^{\tilde{\mu}_1 t} - 1}{\tilde{\mu}_1} \leq \kappa\left(\frac{1+\kappa r}{1+\kappa\alpha}\alpha \cdot e^{\tilde{\mu}_1 t}\right)^2 \leq \kappa r^2,$$
which proves the lemma. $\square$

**Lemma G.5.** *Let $\boldsymbol{\theta}(t) = \phi(\boldsymbol{\theta}_0, t), \hat{\boldsymbol{\theta}}(t) = \phi(\hat{\boldsymbol{\theta}}_0, t)$. If $\max\{\|\boldsymbol{\theta}_0\|_2, \|\hat{\boldsymbol{\theta}}_0\|_2\} \le \alpha$, then for $t \le T_\alpha(r)$,*

$$\|\boldsymbol{\theta}(t) - \hat{\boldsymbol{\theta}}(t)\|_2 \le e^{\tilde{\mu}_1 t + \kappa r}\|\boldsymbol{\theta}_0 - \hat{\boldsymbol{\theta}}_0\|_2.$$

*Proof.* For $t \le T_\alpha(r)$, by (20),

$$\frac{1}{2}\frac{\mathrm{d}}{\mathrm{d}t}\|\boldsymbol{\theta}(t) - \hat{\boldsymbol{\theta}}(t)\|_2^2 = \left\langle \boldsymbol{g}(\boldsymbol{\theta}(t)) - \boldsymbol{g}(\hat{\boldsymbol{\theta}}(t)), \boldsymbol{\theta}(t) - \hat{\boldsymbol{\theta}}(t) \right\rangle$$

$$= (\boldsymbol{\theta}(t) - \hat{\boldsymbol{\theta}}(t))^\top \left( \int_0^1 \boldsymbol{J}(\boldsymbol{\theta}_\xi(t))\mathrm{d}\xi \right)(\boldsymbol{\theta}(t) - \hat{\boldsymbol{\theta}}(t)),$$

where $\boldsymbol{\theta}_\xi(t) := \xi\boldsymbol{\theta}(t) + (1 - \xi)\hat{\boldsymbol{\theta}}(t)$. By Lemma G.3, $\max\{\|\boldsymbol{\theta}(t)\|_2, \|\hat{\boldsymbol{\theta}}(t)\|_2\} \le \frac{1+\kappa r}{1+\kappa\alpha}\alpha \cdot e^{\tilde{\mu}_1 t}$ for all $t \le T_\alpha(r)$. So $\|\boldsymbol{\theta}_\xi(t)\|_2 \le \frac{1+\kappa r}{1+\kappa\alpha}\alpha \cdot e^{\tilde{\mu}_1 t}$. Combining these with (19) and Lemma G.2, we have

$$\boldsymbol{h}^\top \boldsymbol{J}(\boldsymbol{\theta}_\xi(t))\boldsymbol{h} = \boldsymbol{h}^\top \boldsymbol{J}(\boldsymbol{0})\boldsymbol{h} + \boldsymbol{h}^\top (\boldsymbol{J}(\boldsymbol{\theta}_\xi(t)) - \boldsymbol{J}(\boldsymbol{0}))\boldsymbol{h} \le \left( \tilde{\mu}_1 + \beta \cdot \frac{1+\kappa r}{1+\kappa\alpha}\alpha \cdot e^{\tilde{\mu}_1 t} \right)\|\boldsymbol{h}\|_2^2,$$

for all $\boldsymbol{h} \in \mathbb{R}^d$. Thus, $\frac{\mathrm{d}}{\mathrm{d}t}\|\boldsymbol{\theta}(t) - \hat{\boldsymbol{\theta}}(t)\|_2 \le \left( \tilde{\mu}_1 + \beta \cdot \frac{1+\kappa r}{1+\kappa\alpha}\alpha \cdot e^{\tilde{\mu}_1 t} \right)\|\boldsymbol{\theta}(t) - \hat{\boldsymbol{\theta}}(t)\|_2$. This implies

$$\log \frac{\|\boldsymbol{\theta}(t) - \hat{\boldsymbol{\theta}}(t)\|_2}{\|\boldsymbol{\theta}(0) - \hat{\boldsymbol{\theta}}(0)\|_2} \le \int_0^t \left( \tilde{\mu}_1 + \beta \cdot \frac{1+\kappa r}{1+\kappa\alpha}\alpha \cdot e^{\tilde{\mu}_1 \tau} \right)d\tau$$

$$\le \tilde{\mu}_1 t + \kappa \cdot \frac{1+\kappa r}{1+\kappa\alpha}\alpha e^{\tilde{\mu}_1 t}$$

$$\le \tilde{\mu}_1 t + \kappa r.$$

Therefore, $\|\boldsymbol{\theta}(t) - \hat{\boldsymbol{\theta}}(t)\|_2 \le e^{\tilde{\mu}_1 t + \kappa r}\|\boldsymbol{\theta}(0) - \hat{\boldsymbol{\theta}}(0)\|_2$. $\qquad\square$

**Lemma G.6.** *For every $t \in (-\infty, +\infty)$, $\boldsymbol{z}(t)$ exists and $\boldsymbol{z}_\alpha(t)$ converges to $\boldsymbol{z}(t)$ in the following rate:*

$$\|\boldsymbol{z}_\alpha(t) - \boldsymbol{z}(t)\|_2 = O(\alpha),$$

*where $O$ hides constants depending on $g(\boldsymbol{\theta})$ and $t$.*

*Proof.* We prove the lemma in the cases of $t \in (-\infty, F(R)/\tilde{\mu}_1]$ and $t > F(R)/\tilde{\mu}_1$ respectively.

**Case 1.** Fix $t \in (-\infty, F(R)/\tilde{\mu}_1]$. Let $\tilde{\alpha}$ be the unique number such that $\frac{\tilde{\alpha}}{1+\kappa\tilde{\alpha}} = \alpha$ (i.e., $F(\tilde{\alpha}) = \log\alpha$). Let $\alpha'$ be an arbitrary number less than $\alpha$. Let $t_0 := \frac{1}{\tilde{\mu}_1}\log\frac{\alpha}{\alpha'}$. Then $t_0 = \frac{1}{\tilde{\mu}_1}(F(\tilde{\alpha}) - \log\alpha') \le T_{\alpha'}(\tilde{\alpha})$. By Lemma G.4, we have

$$\|\phi(\alpha'\boldsymbol{e}_1, t_0) - \alpha\boldsymbol{e}_1\|_2 = \left\| \phi(\alpha'\boldsymbol{e}_1, t_0) - e^{t_0 \boldsymbol{J}(\boldsymbol{0})}\alpha'\boldsymbol{e}_1 \right\|_2 = O(\tilde{\alpha}^2).$$

Let $r := F^{-1}(\tilde{\mu}_1 t) \le R$. Then $t + \frac{1}{\tilde{\mu}_1}\log\frac{1}{\alpha} = T_{\tilde{\alpha}}(r)$ if $\tilde{\alpha} < r$.

By Lemma G.3, $\|\phi(\alpha'\boldsymbol{e}_1, t_0)\|_2 \le \tilde{\alpha}$. Also, $\|\alpha\boldsymbol{e}_1\|_2 = \frac{\tilde{\alpha}}{1+\kappa\tilde{\alpha}} \le \tilde{\alpha}$. By Lemma G.5,

$$\|\boldsymbol{z}_\alpha(t) - \boldsymbol{z}_{\alpha'}(t)\|_2 = \left\| \phi\left( \alpha'\boldsymbol{e}_1, t + \frac{1}{\tilde{\mu}_1}\log\frac{1}{\alpha'} \right) - \phi\left( \alpha\boldsymbol{e}_1, t + \frac{1}{\tilde{\mu}_1}\log\frac{1}{\alpha} \right) \right\|_2$$

$$= \left\| \phi\left( \phi(\alpha'\boldsymbol{e}_1, t_0), t + \frac{1}{\tilde{\mu}_1}\log\frac{1}{\alpha} \right) - \phi\left( \alpha\boldsymbol{e}_1, t + \frac{1}{\tilde{\mu}_1}\log\frac{1}{\alpha} \right) \right\|_2$$

$$\le O(\tilde{\alpha}^2 \cdot e^{\tilde{\mu}_1(t + \frac{1}{\tilde{\mu}_1}\log\frac{1}{\alpha}) + \kappa r})$$

$$\le O\left( \frac{\tilde{\alpha}^2}{\alpha} \right).$$

For $\alpha$ small enough, we have $\tilde{\alpha} = O(\alpha)$, so for any $\alpha' \in (0, \alpha)$,

$$\|\boldsymbol{z}_\alpha(t) - \boldsymbol{z}_{\alpha'}(t)\|_2 = O(\alpha).$$

This implies that $\{\boldsymbol{z}_\alpha(t)\}$ satisfies Cauchy's criterion for every $t$, and thus the limit $\boldsymbol{z}(t)$ exists for $t \le F(R)/\tilde{\mu}_1$. The convergence rate can be deduced by taking limits for $\alpha' \to 0$ on both sides.

**Case 2.** For $t = F(R)/\tilde{\mu}_1 + \tau$ with $\tau > 0$, $\phi(\boldsymbol{\theta}, \tau)$ is locally Lipschitz with respect to $\boldsymbol{\theta}$. So

$$\|\boldsymbol{z}_\alpha(t) - \boldsymbol{z}_{\alpha'}(t)\|_2 = \|\phi(\boldsymbol{z}_\alpha(F(R)/\tilde{\mu}_1), \tau) - \phi(\boldsymbol{z}_{\alpha'}(F(R)/\tilde{\mu}_1), \tau)\|_2$$
$$= O(\|\boldsymbol{z}_\alpha(F(R)/\tilde{\mu}_1) - \boldsymbol{z}_{\alpha'}(F(R)/\tilde{\mu}_1)\|_2)$$
$$= O(\alpha),$$

which proves the lemma for $t > F(R)/\tilde{\mu}_1$. $\qquad\square$

*Proof for Theorem G.1.* The existence of $\boldsymbol{z}(t) := \lim_{\alpha \to 0} \boldsymbol{z}_\alpha(t) = \lim_{\alpha \to 0} \phi\left(\alpha\boldsymbol{e}_1, t + \frac{1}{\tilde{\mu}_1}\log\frac{1}{\alpha}\right)$ has already been proved in Lemma G.6, where we show $\|\boldsymbol{z}_\alpha(t) - \boldsymbol{z}(t)\|_2 = O(\alpha)$.

By the continuity of $\phi(\,\cdot\,, t)$ for every $t \in \mathbb{R}$, we have

$$\boldsymbol{z}(t) = \lim_{\alpha \to 0} \phi\left(\alpha\tilde{\boldsymbol{v}}_1, t + \frac{1}{\tilde{\mu}_1}\log\frac{1}{\alpha}\right) = \phi\left(\lim_{\alpha \to 0}\phi\left(\alpha\tilde{\boldsymbol{v}}_1, \frac{1}{\tilde{\mu}_1}\log\frac{1}{\alpha}\right), t\right) = \phi\left(\boldsymbol{z}(0), t\right).$$

Now it is only left to prove (7). WLOG we can assume that $\|\boldsymbol{\delta}_\alpha\|_2$ is decreasing and $\frac{\alpha}{2} \le \|\boldsymbol{\delta}_\alpha\|_2 \le \alpha$ (otherwise we can do reparameterization). Then our goal becomes proving

$$\|\boldsymbol{\theta}_\alpha(t) - \boldsymbol{z}(t)\|_2 = O\left(\alpha^{\frac{\tilde{\gamma}}{\tilde{\mu}_1 + \tilde{\gamma}}}\right). \tag{23}$$

where $\boldsymbol{\theta}_\alpha(t) := \phi\left(\boldsymbol{\delta}_\alpha, t + \frac{1}{\tilde{\mu}_1}\log\frac{1}{\langle\boldsymbol{\delta}_\alpha, \boldsymbol{e}_1\rangle}\right)$. We prove (23) in the cases of $t \in (-\infty, F(R)/\tilde{\mu}_1]$ and $t > F(R)/\tilde{\mu}_1$ respectively.

**Case 1.** Fix $t \in (-\infty, (F(R) + \log q)/\tilde{\mu}_1]$. Let $\tilde{\alpha}_1 = \alpha^{\frac{\tilde{\gamma}}{\tilde{\mu}_1 + \tilde{\gamma}}}$. Let $\alpha_1 := e^{F(\tilde{\alpha}_1)} = \frac{\tilde{\alpha}_1}{1 + \kappa\tilde{\alpha}_1}$. Let $t_0 := \frac{1}{\tilde{\mu}_1}(F(\tilde{\alpha}_1) - \log\alpha) \le T_{\|\boldsymbol{\delta}_\alpha\|_2}(\tilde{\alpha}_1)$. At time $t_0$, by Lemma G.2 we have

$$\left\|e^{t_0 \boldsymbol{J}(\boldsymbol{0})} - e^{\tilde{\mu}_1 t_0}\boldsymbol{e}_1\boldsymbol{e}_1^\top\right\|_2 = e^{\tilde{\mu}_2 t_0} = e^{\frac{\tilde{\mu}_2}{\tilde{\mu}_1}(F(\tilde{\alpha}_1) - \log\alpha)} = \left(\frac{\alpha_1}{\alpha}\right)^{\frac{\tilde{\mu}_2}{\tilde{\mu}_1}}. \tag{24}$$

Let $q_\alpha := \left\langle\frac{\boldsymbol{\delta}_\alpha}{\alpha}, \boldsymbol{e}_1\right\rangle$. By Definition 5.2, there exists $q > 0$ such that $q_\alpha \ge q$ for all sufficiently small $\alpha$. Then we have

$$\|\phi(\boldsymbol{\delta}_\alpha, t_0) - \alpha_1 q_\alpha \boldsymbol{e}_1\|_2 = \left\|\phi(\boldsymbol{\delta}_\alpha, t_0) - e^{t_0 \boldsymbol{J}(\boldsymbol{0})}\boldsymbol{\delta}_\alpha\right\|_2 + \left\|\left(e^{t_0 \boldsymbol{J}(\boldsymbol{0})} - e^{\tilde{\mu}_1 t_0}\boldsymbol{e}_1\boldsymbol{e}_1^\top\right)\boldsymbol{\delta}_\alpha\right\|_2$$
$$= O(\tilde{\alpha}_1^2) + \left(\frac{\alpha_1}{\alpha}\right)^{\frac{\tilde{\mu}_2}{\tilde{\mu}_1}}\|\boldsymbol{\delta}_\alpha\|_2$$
$$= O(\tilde{\alpha}_1^2 + \alpha_1^{\tilde{\mu}_2/\tilde{\mu}_1}\alpha^{1 - \tilde{\mu}_2/\tilde{\mu}_1})$$
$$= O(\tilde{\alpha}_1^2).$$

Let $r := F^{-1}(\tilde{\mu}_1 t + \log\frac{1}{q_\alpha}) \le R$. Then $t + \frac{1}{\tilde{\mu}_1}\log\frac{1}{\alpha_1 q_\alpha} = T_{\tilde{\alpha}}(r)$ if $\tilde{\alpha} < r$. By Lemma G.3, $\|\phi(\boldsymbol{\delta}_\alpha, t_0)\|_2 \le \tilde{\alpha}_1$. Also, $\|\alpha_1 q_\alpha \boldsymbol{e}_1\|_2 \le \alpha_1 = \frac{\tilde{\alpha}_1}{1 + \kappa\tilde{\alpha}_1} \le \tilde{\alpha}_1$. By Lemma G.5,

$$\|\boldsymbol{\theta}_\alpha(t) - \boldsymbol{z}_{\alpha_1}(t)\|_2 \le \left\|\phi\left(\phi(\boldsymbol{\delta}_\alpha, t_0), t + \frac{1}{\tilde{\mu}_1}\log\frac{1}{\alpha_1 q_\alpha}\right) - \phi\left(\alpha_1 q_\alpha \boldsymbol{e}_1, t + \frac{1}{\tilde{\mu}_1}\log\frac{1}{\alpha_1 q_\alpha}\right)\right\|_2$$
$$= O\left(\alpha_1^2 \cdot e^{\tilde{\mu}_1\left(t + \frac{1}{\tilde{\mu}_1}\log\frac{1}{\alpha_1 q_\alpha}\right) + \kappa r}\right)$$
$$= O(\alpha_1).$$

Combining this with the convergence rate for $\boldsymbol{z}_{\alpha_1}(t)$, we have

$$\|\boldsymbol{\theta}_\alpha(t) - \boldsymbol{z}(t)\|_2 \le \|\boldsymbol{\theta}_\alpha(t) - \boldsymbol{z}_{\alpha_1}(t)\|_2 + \|\boldsymbol{z}_{\alpha_1}(t) - \boldsymbol{z}(t)\|_2 = O(\alpha_1).$$

**Case 2.** For $t = (F(R) + \log q)/\tilde{\mu}_1 + \tau$ with $\tau > 0$, $\phi(\boldsymbol{\theta}, \tau)$ is locally Lipschitz with respect to $\boldsymbol{\theta}$. So

$$\|\boldsymbol{\theta}_\alpha(t) - \boldsymbol{z}(t)\|_2 = \|\phi(\boldsymbol{\theta}_\alpha((F(R) + \log q)/\tilde{\mu}_1), \tau) - \phi(\boldsymbol{z}((F(R) + \log q)/\tilde{\mu}_1), \tau)\|_2$$
$$= O(\|\boldsymbol{\theta}_\alpha((F(R) + \log q)/\tilde{\mu}_1) - \boldsymbol{z}((F(R) + \log q)/\tilde{\mu}_1)\|_2)$$
$$= O(\alpha_1),$$

which proves (23) for $t > (F(R) + \log q)/\tilde{\mu}_1$. $\qquad\square$

## G.3 EXTENSION TO NON-DIAGONALIZABLE CASE

The proof in Appendix G.2 can be generalized to the case where $J(0)$. Now we state the theorem formally and sketch the proof idea. We use the notations $g(\theta), \phi(\theta_0, t), J(\theta)$ as in Section 5.1, but we do not assume that $J(0)$ is diagonalizable. Instead, we use $\tilde{\mu}_1, \tilde{\mu}_2, \ldots, \tilde{\mu}_d \in \mathbb{C}$ to denote the eigenvalues of $J(0)$, repeated according to algebraic multiplicity. We sort the eigenvalues in the descending order of the real part of each eigenvalue, i.e., $\Re(\tilde{\mu}_1) \geq \Re(\tilde{\mu}_2) \geq \cdots \geq \Re(\tilde{\mu}_d)$, where $\Re(z)$ stands for the real part of a complex number $z \in \mathbb{C}$. We call the eigenvalue with the largest real part the top eigenvalue.

**Theorem G.7.** *Assume that $\theta = 0$ is a critical point and the following regularity conditions hold:*

1. *$g(\theta)$ is $\mathcal{C}^2$-smooth;*

2. *$\phi(\theta_0, t)$ exists for all $\theta_0$ and $t$;*

3. *The top eigenvalue of $J(0)$ is unique and is a positive real number, i.e.,*

$$\tilde{\mu}_1 > \max\{\Re(\tilde{\mu}_2), 0\}.$$

*Let $\tilde{v}_1, \tilde{u}_1$ be the left and right eigenvectors associated with $\tilde{\mu}_1$, satisfying $\tilde{u}_1^\top \tilde{v}_1 = 1$. Let $z_\alpha(t) := \phi(\alpha \tilde{v}_1, t + \frac{1}{\tilde{\mu}_1} \log \frac{1}{\alpha})$ for every $\alpha > 0$, then $\forall t \in \mathbb{R}$, $z(t) := \lim_{\alpha \to 0} z_\alpha(t)$ exists and $z(t) = \phi(z(0), t)$. If $\delta_\alpha$ converges to $0$ with positive alignment with $\tilde{u}_1$ as $\alpha \to 0$, then for any $t \in \mathbb{R}$ and for any $\epsilon > 0$, there is a constant $C > 0$ such that for every sufficiently small $\alpha$,*

$$\left\| \phi\left(\delta_\alpha, t + \frac{1}{\tilde{\mu}_1} \log \frac{1}{\langle \delta_\alpha, \tilde{u}_1 \rangle}\right) - z(t) \right\|_2 \leq C \cdot \|\delta_\alpha\|_2^{\frac{\tilde{\gamma}}{\tilde{\mu}_1 + \tilde{\gamma}} - \epsilon}, \tag{25}$$

*where $\tilde{\gamma} := \tilde{\mu}_1 - \Re(\tilde{\mu}_2)$ is the eigenvalue gap.*

*Proof Sketch.* Define the following two types of matrices. For $r \geq 1, a, \delta \in \mathbb{R}$, we define

$$J_{a,\delta}^{(r)} := \begin{bmatrix} a & \delta & & & & \\ & a & \delta & & & \\ & & a & \delta & & \\ & & & \ddots & \ddots & \\ & & & & a & \delta \\ & & & & & a \end{bmatrix} \in \mathbb{R}^{r \times r}.$$

For $r \geq 1, a, b, \delta \in \mathbb{R}$, we define

$$J_{a,b,\delta}^{(r)} := \begin{bmatrix} C & \delta I & & & \\ & C & \delta I & & \\ & & C & \delta I & \\ & & & \ddots & \ddots \\ & & & & C & \delta I \\ & & & & & C \end{bmatrix} \in \mathbb{R}^{2r \times 2r},$$

where $C = \begin{bmatrix} a & -b \\ b & a \end{bmatrix} \in \mathbb{R}^{2 \times 2}$.

By linear algebra, the real matrix $J(0)$ can be written in the real Jordan normal form, i.e., $J(0) = \tilde{V} \mathrm{diag}(J_{[1]}, \ldots, J_{[m]}) \tilde{V}^{-1}$, where $\tilde{V} \in \mathbb{R}^{d \times d}$ is an invertible matrix, and each $J_{[j]}$ is a real Jordan block. Recall that there are two types of real Jordan blocks, $J_{a,1}^{(r)}$ or $J_{a,b,1}^{(r)}$. The former one is associated with a real eigenvalue $a$, and the latter one is associated with a pair of complex eigenvalues $a \pm bi$. The sum of sizes of all Jordan blocks corresponding to a real eigenvalue $a$ is its algebraic multiplicity. The sum of sizes of all Jordan blocks corresponding to a pair of complex eigenvalues $a \pm bi$ is two times the algebraic multiplicity of $a + bi$ or $a - bi$ (note that $a \pm bi$ have the same multiplicity).

It is easy to see that $J_{a,\delta}^{(r)} = D J_{a,1}^{(r)} D^{-1}$ for $D = \mathrm{diag}(\delta^r, \delta^{r-1}, \ldots, \delta) \in \mathbb{R}^{r \times r}$ and $J_{a,b,\delta}^{(r)} = D J_{a,b,1}^{(r)} D^{-1}$ for $D = \mathrm{diag}(\delta^r, \delta^r, \delta^{r-1}, \delta^{r-1}, \ldots, \delta, \delta) \in \mathbb{R}^{2r \times 2r}$. This means for every $\delta > 0$

there exists $\tilde{\boldsymbol{V}}_\delta$ such that $\boldsymbol{J}(\boldsymbol{0}) = \tilde{\boldsymbol{V}}_\delta \boldsymbol{J}_\delta \tilde{\boldsymbol{V}}_\delta^{-1}$, where $\boldsymbol{J}_\delta := \mathrm{diag}(\boldsymbol{J}_{\delta[1]}, \ldots, \boldsymbol{J}_{\delta[m]})$, $\boldsymbol{J}_{\delta[j]} := \boldsymbol{J}_{a,\delta}^{(r)}$ if $\boldsymbol{J}_{[j]} := \boldsymbol{J}_{a,1}^{(r)}$, or $\boldsymbol{J}_{\delta[j]} := \boldsymbol{J}_{a,b,\delta}^{(r)}$ if $\boldsymbol{J}_{[j]} := \boldsymbol{J}_{a,b,1}^{(r)}$. Since the top eigenvalue of $\boldsymbol{J}(\boldsymbol{0})$ is positive and unique, $\tilde{\mu}_1$ corresponds to only one block $[\tilde{\mu}_1] \in \mathbb{R}^{1\times 1}$. WLOG we let $\boldsymbol{J}_1 = [\tilde{\mu}_1]$, and thus $\boldsymbol{J}_{\delta[1]} = [\tilde{\mu}_1]$.

We only need to select a parameter $\delta > 0$ and prove the theorem in the case of $\boldsymbol{J}(\boldsymbol{0}) = \boldsymbol{J}_\delta$ since we can change the basis in a similar way as we have done in Appendix G.1. By scrutinizing the proof for Theorem G.1, we can find that we only need to reprove Lemma G.2. However, Lemma G.2 may not be correct since $\boldsymbol{J}(\boldsymbol{0})$ is not diagonal anymore. Instead, we prove the following:

1. If $\delta \in (0, \tilde{\gamma})$, then $\boldsymbol{h}^\top \boldsymbol{J}_\delta \boldsymbol{h} \le \tilde{\mu}_1 \|\boldsymbol{h}\|_2^2$ for all $\boldsymbol{h} \in \mathbb{R}^d$;

2. For any $\tilde{\mu}_2' \in (\Re(\tilde{\mu}_2), \tilde{\mu}_1)$, if $\delta \in (0, \tilde{\mu}_2' - \Re(\tilde{\mu}_2))$, then $\left\| e^{t\boldsymbol{J}_\delta} - e^{\tilde{\mu}_1 t} \boldsymbol{e}_1 \boldsymbol{e}_1^\top \right\|_2 \le e^{\tilde{\mu}_2' t}$ for all $t \ge 0$.

**Proof for Item 1.** Let $\mathcal{K}$ be the set of pairs $(k_1, k_2)$ such that $k_1 \ne k_2$ and the entry of $\boldsymbol{J}_\delta$ at the $k_1$-th row and the $k_2$-th column is non-zero. Then we have

$$\boldsymbol{h}^\top \boldsymbol{J}_\delta \boldsymbol{h} = \boldsymbol{h}^\top \frac{\boldsymbol{J}_\delta + \boldsymbol{J}_\delta^\top}{2} \boldsymbol{h} = \sum_{k=1}^d \Re(\tilde{\mu}_k) h_k^2 + \sum_{(k_1,k_2)\in\mathcal{K}} h_{k_1} h_{k_2} \delta$$
$$\le \sum_{k=1}^d \Re(\tilde{\mu}_k) h_k^2 + \sum_{(k_1,k_2)\in\mathcal{K}} \frac{h_{k_1}^2 + h_{k_2}^2}{2} \delta.$$

Note that $\Re(\tilde{\mu}_k) \le \Re(\tilde{\mu}_2)$ for $k \ge 2$. Also note that there is no pair in $\mathcal{K}$ has $k_1 = 1$ or $k_2 = 1$, and for every $k \ge 2$ there are at most two pairs in $\mathcal{K}$ has $k_1 = k$ or $k_2 = k$. Combining all these together gives

$$\boldsymbol{h}^\top \boldsymbol{J}_\delta \boldsymbol{h} \le \tilde{\mu}_1 h_1^2 + (\Re(\tilde{\mu}_2) + \delta) \sum_{k=2}^d h_k^2 \le \tilde{\mu}_1 \|\boldsymbol{h}\|_2^2,$$

which proves Item 1.

**Proof for Item 2.** Since $\boldsymbol{J}_\delta$ is block diagonal, we only need to prove that $\|e^{t\boldsymbol{J}_{\delta[j]}}\|_2 \le e^{\tilde{\mu}_2' t}$ for every $j \ge 2$. If $\boldsymbol{J}_{\delta[j]} = \boldsymbol{J}_{a,\delta}^{(r)} = a\boldsymbol{I} + \delta\boldsymbol{N}$, where $\boldsymbol{N}$ is the nilpotent matrix, then

$$e^{t\boldsymbol{J}_{\delta[j]}} = e^{at\boldsymbol{I} + \delta t\boldsymbol{N}} = e^{at\boldsymbol{I}} e^{\delta t\boldsymbol{N}} = e^{at} e^{\delta t\boldsymbol{N}},$$

where the second equality uses the fact that $\boldsymbol{I}$ and $\boldsymbol{N}$ are commutable. So we have

$$\|e^{t\boldsymbol{J}_{\delta[j]}}\|_2 \le e^{at} \|e^{\delta t\boldsymbol{N}}\|_2 = e^{at} e^{\delta t\|\boldsymbol{N}\|_2} \le e^{(a+\delta)t}.$$

If $\boldsymbol{J}_{\delta[j]} = \boldsymbol{J}_{a,\delta}^{(r)} = \boldsymbol{D} + \delta\boldsymbol{N}^2$, where $\boldsymbol{D} = \mathrm{diag}(\boldsymbol{C}, \boldsymbol{C}, \ldots, \boldsymbol{C})$ and $\boldsymbol{N}$ is the nilpotent matrix, then

$$e^{t\boldsymbol{J}_{\delta[j]}} = e^{t\boldsymbol{D} + \delta t\boldsymbol{N}^2} = e^{t\boldsymbol{D}} e^{\delta t\boldsymbol{N}^2},$$

where the second equality uses the fact that $\boldsymbol{D}$ and $\boldsymbol{N}^2$ are commutable. Note that $e^{t\boldsymbol{C}} = e^{at}\begin{bmatrix} \cos(bt) & -\sin(bt) \\ \sin(bt) & \cos(bt) \end{bmatrix}$, which implies $\|e^{t\boldsymbol{D}}\|_2 = \|e^{t\boldsymbol{C}}\|_2 = e^{at}$. So we have

$$\|e^{t\boldsymbol{J}_{\delta[j]}}\|_2 \le \|e^{t\boldsymbol{D}}\|_2 \cdot \|e^{\delta t\boldsymbol{N}^2}\|_2 = e^{at} e^{\delta t\|\boldsymbol{N}^2\|_2} \le e^{(a+\delta)t}.$$

Since $\delta \in (0, \tilde{\mu}_2' - \Re(\tilde{\mu}_2))$, we know that $a + \delta < \tilde{\mu}_2'$, which completes the proof.

**Proof for a fixed $\delta$.** Since Item 1 continues to hold for $\delta \in (0, \tilde{\gamma})$, Lemmas G.3 to G.6 also hold. This proves that $\boldsymbol{z}(t)$ exists and satisfies (6).

It remains to prove (25) for any $\epsilon > 0$. Let $\tilde{\gamma}' \in (0, \tilde{\gamma})$ be a number such that $\frac{\tilde{\gamma}'}{\tilde{\mu}_1 + \tilde{\gamma}'} \ge \frac{\tilde{\gamma}}{\tilde{\mu}_1 + \tilde{\gamma}} - \epsilon$. Fix $\tilde{\mu}_2' = \tilde{\mu}_1 - \tilde{\gamma}'$, $\delta = \tilde{\mu}_2' - \Re(\tilde{\mu}_2)$. By Item 2, we have $\left\| e^{t\boldsymbol{J}_\delta} - e^{\tilde{\mu}_1 t} \boldsymbol{e}_1 \boldsymbol{e}_1^\top \right\|_2 \le e^{\tilde{\mu}_2' t}$ for all $t \ge 0$. By scrutinizing the proof for Theorem G.1, we can find that the only place we use Item 2 in Lemma G.2

is in (24). For proving (25), we can repeat the proof while replacing all the occurrences of $\tilde{\mu}_2$ by $\tilde{\mu}_2'$. Then we know that for every $t \in \mathbb{R}$, there is a constant $C > 0$ such that

$$\left\| \tilde{V}_\delta^{-1} \phi \left( \boldsymbol{\delta}_\alpha, t + \tfrac{1}{\tilde{\mu}_1} \log \tfrac{1}{\langle \boldsymbol{\delta}_\alpha, \tilde{u}_1 \rangle} \right) - \tilde{V}_\delta^{-1} z(t) \right\|_2 \le C \cdot \| \tilde{V}_\delta^{-1} \boldsymbol{\delta}_\alpha \|_2^{\frac{\tilde{\gamma}'}{\tilde{\mu}_1 + \tilde{\gamma}'}},$$

for every sufficiently small $\alpha$. By definition of $\tilde{\gamma}'$, $\frac{\tilde{\gamma}'}{\tilde{\mu}_1 + \tilde{\gamma}'} \ge \frac{\tilde{\gamma}}{\tilde{\mu}_1 + \tilde{\gamma}} - \epsilon$. Since $\boldsymbol{\delta}_\alpha \to \mathbf{0}$ as $\alpha \to 0$, we have $\| \tilde{V}_\delta^{-1} \boldsymbol{\delta}_\alpha \|_2 < 1$ for sufficiently small $\alpha$. Then we have

$$\left\| \phi \left( \boldsymbol{\delta}_\alpha, t + \tfrac{1}{\tilde{\mu}_1} \log \tfrac{1}{\langle \boldsymbol{\delta}_\alpha, \tilde{u}_1 \rangle} \right) - z(t) \right\|_2 \le \| \tilde{V}_\delta \|_2 \cdot \left\| \tilde{V}_\delta^{-1} \phi \left( \boldsymbol{\delta}_\alpha, t + \tfrac{1}{\tilde{\mu}_1} \log \tfrac{1}{\langle \boldsymbol{\delta}_\alpha, \tilde{u}_1 \rangle} \right) - \tilde{V}_\delta^{-1} z(t) \right\|_2$$

$$\le \| \tilde{V}_\delta \|_2 \cdot C \cdot \| \tilde{V}_\delta^{-1} \boldsymbol{\delta}_\alpha \|_2^{\frac{\tilde{\gamma}'}{\tilde{\mu}_1 + \tilde{\gamma}'}}$$

$$\le C \cdot \| \tilde{V}_\delta \|_2 \cdot \| \tilde{V}_\delta^{-1} \|_2^{\frac{\tilde{\gamma}'}{\tilde{\mu}_1 + \tilde{\gamma}'}} \cdot \| \boldsymbol{\delta}_\alpha \|_2^{\frac{\tilde{\gamma}}{\tilde{\mu}_1 + \tilde{\gamma}} - \epsilon}.$$

Absorbing $\| \tilde{V}_\delta \|_2 \cdot \| \tilde{V}_\delta^{-1} \|_2^{\frac{\tilde{\gamma}'}{\tilde{\mu}_1 + \tilde{\gamma}'}}$ into $C$ proves (25). $\qquad\qquad\square$

## H  EIGENVALUES OF JACOBIANS AND HESSIANS

In this section we analyze the eigenvalues of the Jacobian $\boldsymbol{J}(\boldsymbol{W})$ at critical points of (2).

For notation simplicity, we write $\mathrm{sz}(\boldsymbol{A}) := \boldsymbol{A} + \boldsymbol{A}^\top$ to denote the symmetric matrix produced by adding up $\boldsymbol{A}$ and its transpose, and write $\mathrm{ac}\{\boldsymbol{A}, \boldsymbol{B}\} = \boldsymbol{A}\boldsymbol{B} + \boldsymbol{B}\boldsymbol{A}$ to denote the anticommutator of two matrices $\boldsymbol{A}, \boldsymbol{B}$. Then $\boldsymbol{g}(\boldsymbol{W})$ can be written as $\boldsymbol{g}(\boldsymbol{W}) := -\mathrm{ac}\{\nabla f(\boldsymbol{W}), \boldsymbol{W}\}$.

Let $\boldsymbol{U}_0 \in \mathbb{R}^{d \times r}$ be a stationary point of the function $\mathcal{L} : \mathbb{R}^{d \times r} \to \mathbb{R}, \mathcal{L}(\boldsymbol{U}) = \frac{1}{2} f(\boldsymbol{U}\boldsymbol{U}^\top)$, i.e., $\nabla \mathcal{L}(\boldsymbol{U}_0) = \nabla f(\boldsymbol{U}_0 \boldsymbol{U}_0^\top) \boldsymbol{U}_0 = \mathbf{0}$. By Lemma C.1, this implies

$$\nabla f(\boldsymbol{W}_0) \boldsymbol{W}_0 = \mathbf{0} \tag{26}$$

for $\boldsymbol{W}_0 := \boldsymbol{U}_0 \boldsymbol{U}_0^\top$, and thus $\boldsymbol{W}_0$ is a critical point of (2).

For a real-valued or vector-valued function $F(\boldsymbol{\theta})$, we use $DF(\boldsymbol{\theta})[\boldsymbol{\delta}], D^2 F(\boldsymbol{\theta})[\boldsymbol{\delta}_1, \boldsymbol{\delta}_2]$ to denote the first- and second-order directional derivatives of $F(\,\cdot\,)$ at $\boldsymbol{\theta}$.

Let $\mathcal{X}$ be a linear space, which can be $\mathbb{R}^{d \times d}$ or $\mathbb{R}^{d \times r}$. For a function $F : \mathcal{X} \to \mathcal{X}$, we use $DF(\boldsymbol{\theta})$ to denote the directional derivative of $F$ at $\boldsymbol{\theta}$, represented by the linear operator

$$DF(\boldsymbol{\theta})[\boldsymbol{\Delta}] : \mathcal{X} \to \mathcal{X}, \boldsymbol{\Delta} \mapsto DF(\boldsymbol{\theta})[\boldsymbol{\Delta}] = \lim_{t \to 0} \frac{F(\boldsymbol{\theta} + t\boldsymbol{\Delta}) - F(\boldsymbol{\theta})}{t}.$$

We also write $DF(\boldsymbol{\theta})[\boldsymbol{\Delta}_1, \boldsymbol{\Delta}_2] := \langle DF(\boldsymbol{\theta})[\boldsymbol{\Delta}_1], \boldsymbol{\Delta}_2 \rangle$.

For a function $F : \mathcal{X} \to \mathbb{R}$, we use $D^2 F(\boldsymbol{\theta}) = D(\nabla F(\boldsymbol{\theta}))$ to denote the second directional derivative of $F$ at $\boldsymbol{\theta}$, i.e., $D^2 F(\boldsymbol{\theta})[\boldsymbol{\Delta}] = D(\nabla F(\boldsymbol{\theta}))[\boldsymbol{\Delta}], D^2 F(\boldsymbol{\theta})[\boldsymbol{\Delta}_1, \boldsymbol{\Delta}_2] = D(\nabla F(\boldsymbol{\theta}))[\boldsymbol{\Delta}_1, \boldsymbol{\Delta}_2]$.

Define $\boldsymbol{J}(\boldsymbol{W}) := D\boldsymbol{g}(\boldsymbol{W})$. By simple calculus, we can compute the formula for $\boldsymbol{J}(\boldsymbol{W}_0)$:

$$\boldsymbol{J}(\boldsymbol{W}_0)[\boldsymbol{\Delta}] = -\mathrm{ac}\{\nabla f(\boldsymbol{W}_0), \boldsymbol{\Delta}\} - \mathrm{ac}\{D^2 f(\boldsymbol{W}_0)[\boldsymbol{\Delta}], \boldsymbol{W}_0\},$$

$$\boldsymbol{J}(\boldsymbol{W}_0)[\boldsymbol{\Delta}_1, \boldsymbol{\Delta}_2] = -\left\langle \nabla f(\boldsymbol{W}_0), \mathrm{sz}(\boldsymbol{\Delta}_1 \boldsymbol{\Delta}_2^\top) \right\rangle - D^2 f(\boldsymbol{W}_0)[\boldsymbol{\Delta}_1, \mathrm{sz}(\boldsymbol{W}_0 \boldsymbol{\Delta}_2^\top)],$$

where $\boldsymbol{\Delta}, \boldsymbol{\Delta}_1, \boldsymbol{\Delta}_2 \in \mathbb{R}^{d \times d}$.

We can also compute the formula for $D^2 \mathcal{L}(\boldsymbol{U}_0)$:

$$D^2 \mathcal{L}(\boldsymbol{U}_0)[\boldsymbol{\Delta}] = \nabla f(\boldsymbol{W}_0)\boldsymbol{\Delta} + D^2 f(\boldsymbol{W}_0)[\mathrm{sz}(\boldsymbol{\Delta}\boldsymbol{U}_0^\top)]\boldsymbol{U}_0,$$

$$D^2 \mathcal{L}(\boldsymbol{U}_0)[\boldsymbol{\Delta}_1, \boldsymbol{\Delta}_2] = \frac{1}{2} \left( \left\langle \nabla f(\boldsymbol{W}_0), \mathrm{sz}(\boldsymbol{\Delta}_1 \boldsymbol{\Delta}_2^\top) \right\rangle + D^2 f(\boldsymbol{W}_0)[\mathrm{sz}(\boldsymbol{\Delta}_1 \boldsymbol{U}_0^\top), \mathrm{sz}(\boldsymbol{\Delta}_2 \boldsymbol{U}_0^\top)] \right),$$

where $\boldsymbol{\Delta}, \boldsymbol{\Delta}_1, \boldsymbol{\Delta}_2 \in \mathbb{R}^{d \times r}$.

## H.1 EIGENVALUES AT THE ORIGIN

The eigenvalues of $\boldsymbol{J}(\boldsymbol{0})$ is given in Lemma 5.4. Now we provide the proof.

*Proof for Lemma 5.4.* For $\boldsymbol{W}_0 = \boldsymbol{0}$, we have

$$\boldsymbol{J}(\boldsymbol{0})[\boldsymbol{\Delta}] = -\nabla f(\boldsymbol{0})\boldsymbol{\Delta} - \boldsymbol{\Delta}\nabla f(\boldsymbol{0})$$
$$\boldsymbol{J}(\boldsymbol{0})[\boldsymbol{\Delta}_1, \boldsymbol{\Delta}_2] = -\left\langle \nabla f(\boldsymbol{0}), \mathrm{sz}(\boldsymbol{\Delta}_1\boldsymbol{\Delta}_2^\top) \right\rangle$$

It is easy to see from the second equation that $\boldsymbol{J}(\boldsymbol{0})$ is symmetric.

Let $-\nabla f(\boldsymbol{0}) = \sum_{i=1}^{d} \mu_i \boldsymbol{u}_{1[i]} \boldsymbol{u}_{1[i]}^\top$ be the eigendecomposition of the symmetric matrix $-\nabla f(\boldsymbol{0})$. Then we have

$$
\begin{aligned}
\boldsymbol{J}(\boldsymbol{0})[\boldsymbol{\Delta}] &= \sum_{i=1}^{d} \mu_i \left( \boldsymbol{u}_{1[i]} \boldsymbol{u}_{1[i]}^\top \boldsymbol{\Delta} + \boldsymbol{\Delta} \boldsymbol{u}_{1[i]} \boldsymbol{u}_{1[i]}^\top \right) \\
&= \sum_{i=1}^{d} \sum_{j=1}^{d} \mu_i \left( \boldsymbol{u}_{1[i]} \boldsymbol{u}_{1[i]}^\top \boldsymbol{\Delta} \boldsymbol{u}_{1[j]} \boldsymbol{u}_{1[j]}^\top + \boldsymbol{u}_{1[j]} \boldsymbol{u}_{1[j]}^\top \boldsymbol{\Delta} \boldsymbol{u}_{1[i]} \boldsymbol{u}_{1[i]}^\top \right) \\
&= \sum_{i=1}^{d} \sum_{j=1}^{d} (\mu_i + \mu_j) \boldsymbol{u}_{1[i]} \boldsymbol{u}_{1[i]}^\top \boldsymbol{\Delta} \boldsymbol{u}_{1[j]} \boldsymbol{u}_{1[j]}^\top \\
&= \sum_{i=1}^{d} \sum_{j=1}^{d} (\mu_i + \mu_j) \left\langle \boldsymbol{\Delta}, \boldsymbol{u}_{1[i]} \boldsymbol{u}_{1[j]}^\top \right\rangle \boldsymbol{u}_{1[i]} \boldsymbol{u}_{1[j]}^\top,
\end{aligned}
$$

which proves (8).

For $\boldsymbol{\Delta} = \boldsymbol{u}_{1[i]} \boldsymbol{u}_{1[j]}^\top + \boldsymbol{u}_{1[j]} \boldsymbol{u}_{1[i]}^\top$, we have

$$\boldsymbol{J}(\boldsymbol{0})[\boldsymbol{\Delta}] = (\mu_i + \mu_j) \boldsymbol{u}_{1[i]} \boldsymbol{u}_{1[j]}^\top + (\mu_i + \mu_j) \boldsymbol{u}_{1[j]} \boldsymbol{u}_{1[i]}^\top = (\mu_i + \mu_j) \boldsymbol{\Delta}.$$

So $\boldsymbol{u}_{1[i]} \boldsymbol{u}_{1[j]}^\top + \boldsymbol{u}_{1[j]} \boldsymbol{u}_{1[i]}^\top$ is an eigenvector of $\boldsymbol{J}(\boldsymbol{0})$ associated with eigenvalue $\mu_i + \mu_j$. Note that $\{\boldsymbol{u}_{1[i]} \boldsymbol{u}_{1[j]}^\top + \boldsymbol{u}_{1[j]} \boldsymbol{u}_{1[i]}^\top : i, j \in [d]\}$ spans all the symmetric matrices, so these are all the eigenvectors in the space of symmetric matrices.

For every antisymmetric matrix $\boldsymbol{\Delta}$ (i.e., $\boldsymbol{\Delta} = -\boldsymbol{\Delta}^\top$), we have

$$\boldsymbol{J}(\boldsymbol{0})[\boldsymbol{\Delta}] = \boldsymbol{J}(\boldsymbol{0})[\boldsymbol{\Delta}^\top] = \boldsymbol{J}(\boldsymbol{0})[-\boldsymbol{\Delta}].$$

So $\boldsymbol{J}(\boldsymbol{0})[\boldsymbol{\Delta}] = \boldsymbol{0}$ and every antisymmetric matrix is an eigenvector associated with eigenvalue $0$.

Since every matrix can be expressed as the sum of a symmetric matrix and an antisymmetric matrix, we have found all the eigenvalues. □

## H.2 EIGENVALUES AT SECOND-ORDER STATIONARY POINTS

Now we study the eigenvalues of $\boldsymbol{J}(\boldsymbol{W}_0)$ when $\boldsymbol{U}_0$ is a second-order stationary point of $\mathcal{L}(\,\cdot\,)$, i.e., $\nabla \mathcal{L}(\boldsymbol{U}_0) = \boldsymbol{0}, D^2\mathcal{L}(\boldsymbol{U}_0)[\boldsymbol{\Delta}, \boldsymbol{\Delta}] \geq 0$ for all $\boldsymbol{\Delta} \in \mathbb{R}^{d \times r}$. We further assume that $\boldsymbol{U}_0$ is full-rank, i.e., $\mathrm{rank}(\boldsymbol{U}_0) = r$. This condition is meet if $\boldsymbol{W}_0 := \boldsymbol{U}_0\boldsymbol{U}_0^T$ is a local minimizer of $f(\,\cdot\,)$ in $\mathbb{S}_d^+$ but not a minimizer in $\mathbb{S}_d^+$.

**Lemma H.1.** *For $r \leq d$, if $\boldsymbol{U}_0 \in \mathbb{R}^{d \times r}$ is a second-order stationary point of $\mathcal{L}(\,\cdot\,)$, then either $\mathrm{rank}(\boldsymbol{U}_0) = \mathrm{rank}(\boldsymbol{W}_0) = r$, or $\boldsymbol{W}_0$ is a minimizer of $f(\,\cdot\,)$ in $\mathbb{S}_d^+$, where $\boldsymbol{W}_0 = \boldsymbol{U}_0\boldsymbol{U}_0^\top$.*

*Proof.* Assume to the contrary that $\boldsymbol{U}_0$ has rank $< r$ and $\boldsymbol{W}_0$ is a minimizer of $f(\,\cdot\,)$ in $\mathbb{S}_d^+$. The former one implies that there exists a unit vector $\boldsymbol{q} \in \mathbb{R}^r$ such that $\boldsymbol{U}_0\boldsymbol{q} = \boldsymbol{0}$, and the latter one implies that there exists $\boldsymbol{v} \in \mathbb{R}^d$ such that $\boldsymbol{v}^\top \nabla f(\boldsymbol{W}_0)\boldsymbol{v} < 0$ by Lemma C.2.

Let $\boldsymbol{\Delta} = \boldsymbol{v}\boldsymbol{q}^\top$. Then we have

$$
\begin{aligned}
D^2\mathcal{L}(\boldsymbol{U}_0)[\boldsymbol{\Delta}, \boldsymbol{\Delta}] &= \left\langle \nabla f(\boldsymbol{W}_0), \boldsymbol{v}\boldsymbol{v}^\top \right\rangle + \frac{1}{2}D^2 f(\boldsymbol{W}_0)[\mathrm{sz}(\boldsymbol{v}(\boldsymbol{U}_0\boldsymbol{q})^\top), \mathrm{sz}(\boldsymbol{v}(\boldsymbol{U}_0\boldsymbol{q})^\top)] \\
&= \left\langle \nabla f(\boldsymbol{W}_0), \boldsymbol{v}\boldsymbol{v}^\top \right\rangle + \frac{1}{2}D^2 f(\boldsymbol{W}_0)[\boldsymbol{0}, \boldsymbol{0}] \\
&= \left\langle \nabla f(\boldsymbol{W}_0), \boldsymbol{v}\boldsymbol{v}^\top \right\rangle.
\end{aligned}
$$

So $D^2\mathcal{L}(\boldsymbol{U}_0)[\boldsymbol{\Delta}, \boldsymbol{\Delta}] < 0$, which leads to a contradiction. $\qquad\square$

By (26), the symmetric matrices $-\nabla f(\boldsymbol{W}_0)$ and $\boldsymbol{W}_0$ commute, so they can be simultaneously diagonalizable. Since (26) also implies that they have different column spans, we can have the following diagonalization:

$$
-\nabla f(\boldsymbol{W}_0) = \sum_{i=1}^{d-r} \mu_i \boldsymbol{v}_i \boldsymbol{v}_i^\top, \qquad \boldsymbol{W}_0 = \sum_{i=d-r+1}^{d} \mu_i \boldsymbol{v}_i \boldsymbol{v}_i^\top. \tag{27}
$$

First we prove the following lemma on the eigenvalues and eigenvectors of the linear operator $-D^2\mathcal{L}(\boldsymbol{U}_0)$:

**Lemma H.2.** *For every $\boldsymbol{\Delta} \in \mathbb{R}^{d \times d}$, if*

$$
\boldsymbol{U}_0\boldsymbol{\Delta}^\top + \boldsymbol{\Delta}\boldsymbol{U}_0^\top = \boldsymbol{0} \tag{28}
$$

*then $\boldsymbol{\Delta}$ is an eigenvector of the linear operator $-D^2\mathcal{L}(\boldsymbol{U}_0)[\,\cdot\,] : \mathbb{R}^{d \times r} \to \mathbb{R}^{d \times r}$ associated with eigenvalue $0$. Moreover, the solutions of (28) spans a linear space of dimension $\frac{r(r-1)}{2}$.*

*Proof.* Suppose $\boldsymbol{U}_0\boldsymbol{\Delta}^\top + \boldsymbol{\Delta}\boldsymbol{U}_0^\top = \boldsymbol{0}$. Then we have $\boldsymbol{U}_0\boldsymbol{\Delta}^\top = -\boldsymbol{\Delta}\boldsymbol{U}_0^\top$, and thus $\boldsymbol{\Delta}^\top = -\boldsymbol{U}_0^+ \boldsymbol{\Delta}\boldsymbol{U}_0^\top$, where $\boldsymbol{U}_0^+$ is the pseudoinverse of the full-rank matrix $\boldsymbol{U}_0$. This implies that there is a matrix $\boldsymbol{R} \in \mathbb{R}^{r \times r}$, such that $\boldsymbol{\Delta} = \boldsymbol{U}_0\boldsymbol{R}$. Then we have

$$
\begin{aligned}
-D^2\mathcal{L}(\boldsymbol{U}_0)[\boldsymbol{\Delta}] &= -\nabla f(\boldsymbol{W}_0)\boldsymbol{U}_0\boldsymbol{R} - D^2 f(\boldsymbol{W}_0)[\boldsymbol{U}_0\boldsymbol{\Delta}^\top + \boldsymbol{\Delta}\boldsymbol{U}_0^\top]\boldsymbol{U}_0 \\
&= -\left(\nabla f(\boldsymbol{W}_0)\boldsymbol{U}_0\right)\boldsymbol{R} - D^2 f(\boldsymbol{W}_0)[\boldsymbol{0}]\boldsymbol{U}_0 \\
&= \boldsymbol{0}.
\end{aligned}
$$

Replacing $\boldsymbol{\Delta}$ with $\boldsymbol{U}_0\boldsymbol{R}$ in (28) gives $\boldsymbol{U}_0(\boldsymbol{R}+\boldsymbol{R}^\top)\boldsymbol{U}_0^\top = \boldsymbol{0}$, which is equivalent to $\boldsymbol{R} = -\boldsymbol{R}^\top$ since $\boldsymbol{U}_0$ is full-rank. Since the dimension of $r \times r$ antisymmetric matrices is $\frac{r(r-1)}{2}$, the span spanned by the solutions of (28) also has dimension $\frac{r(r-1)}{2}$. $\qquad\square$

**Definition H.3** (Eigendecomposition of $-D^2\mathcal{L}(\boldsymbol{U}_0)$). Let

$$
-D^2\mathcal{L}(\boldsymbol{U}_0)[\boldsymbol{\Delta}] = \sum_{p=1}^{rd} \xi_p \left\langle \boldsymbol{E}_p, \boldsymbol{\Delta} \right\rangle \boldsymbol{E}_p
$$

be the eigendecomposition of the symmetric linear operator $-D^2\mathcal{L}(\boldsymbol{U}_0)[\,\cdot\,] : \mathbb{R}^{d \times r} \to \mathbb{R}^{d \times r}$, where $\xi_1, \ldots, \xi_{rd} \in \mathbb{R}$ are eigenvalues, $\boldsymbol{E}_1, \ldots, \boldsymbol{E}_{rd} \in \mathbb{R}^{d \times r}$ are eigenvectors satisfying $\left\langle \boldsymbol{E}_p, \boldsymbol{E}_q \right\rangle = \delta_{pq}$. We enforce $\xi_p$ to be $0$ and $\boldsymbol{E}_p$ to be a solution of (28) for every $rd - \frac{r(r-1)}{2} < p \le rd$.

**Lemma H.4.** *Let $\boldsymbol{A} \in \mathbb{R}^{D \times D}$ be a matrix. If $\{\hat{\boldsymbol{u}}_1, \ldots, \hat{\boldsymbol{u}}_K\}$ is a set of linearly independent left eigenvectors associated with eigenvalues $\hat{\lambda}_1, \ldots, \hat{\lambda}_K$ and $\{\tilde{\boldsymbol{v}}_1, \ldots, \tilde{\boldsymbol{v}}_{D-K}\}$ is a set of linearly independent right eigenvectors associated with eigenvalues $\tilde{\lambda}_1, \ldots, \tilde{\lambda}_{D-K}$, and $\left\langle \hat{\boldsymbol{u}}_i, \tilde{\boldsymbol{v}}_j \right\rangle = 0$ for all $1 \le i \le K, 1 \le j \le D - K$, then $\hat{\lambda}_1, \ldots, \hat{\lambda}_K, \tilde{\lambda}_1, \ldots, \tilde{\lambda}_{D-K}$ are all the eigenvalues of $\boldsymbol{A}$.*

*Proof.* Let $\hat{\boldsymbol{U}} := (\hat{\boldsymbol{u}}_1, \ldots, \hat{\boldsymbol{u}}_K)^\top \in \mathbb{R}^{K \times D}$ and $\tilde{\boldsymbol{V}} := (\tilde{\boldsymbol{v}}_1, \ldots, \tilde{\boldsymbol{v}}_{D-K}) \in \mathbb{R}^{D \times (D-K)}$. Then both $\hat{\boldsymbol{U}}$ and $\tilde{\boldsymbol{V}}$ are full-rank. Let $\hat{\boldsymbol{U}}^+ = \hat{\boldsymbol{U}}^\top(\hat{\boldsymbol{U}}\hat{\boldsymbol{U}}^\top)^{-1}, \tilde{\boldsymbol{V}}^+ = (\tilde{\boldsymbol{V}}^\top\tilde{\boldsymbol{V}})^{-1}\tilde{\boldsymbol{V}}^\top$ be the pseudoinverses of $\hat{\boldsymbol{U}}$ and $\tilde{\boldsymbol{V}}$.

Now we define

$$\boldsymbol{P} := \begin{bmatrix} \hat{\boldsymbol{U}} \\ \tilde{\boldsymbol{V}}^+ \end{bmatrix}, \quad \boldsymbol{Q} := \begin{bmatrix} \hat{\boldsymbol{U}}^+ & \tilde{\boldsymbol{V}} \end{bmatrix}.$$

Then we have

$$\boldsymbol{PQ} = \begin{bmatrix} \hat{\boldsymbol{U}}\hat{\boldsymbol{U}}^+ & \hat{\boldsymbol{U}}\tilde{\boldsymbol{V}} \\ \tilde{\boldsymbol{V}}^+\hat{\boldsymbol{U}}^+ & \tilde{\boldsymbol{V}}^+\tilde{\boldsymbol{V}} \end{bmatrix}.$$

Note that $\hat{\boldsymbol{U}}\hat{\boldsymbol{U}}^+ = \boldsymbol{I}_K$, $\hat{\boldsymbol{U}}\tilde{\boldsymbol{V}} = \boldsymbol{0}$, $\tilde{\boldsymbol{V}}^+\hat{\boldsymbol{U}}^+ = (\tilde{\boldsymbol{V}}^\top\tilde{\boldsymbol{V}})^{-1}(\hat{\boldsymbol{U}}\tilde{\boldsymbol{V}})^\top(\hat{\boldsymbol{U}}\hat{\boldsymbol{U}}^\top)^{-1} = \boldsymbol{0}$, $\tilde{\boldsymbol{V}}^+\tilde{\boldsymbol{V}} = \boldsymbol{I}_{D-K}$. So $\boldsymbol{PQ} = \boldsymbol{I}_D$, or equivalently $\boldsymbol{Q} = \boldsymbol{P}^{-1}$. Then we have

$$\boldsymbol{P}^{-1}\boldsymbol{A}\boldsymbol{P} = \begin{bmatrix} \mathrm{diag}(\hat{\lambda}_1, \ldots, \hat{\lambda}_K) & * \\ \boldsymbol{0} & \mathrm{diag}(\tilde{\lambda}_1, \ldots, \tilde{\lambda}_{D-K}) \end{bmatrix},$$

where $*$ can be any $K \times (D-K)$ matrix. Since $\boldsymbol{P}^{-1}\boldsymbol{A}\boldsymbol{P}$ is upper-triangular, we know that $\boldsymbol{P}^{-1}\boldsymbol{A}\boldsymbol{P}$ has eigenvalues $\hat{\lambda}_1, \ldots, \hat{\lambda}_K, \tilde{\lambda}_1, \ldots, \tilde{\lambda}_{D-K}$, and so does $\boldsymbol{A}$. $\qquad\square$

**Theorem H.5.** *The eigenvalues of $\boldsymbol{J}(\boldsymbol{W}_0)$ can be fully classified into the following 3 types:*

1. *$\mu_i + \mu_j$ is an eigenvalue for every $1 \leq i \leq j \leq d - r$, and $\hat{\boldsymbol{U}}_{ij} := \boldsymbol{v}_i\boldsymbol{v}_j^\top + \boldsymbol{v}_j\boldsymbol{v}_i^\top$ is an associated left eigenvector.*

2. *$\xi_p$ is an eigenvalue for every $1 \leq p \leq rd - \frac{r(r-1)}{2}$, and $\tilde{\boldsymbol{V}}_p := \boldsymbol{E}_p\boldsymbol{U}_0^\top + \boldsymbol{U}_0\boldsymbol{E}_p^\top$ is an associated right eigenvector.*

3. *$0$ is an eigenvalue, and any antisymmetric matrix is an associated right eigenvector, which spans a linear space of dimension $\frac{d(d-1)}{2}$.*

*Proof of Theorem H.5.* We first prove each item respectively, and then prove that these are all the eigenvalues of $\boldsymbol{J}(\boldsymbol{W}_0)$.

**Proof for Item 1.** For $\hat{\boldsymbol{U}}_{ij} = \boldsymbol{v}_i\boldsymbol{v}_j^\top + \boldsymbol{v}_j\boldsymbol{v}_i^\top$, it is easy to check:

$$\mathrm{ac}\{-\nabla f(\boldsymbol{W}_0), \hat{\boldsymbol{U}}_{ij}\} = (\lambda_i + \lambda_j)\hat{\boldsymbol{U}}_{ij}$$
$$\hat{\boldsymbol{U}}_{ij}\boldsymbol{W}_0 = \boldsymbol{0}$$
$$\boldsymbol{W}_0\hat{\boldsymbol{U}}_{ij} = \boldsymbol{0}$$

So we have

$$\boldsymbol{J}(\boldsymbol{W}_0)[\boldsymbol{\Delta}, \hat{\boldsymbol{U}}_{ij}] = (\lambda_i + \lambda_j)\left\langle \boldsymbol{\Delta}, \hat{\boldsymbol{U}}_{ij} \right\rangle - D^2 f(\boldsymbol{W}_0)[\boldsymbol{\Delta}, \boldsymbol{0}] = (\lambda_i + \lambda_j)\left\langle \boldsymbol{\Delta}, \hat{\boldsymbol{U}}_{ij} \right\rangle,$$

which shows that $\hat{\boldsymbol{U}}_{ij}$ is a left eigenvector associated with eigenvalue $\lambda_i + \lambda_j$.

**Proof for Item 2.** By definition of eigenvector, we have $-D^2\mathcal{L}(\boldsymbol{U}_0)[\boldsymbol{E}_p] = \xi_p\boldsymbol{E}_p$, so

$$\xi_p\boldsymbol{E}_p = -\nabla f(\boldsymbol{W}_0)\boldsymbol{E}_p - D^2 f(\boldsymbol{W}_0)[\boldsymbol{U}_0\boldsymbol{E}_p^\top + \boldsymbol{E}_p\boldsymbol{U}_0^\top]\boldsymbol{U}_0.$$

Right-multiplying both sides by $\boldsymbol{U}_0^\top$, we get

$$\begin{aligned} \xi_p\boldsymbol{E}_p\boldsymbol{U}_0^\top &= -\nabla f(\boldsymbol{W}_0)\boldsymbol{E}_p\boldsymbol{U}_0^\top - D^2 f(\boldsymbol{W}_0)[\tilde{\boldsymbol{V}}_p]\boldsymbol{W}_0 \\ &= -\nabla f(\boldsymbol{W}_0)(\boldsymbol{E}_p\boldsymbol{U}_0^\top + \boldsymbol{U}_0\boldsymbol{E}_p^\top) - D^2 f(\boldsymbol{W}_0)[\tilde{\boldsymbol{V}}_p]\boldsymbol{W}_0 \\ &= -\nabla f(\boldsymbol{W}_0)\tilde{\boldsymbol{V}}_p - D^2 f(\boldsymbol{W}_0)[\tilde{\boldsymbol{V}}_p]\boldsymbol{W}_0, \end{aligned}$$

where the second equality uses the fact that $\nabla f(\boldsymbol{W}_0)\boldsymbol{U}_0 = \boldsymbol{0}$ since $\boldsymbol{U}_0$ is a critical point. Taking both sides into $\mathrm{sz}(\cdot)$ gives

$$\begin{aligned} \xi_p\tilde{\boldsymbol{V}}_p &= -\mathrm{sz}(\nabla f(\boldsymbol{W}_0)\tilde{\boldsymbol{V}}_p) - \mathrm{sz}(D^2 f(\boldsymbol{W}_0)[\tilde{\boldsymbol{V}}_p]\boldsymbol{W}_0) \\ &= \boldsymbol{J}(\boldsymbol{W}_0)[\tilde{\boldsymbol{V}}_p], \end{aligned}$$

which proves that $\tilde{\boldsymbol{V}}_p$ is a right eigenvector associated with eigenvalue $\xi_p$.

**Proof for Item 3.** Since $\nabla f(\boldsymbol{W})$ is symmetric, $\boldsymbol{g}(\boldsymbol{W})$ is also symmetric. For any $\boldsymbol{\Delta} = -\boldsymbol{\Delta}^\top$,

$$\boldsymbol{J}(\boldsymbol{W}_0)[\boldsymbol{\Delta}] = \boldsymbol{J}(\boldsymbol{W}_0)[\boldsymbol{\Delta}^\top] = \boldsymbol{J}(\boldsymbol{W}_0)[-\boldsymbol{\Delta}].$$

So $\boldsymbol{J}(\boldsymbol{W}_0)[\boldsymbol{\Delta}] = \boldsymbol{0}$ and $\boldsymbol{\Delta}$ is an eigenvector associated with eigenvalue 0.

**No other eigenvalues.** Let $\mathbb{S}_d$ be the space of symmetric matrices and $\mathbb{A}_d$ be the space of anti-symmetric matrices. It is easy to see that $\mathbb{S}_d$ and $\mathbb{A}_d$ are orthogonal to each other, and $\mathbb{S}_d$ and $\mathbb{A}_d$ are invariant subspaces of $\boldsymbol{J}(\boldsymbol{W}_0)[\boldsymbol{\Delta}]$. Let $\boldsymbol{h} : \mathbb{S}_d \to \mathbb{S}_d, \boldsymbol{\Delta} \mapsto \boldsymbol{J}(\boldsymbol{W}_0)[\boldsymbol{\Delta}]$ be the linear operator $\boldsymbol{J}(\boldsymbol{W}_0)[\boldsymbol{\Delta}]$ restricted on symmetric matrices. We only need to prove that $\boldsymbol{h}$ is diagonalizable.

It is easy to see that $\{\hat{\boldsymbol{U}}_{ij}\}$ are linearly independent to each other and thus spans a subspace of $\mathbb{S}_d$ with dimension $\frac{(d-r)(d-r+1)}{2}$. We can also prove that $\{\tilde{\boldsymbol{V}}_p\}$ spans a subspace of $\mathbb{S}_d$ with dimension $rd - \frac{r(r-1)}{2}$ by contradiction. Assume to the contrary that there exists scalars $\alpha_p$ for $1 \leq p \leq rd - r(r-1)/2$, not all zero, such that $\sum_{p=1}^{rd-r(r-1)/2} \alpha_p \tilde{\boldsymbol{V}}_p = \boldsymbol{0}$. Then $\sum_{p=1}^{rd-r(r-1)/2} \alpha_p \boldsymbol{E}_p$ is a solution of (28). However, this suggests that $\sum_{p=1}^{rd-r(r-1)/2} \alpha_p \boldsymbol{E}_p$ lies in the span of $\{\boldsymbol{E}_p\}_{rd-r(r-1)/2 < p \leq rd}$, which contradicts to the linear independence of $\{\boldsymbol{E}_p\}_{1 \leq p \leq rd}$.

Note that

$$\frac{(d-r)(d-r+1)}{2} + \left(rd - \frac{r(r-1)}{2}\right) = \frac{d(d+1)}{2} = \dim(\mathbb{S}_d).$$

Also note that $\left\langle \hat{\boldsymbol{U}}_{ij}, \tilde{\boldsymbol{V}}_p \right\rangle = 2\boldsymbol{v}_i^\top \boldsymbol{E}_p \boldsymbol{U}_0^\top \boldsymbol{v}_j + 2\boldsymbol{v}_j^\top \boldsymbol{E}_p \boldsymbol{U}_0^\top \boldsymbol{v}_i = 0$. By Lemma H.4, Items 1 and 2 give all the eigenvalues of $\boldsymbol{h}$, and thus Items 1, 2, 3 give all the eigenvalues of $\boldsymbol{J}(\boldsymbol{W}_0)$. $\qquad\square$

# I   PROOFS FOR THE DEPTH-2 CASE

## I.1   PROOF FOR THEOREM 5.6

*Proof for Theorem 5.6.* Since $\boldsymbol{W}(t)$ is always symmetric, it suffices to study the dynamics of the lower triangle of $\boldsymbol{W}(t)$. For any symmetric matrix $\boldsymbol{W} \in \mathbb{S}_d$, let $\text{vec}_{\text{LT}}(\boldsymbol{W}) \in \mathbb{R}^{\frac{d(d+1)}{2}}$ be the vector consisting of the $\frac{d(d+1)}{2}$ entries of $\boldsymbol{W}$ in the lower triangle, permuted according to some fixed order.

Let $\boldsymbol{g}(\boldsymbol{W})$ be the function defined in (2), which always maps symmetric matrices to symmetric matrices. Let $\tilde{\boldsymbol{g}} : \mathbb{R}^{\frac{d(d+1)}{2}} \to \mathbb{R}^{\frac{d(d+1)}{2}}$ be the function such that $\tilde{\boldsymbol{g}}(\text{vec}_{\text{LT}}(\boldsymbol{W})) = \text{vec}_{\text{LT}}(\boldsymbol{g}(\boldsymbol{W}))$ for any $\boldsymbol{W} \in \mathbb{S}_d$. For $\boldsymbol{W}(t)$ evolving with (2), we view $\text{vec}_{\text{LT}}(\boldsymbol{W}(t))$ as a dynamical system.

$$\frac{\mathrm{d}}{\mathrm{d}t} \text{vec}_{\text{LT}}(\boldsymbol{W}(t)) = \tilde{\boldsymbol{g}}(\text{vec}_{\text{LT}}(\boldsymbol{W}(t))).$$

By Lemma 5.4, the spaces of symmetric matrices $\mathbb{S}_d$ and antisymmetric matrices $\mathbb{A}_d$ are invariant subspaces of $\boldsymbol{J}(\boldsymbol{0})$, and $\left\{(\mu_i + \mu_j, \boldsymbol{u}_{1[i]}\boldsymbol{u}_{1[j]}^\top + \boldsymbol{u}_{1[j]}\boldsymbol{u}_{1[i]}^\top)\right\}_{1 \leq i \leq j \leq d}$ is the set of all the eigenvalues and eigenvectors in the invariant subspace $\mathbb{S}_d$. Thus, $\tilde{\mu}_1 := 2\mu_1$ and $\tilde{\mu}_2 := \mu_1 + \mu_2$ are the largest and second largest eigenvalues of the Jacobian of $\tilde{\boldsymbol{g}}(\cdot)$ at $\text{vec}_{\text{LT}}(\boldsymbol{W}) = \boldsymbol{0}$, and $\tilde{\boldsymbol{u}}_1 = \tilde{\boldsymbol{v}}_1 = \boldsymbol{u}_1\boldsymbol{u}_1^\top$ are the corresponding left and right eigenvectors of the top eigenvalue. Then it is easy to translate Theorem 5.3 to Theorem 5.6. $\qquad\square$

## I.2   PROOF FOR THEOREM 5.8

The proof for Theorem 5.8 relies on the following Lemma on the gradient flow around a local minimizer:

**Lemma I.1.** *If $\bar{\boldsymbol{\theta}}$ is a local minimizer of $\mathcal{L}(\boldsymbol{\theta})$ and for all $\|\boldsymbol{\theta} - \bar{\boldsymbol{\theta}}\|_2 \leq r$, $\boldsymbol{\theta}$ satisfies Łojasiewicz inequality:*

$$\|\nabla\mathcal{L}(\boldsymbol{\theta})\|_2 \geq c\left(\mathcal{L}(\boldsymbol{\theta}) - \mathcal{L}(\bar{\boldsymbol{\theta}})\right)^\mu$$

*for some $\mu \in [1/2, 1)$, then the gradient flow $\boldsymbol{\theta}(t) = \phi(\boldsymbol{\theta}_0, t)$ converges to a point $\boldsymbol{\theta}_\infty$ near $\bar{\boldsymbol{\theta}}$ if $\boldsymbol{\theta}_0$ is close enough to $\bar{\boldsymbol{\theta}}$, and the distance can be bounded by $\|\boldsymbol{\theta}_\infty - \bar{\boldsymbol{\theta}}\|_2 = O(\|\boldsymbol{\theta}_0 - \bar{\boldsymbol{\theta}}\|_2^{2(1-\mu)})$.*

*Proof.* For every $t \geq 0$, if $\|\boldsymbol{\theta}(t) - \bar{\boldsymbol{\theta}}\|_2 \leq r$,

$$\frac{\mathrm{d}}{\mathrm{d}t}\left(\mathcal{L}(\boldsymbol{\theta}(t)) - \mathcal{L}(\bar{\boldsymbol{\theta}})\right)^{1-\mu} = (1-\mu)\left(\mathcal{L}(\boldsymbol{\theta}(t)) - \mathcal{L}(\bar{\boldsymbol{\theta}})\right)^{-\mu} \cdot \left\langle \nabla\mathcal{L}, \frac{\mathrm{d}\boldsymbol{\theta}}{\mathrm{d}t} \right\rangle$$

$$= -(1-\mu)\left(\mathcal{L}(\boldsymbol{\theta}(t)) - \mathcal{L}(\bar{\boldsymbol{\theta}})\right)^{-\mu} \cdot \|\nabla\mathcal{L}\|_2 \cdot \left\|\frac{\mathrm{d}\boldsymbol{\theta}}{\mathrm{d}t}\right\|_2$$

$$\leq -(1-\mu)c\left\|\frac{\mathrm{d}\boldsymbol{\theta}}{\mathrm{d}t}\right\|_2.$$

Therefore, $\|\boldsymbol{\theta}(t) - \boldsymbol{\theta}_0\|_2 \leq \int_0^t \left\|\frac{\mathrm{d}\boldsymbol{\theta}}{\mathrm{d}t}\right\|_2 \mathrm{d}t \leq \frac{1}{(1-\mu)c}\mathcal{L}(\boldsymbol{\theta}_0)^{1-\mu} = O(\|\boldsymbol{\theta}_0 - \bar{\boldsymbol{\theta}}\|_2^{2(1-\mu)})$. If we choose $\|\boldsymbol{\theta}(t) - \bar{\boldsymbol{\theta}}\|_2$ small enough, then $\|\boldsymbol{\theta}(t) - \bar{\boldsymbol{\theta}}\|_2 \leq \|\boldsymbol{\theta}(t) - \boldsymbol{\theta}_0\|_2 + \|\boldsymbol{\theta}_0 - \bar{\boldsymbol{\theta}}\|_2 = O(\|\boldsymbol{\theta}_0 - \bar{\boldsymbol{\theta}}\|_2^{2(1-\mu)}) < r$, and thus $\int_0^{+\infty} \left\|\frac{\mathrm{d}\boldsymbol{\theta}}{\mathrm{d}t}\right\|_2 \mathrm{d}t$ is convergent and finite. This implies that $\boldsymbol{\theta}_\infty := \lim_{t\to+\infty}\boldsymbol{\theta}(t)$ exists and $\|\boldsymbol{\theta}_\infty - \bar{\boldsymbol{\theta}}\|_2 = O(\|\boldsymbol{\theta}_0 - \bar{\boldsymbol{\theta}}\|_2^{2(1-\mu)})$. $\square$

*Proof for Theorem 5.8.* Since $\boldsymbol{W}_1^{\mathrm{G}}(t) \in \mathbb{S}_{d,\leq 1}^+$ satisfies (2), there exists $\boldsymbol{u}(t) \in \mathbb{R}^d$ such that $\boldsymbol{u}(t)\boldsymbol{u}(t)^\top = \boldsymbol{W}_1^{\mathrm{G}}(t)$ and $\boldsymbol{u}(t)$ satisfies (1), i.e., $\frac{\mathrm{d}\boldsymbol{u}}{\mathrm{d}t} = -\nabla\mathcal{L}(\boldsymbol{u})$, where $\mathcal{L} : \mathbb{R}^d \to \mathbb{R}, \boldsymbol{u} \mapsto \frac{1}{2}f(\boldsymbol{u}\boldsymbol{u}^\top)$. If $\boldsymbol{W}_1^{\mathrm{G}}(t)$ does not diverge to infinity, then so does $\boldsymbol{u}(t)$. This implies that there is a limit point $\bar{\boldsymbol{u}}$ of the set $\{\boldsymbol{u}(t) : t \geq 0\}$.

Let $\mathcal{U} := \{\boldsymbol{u} : \mathcal{L}(\boldsymbol{u}) \geq \mathcal{L}(\bar{\boldsymbol{u}})\}$. Since $\mathcal{L}(\boldsymbol{u}(t))$ is non-increasing, we have $\boldsymbol{u}(t) \in \mathcal{U}$ for all $t$. Note that $\bar{\boldsymbol{u}}$ is a local minimizer of $\mathcal{L}(\cdot)$ in $\mathcal{U}$. By analyticity of $f(\cdot)$, Łojasiewicz inequality holds for $\mathcal{L}(\cdot)$ around $\bar{\boldsymbol{u}}$ (Łojasiewicz, 1965). Applying Lemma I.1 for $\mathcal{L}$ restricted on $\mathcal{U}$, we know that if $\boldsymbol{u}(t_0)$ is sufficiently close to $\bar{\boldsymbol{u}}$, the remaining length of the trajectory of $\boldsymbol{u}(t)$ ($t \geq t_0$) is finite and thus $\lim_{t\to+\infty}\boldsymbol{u}(t)$ exists. As $\bar{\boldsymbol{u}}$ is a limit point, this limit can only be $\bar{\boldsymbol{u}}$. Therefore, $\overline{\boldsymbol{W}}_1 := \lim_{t\to+\infty}\boldsymbol{W}_1^{\mathrm{G}}(t) = \bar{\boldsymbol{u}}\bar{\boldsymbol{u}}^\top$ exists.

If $\overline{\boldsymbol{W}}_1$ is a minimizer of $f(\cdot)$, $\overline{\boldsymbol{U}} = (\bar{\boldsymbol{u}}, \boldsymbol{0}, \cdots, \boldsymbol{0}) \in \mathbb{R}^{d\times d}$ is also a minimizer of $\mathcal{L} : \mathbb{R}^{d\times d} \to \mathbb{R}, \boldsymbol{U} \mapsto \frac{1}{2}f(\boldsymbol{U}\boldsymbol{U}^\top)$. By analyticity of $f(\cdot)$, Łojasiewicz inequality holds for $\mathcal{L}(\cdot)$ around $\overline{\boldsymbol{U}}$. For every $\epsilon > 0$, we can always find a time $t_\epsilon$ such that $\|\boldsymbol{u}(t_\epsilon) - \bar{\boldsymbol{u}}\|_2 \leq \epsilon/2$. On the other hand, by Theorem 5.6, there exists a number $\alpha_\epsilon$ such that for every $\alpha < \alpha_\epsilon$,

$$\left\|\phi(\boldsymbol{W}_\alpha, T(\boldsymbol{W}_\alpha) + t_\epsilon) - \boldsymbol{W}_1^{\mathrm{G}}(t_\epsilon)\right\|_2 \leq \epsilon/2, \quad \text{where} \quad T(\boldsymbol{W}) := \frac{1}{2\mu_1}\log\frac{1}{\langle\boldsymbol{W}, \boldsymbol{u}_1\boldsymbol{u}_1^\top\rangle}.$$

Combining these together we have $\left\|\phi(\boldsymbol{W}_\alpha, T(\boldsymbol{W}_\alpha) + t_\epsilon) - \overline{\boldsymbol{W}}_1\right\|_2 \leq \epsilon$.

It is easy to construct a factorization $\phi(\boldsymbol{W}_\alpha, T(\boldsymbol{W}_\alpha) + t_\epsilon) := \boldsymbol{U}_{\alpha,\epsilon}\boldsymbol{U}_{\alpha,\epsilon}^\top$ such that $\left\|\boldsymbol{U}_{\alpha,\epsilon} - \overline{\boldsymbol{U}}\right\|_2 = O(\epsilon)$, e.g., we can find an arbitrary factorization and then right-multiply an orthogonal matrix so that the row vector with the largest norm aligns with the direction of $\bar{\boldsymbol{u}}$. Applying Lemma I.1, we know that gradient flow starting with $\boldsymbol{U}_{\alpha,\epsilon}$ converges to a point that is only $O(\epsilon^{2(1-\mu)})$ far from $\bar{\boldsymbol{u}}$. So we have

$$\left\|\lim_{t\to+\infty}\phi(\boldsymbol{W}_\alpha, T(\boldsymbol{W}_\alpha) + t) - \overline{\boldsymbol{W}}_1\right\|_2 = O(\epsilon^{2(1-\mu)}).$$

Taking $\epsilon \to 0$ complete the proof. $\square$

### I.3 PROOF FOR THEOREM 5.11

**Theorem I.2.** *Let $\overline{\boldsymbol{W}}$ be a critical point of (2) satisfying that $\overline{\boldsymbol{W}}$ is a local minimizer of $f(\cdot)$ in $\mathbb{S}_{d,\leq r}^+$ for some $r \geq 1$ but not a minimizer in $\mathbb{S}_d^+$. Let $-\nabla f(\overline{\boldsymbol{W}}) = \sum_{i=1}^d \mu_i \boldsymbol{v}_i \boldsymbol{v}_i^\top$ be the eigendecomposition of $-\nabla f(\overline{\boldsymbol{W}})$. If $\mu_1 > \mu_2$, the following limit exists and is a solution of (2).*

$$\boldsymbol{W}^{\mathrm{G}}(t) := \lim_{\epsilon\to 0}\phi\left(\overline{\boldsymbol{W}} + \epsilon\boldsymbol{v}_1\boldsymbol{v}_1^\top, \frac{1}{2\mu_1}\log\frac{1}{\epsilon} + t\right).$$

*For $\{\boldsymbol{W}_\alpha\} \subseteq \mathbb{S}_d^+$, if there exists time $T_\alpha \in \mathbb{R}$ for every $\alpha$ so that $\phi(\boldsymbol{W}_\alpha, T_\alpha)$ converges to $\overline{\boldsymbol{W}}$ with positive alignment with the top principal component $\boldsymbol{v}_1\boldsymbol{v}_1^\top$ as $\alpha \to 0$, then $\forall t \in \mathbb{R}$,*

$$\lim_{\alpha\to 0}\phi\left(\boldsymbol{W}_\alpha, T_\alpha + \frac{1}{2\mu_1}\log\frac{1}{\langle\phi(\boldsymbol{W}_\alpha, T_\alpha), \boldsymbol{v}_1\boldsymbol{v}_1^\top\rangle} + t\right) = \boldsymbol{W}^{\mathrm{G}}(t).$$

*Moreover, there exists a constant $C > 0$ such that*

$$\left\| \phi \left( \boldsymbol{W}_\alpha, T_\alpha + \frac{1}{2\mu_1} \log \frac{1}{\langle \phi(\boldsymbol{W}_\alpha, T_\alpha), \boldsymbol{v}_1 \boldsymbol{v}_1^\top \rangle} + t \right) - \boldsymbol{W}^{\mathrm{G}}(t) \right\|_{\mathrm{F}} \le C \left\| \phi(\boldsymbol{W}_\alpha, T_\alpha) \right\|_{\mathrm{F}}^{\frac{\tilde{\gamma}}{2\mu_1 + \tilde{\gamma}}}$$

*for every sufficiently small $\alpha$, where $\tilde{\gamma} := 2\mu_1 - \max\{\mu_1 + \mu_2, 0\}$.*

*Proof.* Following Appendix I.1, we view $\mathrm{vec}_{\mathrm{LT}}(\boldsymbol{W}(t))$ as a dynamical system.

$$\frac{\mathrm{d}}{\mathrm{d}t} \mathrm{vec}_{\mathrm{LT}}(\boldsymbol{W}(t)) = \tilde{\boldsymbol{g}}(\mathrm{vec}_{\mathrm{LT}}(\boldsymbol{W}(t))).$$

Let $\overline{\boldsymbol{W}} = \overline{\boldsymbol{U}} \, \overline{\boldsymbol{U}}^\top$ be a factorization of $\overline{\boldsymbol{W}}$, where $\overline{\boldsymbol{U}} \in \mathbb{R}^{d \times r}$. Since $\overline{\boldsymbol{W}}$ is a local minimizer of $f(\cdot)$ in $\mathbb{S}_{d, \le r}^+$, $\overline{\boldsymbol{U}}$ is also a local minimizer of $\mathcal{L} : \mathbb{R}^{d \times r} \to \mathbb{R}, \boldsymbol{U} \mapsto \frac{1}{2} f(\boldsymbol{U}\boldsymbol{U}^\top)$. Since $\overline{\boldsymbol{W}}$ is not a minimizer of $f(\cdot)$ in $\mathbb{S}_d^+$, by Lemma H.1, $\overline{\boldsymbol{U}}$ is full-rank. By Theorem H.5, $\boldsymbol{J}(\overline{\boldsymbol{W}})$ has eigenvalues $\mu_i + \mu_j, \xi_p, 0$. By a similar argument as in Appendix I.1, the Jacobian of $\tilde{\boldsymbol{g}}$ at $\mathrm{vec}_{\mathrm{LT}}(\boldsymbol{W}(t))$ has eigenvalues $\mu_i + \mu_j, \xi_p$.

Since $\overline{\boldsymbol{U}}$ is a local minimizer, $\xi_p \le 0$ for all $p$. If $\mu_1 > \mu_2$, then $2\mu_1$ is the unique largest eigenvalue, and Theorem H.5 shows that $\mathrm{vec}_{\mathrm{LT}}(\boldsymbol{v}_1 \boldsymbol{v}_1^\top)$ is a left eigenvector associated with $2\mu_1$. The eigenvalue gap $\tilde{\gamma} := 2\mu_1 - \max\{\mu_1 + \mu_2, \max\{\xi_p : 1 \le p \le rd - \frac{r(r-1)}{2}\}\} \ge 2\mu_1 - \max\{\mu_1 + \mu_2, 0\}$.

Also note that $\langle \phi(\boldsymbol{W}_\alpha, T_\alpha) - \overline{\boldsymbol{W}}, \boldsymbol{v}_1 \boldsymbol{v}_1^\top \rangle = \langle \phi(\boldsymbol{W}_\alpha, T_\alpha), \boldsymbol{v}_1 \boldsymbol{v}_1^\top \rangle$ because $\langle \overline{\boldsymbol{W}}, \boldsymbol{v}_1 \boldsymbol{v}_1^\top \rangle = 0$ by (27). If $\phi(\boldsymbol{W}_\alpha, T_\alpha)$ converges to $\overline{\boldsymbol{W}}$ as $\alpha \to 0$, then it has positive alignment with $\boldsymbol{v}_1 \boldsymbol{v}_1^\top$ iff $\liminf_{\alpha \to 0} \frac{\langle \phi(\boldsymbol{W}_\alpha, T_\alpha), \boldsymbol{v}_1 \boldsymbol{v}_1^\top \rangle}{\| \phi(\boldsymbol{W}_\alpha, T_\alpha) - \overline{\boldsymbol{W}} \|_{\mathrm{F}}} > 0$. Then it is easy to translate Theorem 5.3 to Theorem I.2. $\square$

## I.4    GRADIENT FLOW ONLY FINDS MINIMIZERS (PROOF FOR THEOREM 5.10)

The proof for Theorem 5.10 is based on the following two theorems from the literature.

**Theorem I.3** (Theorem 3.1 in Du and Lee 2018). *Let $f : \mathbb{R}^{d \times d} \to \mathbb{R}$ be a $\mathcal{C}^2$ convex function. Then $\mathcal{L} : \mathbb{R}^{d \times k} \to \mathbb{R}, \mathcal{L}(\boldsymbol{U}) = f(\boldsymbol{U}\boldsymbol{U}^\top), k \ge d$ satisfies that (1). Every local minimizer of $\mathcal{L}$ is also a global minimizer; (2). All saddles are strict. Here saddles denote those stationary points whose hessian are not positive semi-definite (thus including local maximizers).* [2]

**Theorem I.4** (Theorem 2 in Lee et al. 2017). *Let $\boldsymbol{g}$ be a $\mathcal{C}^1$ mapping from $\mathcal{X} \to \mathcal{X}$ and $\det(D\boldsymbol{g}(x)) \ne 0$ for all $\boldsymbol{x} \in \mathcal{X}$. Then the set of initial points that converge to an unstable fixed point has measure zero, $\mu\left(\{\boldsymbol{x}_0 : \lim_{k \to \infty} \boldsymbol{g}^k(\boldsymbol{x}_0) \in \mathcal{A}_{\boldsymbol{g}}^*\}\right) = 0$, where $\mathcal{A}_{\boldsymbol{g}}^* = \{\boldsymbol{x} : \boldsymbol{g}(\boldsymbol{x}) = \boldsymbol{x}, \max_i |\lambda_i(D\boldsymbol{g}(\boldsymbol{x}))| > 1\}$.*

**Theorem I.5** (GF only finds minimizers, a continuous analog of Theorem I.4). *Let $\boldsymbol{f} : \mathbb{R}^d \to \mathbb{R}^d$ be a $\mathcal{C}^1$-smooth function, and $\phi : \mathbb{R}^d \times \mathbb{R} \to \mathbb{R}^d$ be the solution of the following differential equation,*

$$\frac{\mathrm{d}\phi(\boldsymbol{x}, t)}{\mathrm{d}t} = \boldsymbol{f}(\phi(\boldsymbol{x}, t)), \quad \phi(\boldsymbol{x}, 0) = \boldsymbol{x}, \quad \forall \boldsymbol{x} \in \mathbb{R}^d, t \in \mathbb{R}.$$

*Then the set of initial points that converge to a unstable critical point has measure zero, $\mu\left(\left\{\boldsymbol{x}_0 : \lim_{t \to \infty} \phi(\boldsymbol{x}_0, t) \in \mathcal{U}_{\boldsymbol{f}}^*\right\}\right) = 0$, where $\mathcal{U}_{\boldsymbol{f}}^* = \{\boldsymbol{x} : \boldsymbol{f}(\boldsymbol{x}) = \boldsymbol{0}, \lambda_1(D\boldsymbol{f}(\boldsymbol{x})) > 0\}$ and $D\boldsymbol{f}$ is the Jacobian matrix of $\boldsymbol{f}$.*

*Proof of Theorem I.5.* By Theorem 1 in Section 2.3, Perko (2013), we know $\phi(\cdot, \cdot)$ is $\mathcal{C}^1$-smooth for both $x, t$. We let $\boldsymbol{g}(x) = \phi(x, 1)$, then we know $\boldsymbol{g}^{-1}(x) = \phi(x, -1)$ and both $\boldsymbol{g}, \boldsymbol{g}^{-1}$ are $\mathcal{C}^1$-smooth. Note that $D\boldsymbol{g}^{-1}(x)$ is the inverse matrix of $D\boldsymbol{g}(x)$. So both of the two matrices are invertible. Thus we can apply Theorem I.4 and we know $\mu\left(\{x_0 : \lim_{k \to \infty} \boldsymbol{g}^k(x_0) \in \mathcal{A}_{\boldsymbol{g}}^*\}\right) = 0$.

Note that if $\lim_{t \to \infty} \phi(x, t)$ exists, then $\lim_{k \to \infty} \boldsymbol{g}^k(\boldsymbol{x}) = \lim_{t \to \infty} \phi(\boldsymbol{x}, t)$. It remains to show that $\mathcal{U}_{\boldsymbol{f}}^* \subseteq \mathcal{A}_{\boldsymbol{g}}^*$. For $\boldsymbol{f}(\boldsymbol{x}_0) = \boldsymbol{0}$, we have $\phi(\boldsymbol{x}_0, t) = \boldsymbol{x}_0$ and thus $\boldsymbol{g}(\boldsymbol{x}_0) = \boldsymbol{x}_0$. Now it suffices to prove

---

[2] Though the original theorem is proven for convex functions of form $\sum_{i=1}^n \ell(\boldsymbol{x}_i \boldsymbol{U}\boldsymbol{U}^\top \boldsymbol{x}_i^\top, y_i)$, where $\ell(\cdot, \cdot)$ is $\mathcal{C}^2$ convex for its first variable. By scrutinizing their proof, we can see the assumption can be relaxed to $f$ is $\mathcal{C}^2$ convex.

that $\lambda_1(D\boldsymbol{g}(\boldsymbol{x}_0)) > 1$. For every $t \in [0, 1]$, by Corollary of Theorem 1 in Section 2.3, Perko (2013), we have $\frac{\partial}{\partial t} D\phi(\boldsymbol{x}, t) = D\boldsymbol{f}(\phi(\boldsymbol{x}, t))D\phi(\boldsymbol{x}, t), \forall \boldsymbol{x}, t$. Thus,

$$\frac{\partial}{\partial t} D\phi(x_0, t) = D\boldsymbol{f}(\phi(x_0, t))D\phi(x_0, t) = D\boldsymbol{f}(x_0)D\phi(x_0, t).$$

Solving this ODE gives $D\boldsymbol{g}(\boldsymbol{x}_0) = D\phi(\boldsymbol{x}, 1) = e^{D\boldsymbol{f}(x_0)}D\phi(\boldsymbol{x}, 0) = e^{D\boldsymbol{f}(x_0)}$, where the last equality is due to $D\phi(\boldsymbol{x}, 0) \equiv \boldsymbol{I}, \forall \boldsymbol{x}$. Combining this with $\lambda_1(D\boldsymbol{f}(x_0)) > 0$, we have $\lambda_1(D\boldsymbol{g}(x_0)) > 1$.

Thus we have $\mathcal{U}_{\boldsymbol{f}}^* := \{\boldsymbol{x}_0 : \boldsymbol{f}(\boldsymbol{x}_0) = \boldsymbol{0}, \lambda_1(D\boldsymbol{f}(\boldsymbol{x}_0)) > 0\} \subseteq \mathcal{A}_{\boldsymbol{g}}^*$, which implies that $\{\boldsymbol{x}_0 : \lim_{t\to\infty} \phi(\boldsymbol{x}_0, t) \in \mathcal{U}^*\} \subseteq \{\boldsymbol{x}_0 : \lim_{k\to\infty} \boldsymbol{g}^k(\boldsymbol{x}_0) \in \mathcal{A}_{\boldsymbol{g}}^*\}$ $\qquad\square$

**Theorem 5.10.** *Let $f : \mathbb{R}^{d\times d} \to \mathbb{R}$ be a convex $\mathcal{C}^2$-smooth function. (1). All stationary points of $\mathcal{L} : \mathbb{R}^{d\times d} \to \mathbb{R}, \mathcal{L}(\boldsymbol{U}) = \frac{1}{2}f(\boldsymbol{U}\boldsymbol{U}^\top)$ are either strict saddles or global minimizers; (2). For any random initialization, GF (1) converges to strict saddles of $\mathcal{L}(\boldsymbol{U})$ with probability 0.*

*Proof of Theorem 5.10.* For (1), by Theorem I.3, we immediately know all the stationary points of $\mathcal{L}(\cdot)$ are either global minimizers or strict saddles. (2) is just a direct consequence of Theorem I.5 by setting $\boldsymbol{f}$ in the above proof to $-\nabla\mathcal{L}$. $\qquad\square$

## J   EQUIVALENCE BETWEEN GF AND GLRL

In this section we elaborate on the theoretical evidence that GF and GLRL are equivalent generically, including the case where GLRL does not end in the first phase. The word "generically" used when we want to assume one of the following regularity conditions:

1. We want to assume that GF converges to a local minimizer (i.e., GF does not get stuck on saddle points);

2. We want to assume that the top eigenvalue $\lambda_1(-\nabla f(\boldsymbol{W}))$ is unique for a critical point $\boldsymbol{W}$ of (2) that is not a minimizer of $f(\cdot)$ in $\mathbb{S}_d^+$;

3. We want to assume that a convergent sequence of PSD matrices $\boldsymbol{W}_\alpha \to \overline{\boldsymbol{W}}$ has positive alignment with $\boldsymbol{v}\boldsymbol{v}^\top$ for some fixed vector $\boldsymbol{v}$ with $\langle \overline{\boldsymbol{W}}, \boldsymbol{v}\boldsymbol{v}^\top \rangle = 0$, i.e., for a convergent sequence of PSD matrices $\boldsymbol{W}_\alpha \to \overline{\boldsymbol{W}}$, it holds for sure that $\liminf_{\alpha\to 0} \left\langle \frac{\boldsymbol{W}_\alpha - \overline{\boldsymbol{W}}}{\|\boldsymbol{W}_\alpha - \overline{\boldsymbol{W}}\|_{\mathrm{F}}}, \boldsymbol{v}\boldsymbol{v}^\top \right\rangle = \liminf_{\alpha\to 0} \frac{\langle \boldsymbol{W}_\alpha, \boldsymbol{v}\boldsymbol{v}^\top \rangle}{\|\boldsymbol{W}_\alpha - \overline{\boldsymbol{W}}\|_{\mathrm{F}}} \geq 0$, and we further assume that the inequality is strict generically.

Theorem I.2 uncovers how GF with infinitesimal initialization generically behaves. Let $\overline{\boldsymbol{W}}_0 := \boldsymbol{0}$. For every $r \geq 1$, if $\overline{\boldsymbol{W}}_{r-1}$ is a local minimizer in $\mathbb{S}_{d, \leq r-1}^+$ but not a minimizer in $\mathbb{S}_d^+$, then $\lambda_1(-\nabla f(\overline{\boldsymbol{W}}_{r-1})) > 0$ by Lemma C.2. Generically, the top eigenvalue $\lambda_1(-\nabla f(\overline{\boldsymbol{W}}_{r-1}))$ should be unique, i.e., $\lambda_1(-\nabla f(\overline{\boldsymbol{W}}_{r-1})) > \lambda_2(-\nabla f(\overline{\boldsymbol{W}}_{r-1}))$. This enables us to apply Theorem I.2 and deduce that the limiting trajectory

$$\boldsymbol{W}_r^{\mathrm{G}}(t) := \lim_{\epsilon\to 0} \phi\left(\overline{\boldsymbol{W}}_{r-1} + \epsilon\boldsymbol{u}_r\boldsymbol{u}_r^\top, \frac{1}{2\lambda_1(-\nabla f(\overline{\boldsymbol{W}}_{r-1}))} \log\frac{1}{\epsilon} + t\right)$$

exists, where $\boldsymbol{u}_r$ is the top eigenvector of $-\nabla f(\overline{\boldsymbol{W}}_{r-1})$. This $\boldsymbol{W}_r^{\mathrm{G}}(\cdot)$ is exactly the trajectory of GLRL in phase $r$ as $\epsilon \to 0$.

Note that $\boldsymbol{W}_r^{\mathrm{G}}(\cdot)$ corresponds to a trajectory of GF minimizing $\mathcal{L}(\cdot)$ in $\mathbb{R}^{d\times r}$, which should generically converge to a local minimizer of $\mathcal{L}(\cdot)$ in $\mathbb{R}^{d\times r}$. This means the limit $\overline{\boldsymbol{W}}_r := \lim_{t\to+\infty} \boldsymbol{W}_r^{\mathrm{G}}(t)$ should generically be a local minimizer of $f(\cdot)$ in $\mathbb{S}_{d, \leq r}^+$. If $\overline{\boldsymbol{W}}_r$ is further a minimizer in $\mathbb{S}_d^+$, then $\lambda_1(-\nabla f(\overline{\boldsymbol{W}}_r)) \leq 0$ and GLRL exits with $\overline{\boldsymbol{W}}_r$; otherwise GLRL enters phase $r + 1$.

If GF aligns well with GLRL in the beginning of phase $r$ (defined below), then by Theorem I.2, as $\alpha \to 0$, the minimum distance from GF to $\boldsymbol{W}_r^{\mathrm{G}}(t)$ converges to 0 for every $t \in \mathbb{R}$. Therefore, GF can get arbitrarily close to the $r$-th critical point $\overline{\boldsymbol{W}}_r$ of GLRL, i.e., there exists a suitable choice $T_\alpha^{(r)}$ so that $\lim_{\alpha\to 0} \phi(\boldsymbol{W}_\alpha, T_\alpha^{(r)}) = \overline{\boldsymbol{W}}_r$. Note that $\langle \overline{\boldsymbol{W}}_r, \boldsymbol{u}_r\boldsymbol{u}_r^\top \rangle = 0$ by (27) and thus

$\liminf_{\alpha \to 0} \left\langle \frac{\phi(\boldsymbol{W}_\alpha, T_\alpha^{(r)}) - \overline{\boldsymbol{W}}_r}{\|\phi(\boldsymbol{W}_\alpha, T_\alpha^{(r)}) - \overline{\boldsymbol{W}}_r\|_{\mathrm{F}}}, \boldsymbol{u}_r \boldsymbol{u}_r^\top \right\rangle = \liminf_{\alpha \to 0} \frac{\langle \phi(\boldsymbol{W}_\alpha, T_\alpha^{(r)}), \boldsymbol{u}_r \boldsymbol{u}_r^\top \rangle}{\|\phi(\boldsymbol{W}_\alpha, T_\alpha^{(r)}) - \overline{\boldsymbol{W}}_r\|_{\mathrm{F}}} \geq 0$. Generically, there should exist a suitable choice of $T_\alpha^{(r)}$ so that $\phi(\boldsymbol{W}_\alpha, T_\alpha^{(r)})$ not only converges to $\overline{\boldsymbol{W}}_r$ but also has positive alignment with $\boldsymbol{u}_r \boldsymbol{u}_r^\top$, that is, GF should generically align well with GLRL in the beginning of phase $r + 1$.

**Definition J.1.** We say that GF aligns well with GLRL in the beginning of phase $r$ if there exists $T_\alpha^{(r)}$ for every $\alpha > 0$ such that $\phi(\boldsymbol{W}_\alpha, T_\alpha^{(r)})$ converges to $\overline{\boldsymbol{W}}_{r-1}$ with positive alignment with $\boldsymbol{u}_r \boldsymbol{u}_r^\top$ as $\alpha \to 0$.

If the initialization satisfies that $\boldsymbol{W}_\alpha$ converges to $\boldsymbol{0}$ with positive alignment with $\boldsymbol{u}_1 \boldsymbol{u}_1^\top$ as $\alpha \to 0$, then GF aligns well with GLRL in the beginning of phase 1, which can be seen by taking $T_\alpha^{(1)} = 0$. Now assume that GF aligns well with GLRL in the beginning of phase $r - 1$, then the above argument shows that GF should generically align well with GLRL in the beginning of phase $r$, if GLRL does not exit in phase $r - 1$. In the other case, we can use a similar argument as in Theorem 5.8 to show that GF converges to a solution near the minimizer $\overline{\boldsymbol{W}}_r$ of $f(\cdot)$ as $t \to \infty$, and the distance between the solution and $\overline{\boldsymbol{W}}_r$ converges to 0 as $\alpha \to 0$. By this induction we prove that GF with infinitesimal initialization is equivalent to GLRL generically.

# K  PROOFS FOR DEEP MATRIX FACTORIZATION

## K.1  PRELIMINARY LEMMAS

**Lemma K.1.** If $\boldsymbol{W}(0) \succeq \boldsymbol{0}$, then $\boldsymbol{W}(t) \succeq \boldsymbol{0}$ and $\mathrm{rank}(\boldsymbol{W}(t)) = \mathrm{rank}(\boldsymbol{W}(0))$ for all $t$.

*Proof.* Note that we can always find a set of balanced $\boldsymbol{U}_i(t)$, such that $\boldsymbol{U}_1(t) \ldots \boldsymbol{U}_L(t) = \boldsymbol{W}(t)$, $d_2 = d_3 = \cdots = d_L = \mathrm{rank}(\boldsymbol{W}(t))$ and write the dynamics of $\boldsymbol{W}(t)$ in the space of $\{\boldsymbol{U}_i\}_{i=1}^L$. Thus it is clear that for all $t'$, $\mathrm{rank}(\boldsymbol{W}(t')) \leq \mathrm{rank}(\boldsymbol{W}(t))$. We can apply the same argument for $t'$ and we know $\mathrm{rank}(\boldsymbol{W}(t)) \leq \mathrm{rank}(\boldsymbol{W}(t'))$. Thus $\mathrm{rank}(\boldsymbol{W}(t))$ is constant over time, and we denote it by $k$. Since eigenvalues are continuous matrix functions, and $\forall t, \lambda_i(\boldsymbol{W}(t)), i \in [k] \neq 0$. Thus they cannot change their signs and it must hold that $\boldsymbol{W}(t) \succeq \boldsymbol{0}$. $\square$

**Lemma K.2.** $\forall a, b, P \in \mathbb{R}$, if $a > b \geq 0, P \geq 1$, then $\frac{a^P - b^P}{a - b} \leq P a^{P-1}$.

*Proof.* Let $f(x) = P(1 - x) - (1 - x^P)$. Since $f'(x) = -P + Px^{P-1} < 0$ for all $x \in [0, 1)$, $f(x) \geq f(0) = 0$. Then substituting $x$ by $\frac{b}{a}$ completes the proof. $\square$

Recall we use $D\boldsymbol{F}(\boldsymbol{N})[\boldsymbol{M}]$ to denote the directional derivative along $\boldsymbol{M}$ of $\boldsymbol{F}$ at $\boldsymbol{N}$.

**Lemma K.3.** Let $\boldsymbol{F} : \mathbb{S}_d^+ \to \mathbb{S}_d^+, \boldsymbol{M} \mapsto \boldsymbol{M}^P$, where $P \geq 1$ and $P \in \mathbb{Q}$. Then $\forall \boldsymbol{M}, \boldsymbol{N} \succeq \boldsymbol{0}$,

$$\|D\boldsymbol{F}(\boldsymbol{N})[\boldsymbol{M}]\|_{\mathrm{F}} \leq P \|\boldsymbol{N}\|_2^{P-1} \|\boldsymbol{M}\|_{\mathrm{F}},$$

where $D\boldsymbol{F}(\boldsymbol{N})[\boldsymbol{M}] := \lim_{t \to 0} \frac{\boldsymbol{F}(\boldsymbol{N} + t\boldsymbol{M}) - \boldsymbol{F}(\boldsymbol{N})}{t}$ is the directional derivative of $\boldsymbol{F}$ along $\boldsymbol{M}$.

*Proof.* Let $\boldsymbol{N} = \boldsymbol{U}\boldsymbol{\Sigma}\boldsymbol{U}^\top$, where $\boldsymbol{U}\boldsymbol{U}^\top = \boldsymbol{I}$ and $\boldsymbol{\Sigma} = \mathrm{diag}(\sigma_1, \cdots, \sigma_d)$. Note that $\boldsymbol{F}(\boldsymbol{U}\boldsymbol{M}\boldsymbol{U}^\top) = \boldsymbol{U}\boldsymbol{F}(\boldsymbol{M})\boldsymbol{U}^\top$ for any $\boldsymbol{M} \in \mathbb{S}_d^+$. Then we have

$$\begin{aligned}
\|D\boldsymbol{F}(\boldsymbol{N})[\boldsymbol{M}]\|_{\mathrm{F}} &= \lim_{t \to 0} \frac{\|\boldsymbol{F}(\boldsymbol{N} + t\boldsymbol{M}) - \boldsymbol{F}(\boldsymbol{N})\|_{\mathrm{F}}}{t} \\
&= \lim_{t \to 0} \frac{\|\boldsymbol{F}(\boldsymbol{\Sigma} + t\boldsymbol{U}^\top \boldsymbol{M}\boldsymbol{U}) - \boldsymbol{F}(\boldsymbol{\Sigma})\|_{\mathrm{F}}}{t} \\
&= \|D\boldsymbol{F}(\boldsymbol{\Sigma})[\boldsymbol{U}^\top \boldsymbol{M}\boldsymbol{U}]\|_{\mathrm{F}}.
\end{aligned}$$

Therefore, it suffices to prove the lemma for the case where $\boldsymbol{N}$ is diagonal, i.e., $\boldsymbol{N} = \boldsymbol{\Sigma}$.

Assume $P = \frac{q}{p}$, where $p, q \in \mathbb{N}$ and $q \geq p > 0$. Define $\boldsymbol{G}(\boldsymbol{N}) = \boldsymbol{N}^{\frac{1}{p}}$. Then $\boldsymbol{G}(\boldsymbol{\Sigma})^p = \boldsymbol{\Sigma}$. Taking directional derivative on both sides along direction $\boldsymbol{M}$, we have

$$\sum_{i=1}^{p} \boldsymbol{G}(\boldsymbol{\Sigma})^{i-1} D\boldsymbol{G}(\boldsymbol{\Sigma})[\boldsymbol{M}]\boldsymbol{G}(\boldsymbol{\Sigma})^{p-1} = \boldsymbol{M},$$

So we have

$$[D\boldsymbol{G}(\boldsymbol{\Sigma})[\boldsymbol{M}]]_{ij} = \frac{m_{ij}}{\sum_{k=1}^{p} \sigma_i^{\frac{k-1}{p}} \sigma_j^{\frac{p-k}{p}}}.$$

Let $\boldsymbol{H}(\boldsymbol{G}) = \boldsymbol{G}^q$. With the same argument, we know

$$[D\boldsymbol{H}(\boldsymbol{G}(\boldsymbol{\Sigma}))[\boldsymbol{M}]]_{ij} = m_{ij} \sum_{k=1}^{q} \sigma_i^{\frac{k-1}{p}} \sigma_j^{\frac{q-k}{p}}.$$

Note that $\boldsymbol{H}(\boldsymbol{G}(\boldsymbol{\Sigma})) = \boldsymbol{F}(\boldsymbol{\Sigma})$. By chain rule, we have
$$D\boldsymbol{F}(\boldsymbol{\Sigma})[\boldsymbol{M}] = D\boldsymbol{H}(\boldsymbol{G}(\boldsymbol{\Sigma}))[D\boldsymbol{G}(\boldsymbol{\Sigma})[\boldsymbol{M}]].$$

That is,

$$[D\boldsymbol{F}(\boldsymbol{\Sigma})[\boldsymbol{M}]]_{ij} = m_{ij} \frac{\sum_{k=1}^{q} \sigma_i^{\frac{k-1}{p}} \sigma_j^{\frac{q-k}{p}}}{\sum_{k=1}^{p} \sigma_i^{\frac{k-1}{p}} \sigma_j^{\frac{p-k}{p}}}.$$

When $\sigma_i = \sigma_j$, clearly $[D\boldsymbol{F}(\boldsymbol{\Sigma})[\boldsymbol{M}]]_{ij} = m_{ij} \cdot \frac{q}{p} \cdot \sigma_i^{\frac{q-p}{p}} = P m_{ij} \sigma_i^{P-1}$. Otherwise, we assume WLOG that $\sigma_i > \sigma_j$, we multiply $\sigma_i - \sigma_j$ to both numerator and denominator and we have

$$|[D\boldsymbol{F}(\boldsymbol{\Sigma})[\boldsymbol{M}]]_{ij}| = |m_{ij}| \frac{\sigma_i^P - \sigma_j^P}{\sigma_i - \sigma_j} \leq |m_{ij}| P \sigma_i^{P-1} \leq |m_{ij}| P \|\boldsymbol{\Sigma}\|_2^{P-1}.$$

where the first inequality is by Lemma K.2. Thus we conclude the proof. $\qquad \square$

**Lemma K.4.** *For any* $\boldsymbol{A}, \boldsymbol{B} \succeq \boldsymbol{0}$ *and* $P \in \mathbb{R}, P \geq 1$,
$$\left\|\boldsymbol{A}^P - \boldsymbol{B}^P\right\|_{\mathrm{F}} \leq P \|\boldsymbol{A} - \boldsymbol{B}\|_{\mathrm{F}} \max\left\{ \|\boldsymbol{A}\|_2^{P-1}, \|\boldsymbol{B}\|_2^{P-1} \right\}.$$

*Proof.* Since both sides are continuous in $P$ and $\mathbb{Q}$ is dense in $\mathbb{R}$, it suffices to prove the lemma for $P \in \mathbb{Q}$. Let $\rho := \max\left\{ \|\boldsymbol{A}\|_2, \|\boldsymbol{B}\|_2 \right\}$ and $\boldsymbol{F}(\boldsymbol{M}) = \boldsymbol{M}^P$. Define $\boldsymbol{N} : [0, 1] \rightarrow \mathbb{S}_d^+, \boldsymbol{N}(t) = (1-t)\boldsymbol{A} + t\boldsymbol{B}$, we have

1. $\|\boldsymbol{N}(t)\|_2 \leq \rho$, since $\|\cdot\|_2$ is convex.

2. $\|D\boldsymbol{F}(\boldsymbol{N}(t))[\boldsymbol{B} - \boldsymbol{A}]\|_{\mathrm{F}} \leq P \|\boldsymbol{N}(t)\|_2^{P-1} \|\boldsymbol{B} - \boldsymbol{A}\|_{\mathrm{F}}$ by Lemma K.3.

Therefore,

$$\begin{aligned}
\|\boldsymbol{F}(\boldsymbol{N}(1)) - \boldsymbol{F}(\boldsymbol{N}(0))\|_{\mathrm{F}} &\leq \int_0^1 \left\| \frac{\mathrm{d}\boldsymbol{F}(\boldsymbol{N}(t))}{\mathrm{d}t} \right\|_{\mathrm{F}} \mathrm{d}t \\
&= \int_{t=0}^1 \|D\boldsymbol{F}(\boldsymbol{N}(t))[\boldsymbol{B} - \boldsymbol{A}]\|_{\mathrm{F}} \, \mathrm{d}t \\
&\leq P \|\boldsymbol{A} - \boldsymbol{B}\|_{\mathrm{F}} \rho^{P-1},
\end{aligned}$$

which completes the proof. $\qquad \square$

For a locally Lipschitz function $f(\,\cdot\,)$, the Clarke subdifferential (Clarke, 1975; 1990; Clarke et al., 2008) of $f$ at any point $\boldsymbol{x}$ is the following convex set

$$\frac{\partial^\circ f(\boldsymbol{x})}{\partial \boldsymbol{x}} := \mathrm{co}\left\{ \lim_{k \to \infty} \nabla f(\boldsymbol{x}_k) : \boldsymbol{x}_k \to \boldsymbol{x}, f \text{ is differentiable at } \boldsymbol{x}_k \right\},$$

where $\mathrm{co}$ denotes the convex hull.

Clarke subdifferential generalize the standard notion of gradients in the sense that, when $f$ is smooth, $\frac{\partial^\circ f(\boldsymbol{x})}{\partial \boldsymbol{x}} = \{\nabla f(\boldsymbol{x})\}$. Clarke subdifferential satisfies the chain rule:

**Theorem K.5** (Theorem 2.3.10, Clarke 1990). *Let $\boldsymbol{F} : \mathbb{R}^k \rightarrow \mathbb{R}^d$ be a differentiable function and $g : \mathbb{R}^d \rightarrow \mathbb{R}$ Lipschitz around $\boldsymbol{F}(\boldsymbol{x})$. Then $f = g \circ \boldsymbol{F}$ is Lipschitz around $\boldsymbol{x}$ and one has*

$$\frac{\partial^\circ f(\boldsymbol{x})}{\partial \boldsymbol{x}} \subseteq \frac{\partial^\circ g(\boldsymbol{F}(\boldsymbol{x}))}{\partial \boldsymbol{F}} \circ \frac{\mathrm{d}\boldsymbol{F}(\boldsymbol{x})}{\mathrm{d}\boldsymbol{x}}.$$

Let $\lambda_m : \mathbb{S}_d \rightarrow \mathbb{R}, \boldsymbol{M} \mapsto \lambda_m(\boldsymbol{M})$ be the $m$-th largest eigenvalue of a symmetric matrix $\boldsymbol{M}$. The following theorem gives the Clarke's subdifferentials of the eigenvalue:

**Theorem K.6** (Theorem 5.3, Hiriart-Urruty and Lewis 1999). *The Clarke subdifferential of the eigenvalue function $\lambda_m$ is given below, where* co *denotes the convex hull:*

$$\frac{\partial^\circ \lambda_m(\boldsymbol{M})}{\partial \boldsymbol{M}} = \mathrm{co}\{\boldsymbol{v}\boldsymbol{v}^\top : \boldsymbol{M}\boldsymbol{v} = \lambda_m(\boldsymbol{M})\boldsymbol{v}, \|\boldsymbol{v}\|_2 = 1\}.$$

### K.2   PROOF OF LEMMA 6.1

The equation to be proved is:

$$\frac{\mathrm{d}\boldsymbol{M}}{\mathrm{d}t} = -\nabla f(\boldsymbol{M}^{L/2})\boldsymbol{M}^{L/2} - \boldsymbol{M}^{L/2}\nabla f(\boldsymbol{M}^{L/2}). \tag{29}$$

Since $\boldsymbol{W}(t) \succeq \boldsymbol{0}$ by Lemma K.1, (11) can be rewritten as the following:

$$\frac{\mathrm{d}\boldsymbol{W}}{\mathrm{d}t} = -\sum_{i=0}^{L-1} \boldsymbol{W}^{\frac{2i}{L}}\nabla f(\boldsymbol{W})\boldsymbol{W}^{2-\frac{2i+2}{L}}. \tag{30}$$

*Proof for Lemma 6.1.* Suppose $\boldsymbol{W}(t)$ is a symmetric solution of (11). By Lemma K.1, we know $\boldsymbol{W}(t)$ also satisfies (30). Now we let $\boldsymbol{R}(t)$ be the solution of the following ODE with $\boldsymbol{R}(0) := (\boldsymbol{W}(0))^{\frac{1}{L}}$. Note we don't define $\boldsymbol{R}(t)$ by $(\boldsymbol{W}(t))^{\frac{1}{L}}$.

$$\frac{\mathrm{d}\boldsymbol{R}}{\mathrm{d}t} = -\sum_{i=0}^{L-1} (-1)^i \boldsymbol{R}^i \nabla f(\boldsymbol{R}^L)\boldsymbol{R}^{L-1-i}. \tag{31}$$

The calculation below shows that $\boldsymbol{R}^L(t)$ also satisfies (30).

$$\frac{\mathrm{d}\boldsymbol{R}^L}{\mathrm{d}t} = \sum_{j=0}^{L-1} \boldsymbol{R}^j \frac{\mathrm{d}\boldsymbol{R}}{\mathrm{d}t}\boldsymbol{R}^{L-1-j} = \sum_{j=0}^{L-1}\sum_{i=0}^{L-1} (-1)^i \boldsymbol{R}^{i+j}\nabla f(\boldsymbol{R}^L)\boldsymbol{R}^{2L-2-i-j}$$

$$= \sum_{i=0}^{2L-2} \left(\sum_{j=0}^{i}(-1)^j\right) \boldsymbol{R}^i \nabla f(\boldsymbol{R}^L)\boldsymbol{R}^{2T-2-i}$$

$$= \sum_{i=0}^{L-1} (\boldsymbol{R}^L)^{\frac{2i}{L}}\nabla f(\boldsymbol{R}^L)(\boldsymbol{R}^L)^{2-\frac{2+2i}{L}}.$$

Since $\boldsymbol{R}^L(0) = \boldsymbol{W}(0)$, by existence and uniqueness theorem, $\boldsymbol{R}^L(t) = \boldsymbol{W}(t), \forall t \in \mathbb{R}$. So

$$\frac{\mathrm{d}\boldsymbol{M}}{\mathrm{d}t} = \boldsymbol{R}\frac{\mathrm{d}\boldsymbol{R}}{\mathrm{d}t} + \frac{\mathrm{d}\boldsymbol{R}}{\mathrm{d}t}\boldsymbol{R} = -\nabla f(\boldsymbol{M}^{L/2})\boldsymbol{M}^{L/2} - \boldsymbol{M}^{L/2}\nabla f(\boldsymbol{M}^{L/2}),$$

which completes the proof. $\qquad\square$

### K.3   PROOF FOR THEOREM 6.2

Now we turn to prove Theorem 6.2. Let $P = L/2$. Then (29) can be rewritten as

$$\frac{d\boldsymbol{M}}{dt} = -\left(\nabla f(\boldsymbol{M}^P)\boldsymbol{M}^P + \boldsymbol{M}^P\nabla f(\boldsymbol{M}^P)\right). \tag{32}$$

The following lemma about the growth rate of $\lambda_k(\boldsymbol{M})$ is used later in the proof.

**Lemma K.7.** *Suppose $\boldsymbol{M}(t)$ satisfies (32), we have for any $T' > T$, and $k \in [d]$,*

$$\lambda_k(\boldsymbol{M}(T')) - \lambda_k(\boldsymbol{M}(T)) \le \int_T^{T'} 2\lambda_k(\boldsymbol{M}(t))^P \|\nabla f(\boldsymbol{M}^P(t))\|_2 \mathrm{d}t. \tag{33}$$

*and*

$$\frac{1}{P-1}\left(\lambda_k^{1-P}(\boldsymbol{M}(T)) - \lambda_k^{1-P}(\boldsymbol{M}(T'))\right) \le \int_T^{T'} 2\|\nabla f(\boldsymbol{M}^P(t))\|_2 \mathrm{d}t. \tag{34}$$

*Proof.* Since $\lambda_k(\boldsymbol{M}(t))$ is locally Lipschitz in $t$, by Rademacher's theorem, we know $\lambda_k(\boldsymbol{M}(t))$ is differentiable almost everywhere, and the following holds

$$\lambda_k(\boldsymbol{M}(T')) - \lambda_k(\boldsymbol{M}(T)) = \int_T^{T'} \frac{\mathrm{d}\lambda_k(\boldsymbol{M}(t))}{\mathrm{d}t}\mathrm{d}t.$$

When $\frac{\mathrm{d}\lambda_k(\boldsymbol{M}(t))}{\mathrm{d}t}$ exists, we have

$$\begin{aligned}
\frac{\mathrm{d}\lambda_k(\boldsymbol{M}(t))}{\mathrm{d}t} &\in \left\{ \left\langle \boldsymbol{G}, \frac{\mathrm{d}\boldsymbol{M}(t)}{\mathrm{d}t} \right\rangle : \boldsymbol{G} \in \frac{\partial^\circ \lambda_k(\boldsymbol{M})}{\partial \boldsymbol{M}} \right\} \\
&= \left\{ 2\lambda_k(\boldsymbol{M}^P(t)) \left\langle \boldsymbol{G}, -\nabla f(\boldsymbol{M}^P(t)) \right\rangle : \boldsymbol{G} \in \frac{\partial^\circ \lambda_k(\boldsymbol{M})}{\partial \boldsymbol{M}} \right\}
\end{aligned}$$

Note that $\|\boldsymbol{G}\|_{\mathrm{F}} \le \|\boldsymbol{G}\|_* = 1$. So $\left|\left\langle \boldsymbol{G}, -\nabla f(\boldsymbol{M}^P(t)) \right\rangle\right| \le \|\nabla f(\boldsymbol{M}^P(t))\|_2$. We can prove (34) with a similar argument. □

To prove Theorem 6.2, it suffices to consider the case that $\boldsymbol{M}(0) = \hat{\alpha}\boldsymbol{I}$ where $\hat{\alpha} := \alpha^{1/P}$. WLOG we can assume $-\nabla f(\boldsymbol{0}) = \mathrm{diag}(\mu_1, \dots, \mu_d)$ by choosing a suitable standard basis. By assumption in Theorem 6.2, we have $\mu_1 > \max\{\mu_2, 0\}$ and $\mu_1 = \|\nabla f(\boldsymbol{0})\|_2$. We use $\phi_m(\boldsymbol{M}_0, t)$ to denote the solution of $\boldsymbol{M}(t)$ when $\boldsymbol{M}(0) = \boldsymbol{M}_0$.

Let $R > 0$. Since $f(\cdot)$ is $\mathcal{C}^3$-smooth, there exists $\beta > 0$ such that

$$\|\nabla f(\boldsymbol{W}_1) - \nabla f(\boldsymbol{W}_2)\|_{\mathrm{F}} \le \beta \|\boldsymbol{W}_1 - \boldsymbol{W}_2\|_2$$

for all $\boldsymbol{W}_1, \boldsymbol{W}_2$ with $\|\boldsymbol{W}_1\|_2, \|\boldsymbol{W}_2\|_2 \le R$.

Let $\kappa = \beta/\mu_1$. We assume WLOG that $R \le \frac{1}{\kappa(P-1)}$. Let $F_{\hat{\alpha}}(x) := \int_{x^{-(P-1)}}^{\hat{\alpha}^{-(P-1)}} \frac{dz}{1+\kappa z^{-P/(P-1)}}$. Then $F_{\hat{\alpha}}'(x) = \frac{(P-1)x^{-P}}{1+\kappa x^P} = \frac{P-1}{(1+\kappa x^P)x^P}$. We will use this function to bound norm growth. Let $g_{\hat{\alpha},c}(t) = \frac{1}{\hat{\alpha}^{-(P-1)} - \kappa(P-1)c - 2\mu_1(P-1)t}$. Define $T_{\hat{\alpha}}(r) = \frac{\hat{\alpha}^{-(P-1)} - \kappa(P-1)r - r^{-(P-1)}}{2\mu_1(P-1)}$. It is easy to verify that $g_{\hat{\alpha},r}(T_{\hat{\alpha}}(r)) = r^{P-1}$.

**Lemma K.8.** *For any $x \in [\hat{\alpha}, R]$ we have*

$$\left(\hat{\alpha}^{-(P-1)} - x^{-(P-1)}\right) - F_{\hat{\alpha}}(x) \in [0, \kappa(P-1)x].$$

*Proof.* On the one hand, we have

$$\hat{\alpha}^{-(P-1)} - x^{-(P-1)} - F_{\hat{\alpha}}(x) = \int_{x^{-(P-1)}}^{\hat{\alpha}^{-(P-1)}} \left(1 - \frac{1}{1+\kappa z^{-P/(P-1)}}\right) dz \ge 0.$$

On the other hand,

$$\begin{aligned}
\hat{\alpha}^{-(P-1)} - x^{-(P-1)} - F_{\hat{\alpha}}(x) &= \int_{x^{-(P-1)}}^{\hat{\alpha}^{-(P-1)}} \frac{\kappa}{z^{P/(P-1)} + \kappa} dz \le \kappa \int_{x^{-(P-1)}}^{\hat{\alpha}^{-(P-1)}} \frac{1}{z^{P/(P-1)}} dz \\
&= \kappa(P-1) \cdot \frac{-1}{z^{1/(P-1)}} \Bigg|_{x^{-(P-1)}}^{\hat{\alpha}^{-(P-1)}} \\
&\le \kappa(P-1)x,
\end{aligned}$$

which completes the proof. □

**Lemma K.9.** *Let $M_0$ be a PSD matrix with $\|M_0\|_2 \leq 1$. For $M(t) := \phi_m(\hat{\alpha}M_0, t)$ and $t \leq T_{\hat{\alpha}}(c)$,*

$$\|M(t)\|_2 = \lambda_1(M(t)) \leq g_{\hat{\alpha},c}(t)^{\frac{1}{P-1}}.$$

*Proof.* Since $\|\nabla f(M^P)\|_2 \leq \|\nabla f(0)\|_2 + \beta\|M\|_2^P \leq \mu_1 + \beta(\lambda_1(M))^P$, by Lemma K.7, we have

$$\lambda_1(M(t)) \leq \lambda_1(M(0)) + \int_0^t 2(\mu_1 + \beta(\lambda_1(M(\tau)))^P)(\lambda_1(M(\tau)))^P d\tau$$

$$= \hat{\alpha} + 2\mu_1(P-1)\int_0^t \frac{d\tau}{F'_{\hat{\alpha}}(\lambda_1(M(\tau)))}$$

So

$$F_{\hat{\alpha}}(\lambda_1(M(t))) \leq 2\mu_1(P-1)t.$$

If $\|M(t)\|_2 < \hat{\alpha}$, then $\|M(t)\|_2 \leq g_{\hat{\alpha},c}(t)^{\frac{1}{P-1}}$. If $\|M(t)\|_2 \geq \hat{\alpha}$, then by Lemma K.8,

$$F_{\hat{\alpha}}(\|M(t)\|_2) \leq 2\mu_1(P-1)T_{\hat{\alpha}}(c) = \hat{\alpha}^{-(P-1)} - \kappa(P-1)c - c^{-(P-1)} \leq F_{\hat{\alpha}}(c),$$

so $\|M(t)\|_2 \leq c$ for all $t \leq T_{\hat{\alpha}}(c)$. Applying Lemma K.8 again, we have

$$\hat{\alpha}^{-(P-1)} - \|M(t)\|_2^{-(P-1)} \leq F(\|M(t)\|_2) + \kappa(P-1)c \leq 2\mu_1(P-1)t + \kappa(P-1)c,$$

which implies $\|M(t)\|_2 \leq g_{\hat{\alpha},c}(t)^{\frac{1}{P-1}}$ by definition. $\qquad\square$

Consider the following ODE:

$$\frac{d\widehat{M}}{dt} = -\left(\nabla f(0)\widehat{M}^P + \widehat{M}^P\nabla f(0)\right).$$

We use $\hat{\phi}_m(\widehat{M}_0, t)$ to denote the solution of $\widehat{M}(t)$ when $\widehat{M}(0) = \widehat{M}_0$. For diagonal matrix $\widehat{M}_0$, $\widehat{M}(t)$ is also diagonal for any $t$, and it is easy to show that

$$e_i^\top \widehat{M}(t)e_i = \begin{cases} \left(\frac{1}{(\hat{\alpha}e_i^\top \widehat{M}_0 e_i)^{-(P-1)} - 2\mu_i(P-1)t}\right)^{\frac{1}{P-1}} & e_i^\top \widehat{M}_0 e_i \neq 0, \\ 0 & e_i^\top \widehat{M}_0 e_i = 0. \end{cases} \tag{35}$$

**Remark K.10.** *Unlike depth-2 case, the closed form solution, $\widehat{M}(t)$ is only tractable for diagonal initialization, i.e., (35) (note that the identity matrix is diagonal). And this is the main barrier for extending our two-phase analysis to the case of general initialization when $L \geq 3$. In Appendix L, we give a more detailed discussion on this barrier.*

The following lemma shows that the trajectory of $M(t)$ is close to $\widehat{M}(t)$.

**Lemma K.11.** *Let $M_0$ be a diagonal PSD matrix with $\|M_0\|_2 \leq 1$. For $M(t) := \phi_m(\hat{\alpha}M_0, t)$ and $\widehat{M}(t) := \hat{\phi}_m(\hat{\alpha}M_0, t)$, we have*

$$\|M(T_{\hat{\alpha}}(r)) - \widehat{M}(T_{\hat{\alpha}}(r))\|_F = O(r^{P+1}).$$

*Proof.* We bound the difference $D := M - \widehat{M}$ between $M$ and $\widehat{M}$.

$$\left\|\frac{dD}{dt}\right\|_F = 2\left\|\nabla f(0)\left(M^P - \widehat{M}^P\right) + \left(\nabla f(M^P) - \nabla f(0)\right)M^P\right\|_F$$

$$\leq 2\left(\|\nabla f(0)\|_2\|M^P - \widehat{M}^P\|_F + \|\nabla f(M^P) - \nabla f(0)\|_F\|M^P\|_2\right)$$

$$\leq 2\left(\mu_1 P \max\{\|M\|_2^{P-1}, \|\widehat{M}\|_2^{P-1}\}\|D\|_F + \beta\|M\|_2^{2P}\right),$$

where the last step is by Lemma K.4. This implies that

$$\|D(t)\|_F \leq \int_{\tau=0}^t \left\|\frac{dD(\tau)}{d\tau}\right\|_F d\tau \leq \int_0^t 2\left(\mu_1 P g_{\hat{\alpha},r}(\tau)\|D(\tau)\|_F + \beta g_{\hat{\alpha},r}(\tau)^{\frac{2P}{P-1}}\right)d\tau.$$

So

$$
\begin{aligned}
\|\boldsymbol{D}(T_{\hat{\alpha}}(r))\|_{\mathrm{F}} &\le \int_0^{T_{\hat{\alpha}}(r)} 2\beta g_{\hat{\alpha},r}(t)^{\frac{2P}{P-1}} \exp\left(2\mu_1 P \int_t^{T_{\hat{\alpha}}(r)} g_{\hat{\alpha},r}(\tau)d\tau\right) dt \\
&= \int_0^{T_{\hat{\alpha}}(r)} 2\beta g_{\hat{\alpha},r}(t)^{\frac{2P}{P-1}} \exp\left(\frac{P}{P-1}\ln\frac{g_{\hat{\alpha},r}(T_{\hat{\alpha}}(r))}{g_{\hat{\alpha},r}(t)}\right) dt \\
&= \int_0^{T_{\hat{\alpha}}(r)} 2\beta g_{\hat{\alpha},r}(t)^{\frac{P}{P-1}} g_{\hat{\alpha},r}(T_{\hat{\alpha}}(r))^{\frac{P}{P-1}} dt \\
&= 2\beta \cdot \frac{1}{2\mu_1} g_{\hat{\alpha},r}(T_{\hat{\alpha}}(r))^{\frac{1}{P-1}} \cdot g_{\hat{\alpha},r}(T_{\hat{\alpha}}(r))^{\frac{P}{P-1}} \\
&= \kappa g_{\hat{\alpha},r}(T_{\hat{\alpha}}(r))^{\frac{P+1}{P-1}} \\
&= \kappa r^{P+1}.
\end{aligned}
$$

which proves the bound. $\qquad\square$

**Lemma K.12.** *Let* $\boldsymbol{M}(t) = \phi_m(\hat{\alpha}\boldsymbol{M}_0, t)$, $\widetilde{\boldsymbol{M}}(t) = \phi_m(\hat{\alpha}\widetilde{\boldsymbol{M}}_0, t)$. *If* $\max\{\|\boldsymbol{M}_0\|_2, \|\widetilde{\boldsymbol{M}}_0\|_2\} \le 1$. *For* $t \le T_{\hat{\alpha}}(r)$, *we have*

$$
\|\boldsymbol{M}(t) - \widetilde{\boldsymbol{M}}(t)\|_{\mathrm{F}} \le \left(\frac{r}{\hat{\alpha}}\right)^P e^{2\kappa r^P} \|\boldsymbol{M}(0) - \widetilde{\boldsymbol{M}}(0)\|_{\mathrm{F}}.
$$

*Proof.* Define $\boldsymbol{D}(t) = \boldsymbol{M}(t) - \widetilde{\boldsymbol{M}}(t)$. Then we have

$$
\begin{aligned}
\left\|\frac{d\boldsymbol{D}}{dt}\right\|_{\mathrm{F}} &= 2\left\|\left(\nabla f(\boldsymbol{M}^P)\left(\boldsymbol{M}^P - \widetilde{\boldsymbol{M}}^P\right) + \left(\nabla f(\boldsymbol{M}^P) - \nabla f(\widetilde{\boldsymbol{M}}^P)\right)\widetilde{\boldsymbol{M}}^P\right)\right\|_{\mathrm{F}} \\
&\le 2\left(\|\nabla f(\boldsymbol{M}^P)\|_2\|\boldsymbol{M}^P - \widetilde{\boldsymbol{M}}^P\|_{\mathrm{F}} + \beta\|\boldsymbol{M}^P - \widetilde{\boldsymbol{M}}^P\|_{\mathrm{F}}\|\widetilde{\boldsymbol{M}}^P\|_2\right) \\
&\le 2\left(\mu_1 + \beta\|\widetilde{\boldsymbol{M}}\|_2^P + \beta\|\boldsymbol{M}\|_2^P\right)P\max\{\|\boldsymbol{M}\|_2^{P-1}, \|\widetilde{\boldsymbol{M}}\|_2^{P-1}\}\|\boldsymbol{D}\|_{\mathrm{F}},
\end{aligned}
$$

where the last step is by Lemma K.4. So

$$
\begin{aligned}
\|\boldsymbol{D}(T_{\hat{\alpha}}(r))\|_{\mathrm{F}} &\le \|\boldsymbol{D}(0)\|_{\mathrm{F}} \cdot \exp\left(2P\mu_1 \int_0^{T_{\hat{\alpha}}(r)}\left(1 + 2\kappa g_{\hat{\alpha},r}(t)^{\frac{P}{P-1}}\right)g_{\hat{\alpha},r}(t)dt\right) \\
&\le \|\boldsymbol{D}(0)\|_{\mathrm{F}} \cdot \exp\left(\frac{P}{P-1}\ln\frac{g_{\hat{\alpha},r}(T_{\hat{\alpha}}(r))}{g_{\hat{\alpha},r}(0)} + 2\kappa g_{\hat{\alpha},r}(T_{\hat{\alpha}}(r))^{\frac{P}{P-1}}\right) \\
&\le \|\boldsymbol{D}(0)\|_{\mathrm{F}}\left(\frac{r}{\hat{\alpha}}\right)^P e^{2\kappa r^P},
\end{aligned}
$$

which proves the bound. $\qquad\square$

Let $\boldsymbol{M}_\alpha^{\mathrm{G}}(t) := \phi_m\left(\alpha\boldsymbol{e}_1\boldsymbol{e}_1^\top, \frac{\hat{\alpha}^{-(P-1)}}{2\mu_1(P-1)} + t\right)$. Let $\overline{\boldsymbol{M}}(t) := \lim_{\alpha\to 0}\boldsymbol{M}_\alpha^{\mathrm{G}}(t)$.

**Lemma K.13.** *For every* $t \in (-\infty, +\infty)$, $\overline{\boldsymbol{M}}(t)$ *exists and* $\boldsymbol{M}_{\hat{\alpha}}^{\mathrm{G}}(t)$ *converges to* $\overline{\boldsymbol{M}}(t)$ *in the following rate:*

$$
\left\|\boldsymbol{M}_{\hat{\alpha}}^{\mathrm{G}}(t) - \overline{\boldsymbol{M}}(t)\right\|_{\mathrm{F}} = O(\hat{\alpha}).
$$

*Proof.* Let $c$ be a sufficiently small constant. Let $\bar{T} := \frac{-\kappa(P-1)c - c^{-(P-1)}}{2\mu_1(P-1)}$. We prove this lemma in the cases of $t \in (-\infty, \bar{T}]$ and $t > \bar{T}$ respectively.

**Case 1.** Fix $t \in (-\infty, \bar{T}]$. Then $\frac{\hat{\alpha}^{-(P-1)}}{2\mu_1(P-1)} + t \le T_{\hat{\alpha}}(c)$. Let $\tilde{\alpha}$ be the unique number such that $\kappa(P-1)\tilde{\alpha} + \tilde{\alpha}^{-(P-1)} = \hat{\alpha}^{-(P-1)}$. Let $\hat{\alpha}' < \hat{\alpha}$ be an arbitrarily small number. Let $t_0 := T_{\hat{\alpha}'}(\tilde{\alpha}) = \frac{(\hat{\alpha}')^{-(P-1)} - \hat{\alpha}^{-(P-1)}}{2\mu_1(P-1)}$. By Lemma K.11 and (35), we have

$$
\left\|\phi_m(\hat{\alpha}'\boldsymbol{e}_1\boldsymbol{e}_1^\top, t_0) - \hat{\alpha}\boldsymbol{e}_1\boldsymbol{e}_1^\top\right\|_{\mathrm{F}} \le \left\|\phi_m(\hat{\alpha}'\boldsymbol{e}_1\boldsymbol{e}_1^\top, t_0) - \hat{\phi}_m(\hat{\alpha}'\boldsymbol{e}_1\boldsymbol{e}_1^\top, t_0)\right\|_{\mathrm{F}} \le O(\tilde{\alpha}^{P+1}).
$$

By Lemma K.9, $\|\phi_m(\hat{\alpha}'e_1e_1^\top, t_0)\|_2 \le \tilde{\alpha}$. Then by Lemma K.12, we have

$$\left\|\phi_m(\hat{\alpha}'e_1e_1^\top, t_0 + t) - \phi(\hat{\alpha}e_1e_1^\top, t)\right\|_F \le \left(\frac{c}{\tilde{\alpha}}\right)^P e^{2\kappa c^P} \cdot O(\tilde{\alpha}^{P+1}) = O(\tilde{\alpha}) = O(\hat{\alpha}).$$

This implies that $\{M_{\hat{\alpha}}^G(t)\}$ satisfies Cauchy's criterion for every $t$, and thus the limit $\overline{M}(t)$ exists for $t \le \bar{T}$. The convergence rate can be deduced by taking limits for $\hat{\alpha}' \to 0$ on both sides.

**Case 2.** For $t = \bar{T} + \tau$ with $\tau > 0$, $\phi_m(M, \tau)$ is locally Lipschitz with respect to $M$. So

$$\begin{aligned}
\left\|M_{\hat{\alpha}}^G(t) - M_{\hat{\alpha}'}^G(t)\right\|_F &= \left\|\phi_m(M_{\hat{\alpha}}^G(\bar{T}), \tau) - \phi_m(M_{\hat{\alpha}'}^G(\bar{T}), \tau)\right\|_F \\
&= O(\left\|M_{\hat{\alpha}}^G(\bar{T}) - M_{\hat{\alpha}'}^G(\bar{T})\right\|_F) \\
&= O(\hat{\alpha}),
\end{aligned}$$

which proves the lemma for $t > \bar{T}$. $\qquad\square$

**Theorem K.14.** *For every $t \in (-\infty, +\infty)$, as $\alpha \to 0$, we have:*

$$\left\|\phi_m\left(\hat{\alpha}I, \frac{\hat{\alpha}^{-(P-1)}}{2\mu_1(P-1)} + t\right) - \overline{M}(t)\right\|_F = O(\hat{\alpha}^{\frac{1}{P+1}}), \tag{36}$$

*and for any $2 \le k \le d$,*

$$\lambda_k\left(\phi_m\left(\hat{\alpha}I, \frac{\hat{\alpha}^{-(P-1)}}{2\mu_1(P-1)} + t\right)\right) = O(\hat{\alpha}). \tag{37}$$

*Proof.* Let $M_{\hat{\alpha}}(t) := \phi_m\left(\hat{\alpha}I, \frac{\hat{\alpha}^{-(P-1)}}{2\mu_1(P-1)} + t\right)$. Again we let $c$ be a sufficiently small constant and $\bar{T} := \frac{-\kappa(P-1)c - c^{-(P-1)}}{2\mu_1(P-1)}$. We prove in the cases of $t \in (-\infty, \bar{T}]$ and $t > \bar{T}$ respectively.

**Case 1.** Fix $t \in (-\infty, \bar{T}]$. Let $\hat{\alpha}_1 := \hat{\alpha}^{\frac{1}{P+1}}$. Let $\tilde{\alpha}_1$ be the unique number such that $\kappa(P-1)\tilde{\alpha}_1 + \tilde{\alpha}_1^{-(P-1)} = \hat{\alpha}_1^{-(P-1)}$. Let $t_0 := T_{\hat{\alpha}}(\tilde{\alpha}_1) = \frac{\hat{\alpha}^{-(P-1)} - \hat{\alpha}_1^{-(P-1)}}{2\mu_1(P-1)}$. Then

$$\begin{aligned}
\left\|\phi_m(\hat{\alpha}I, t_0) - \hat{\alpha}_1 e_1 e_1^\top\right\|_F &\le \left\|\phi_m(\hat{\alpha}I, t_0) - \hat{\phi}_m(\hat{\alpha}I, t_0)\right\|_F + \left\|\hat{\phi}_m(\hat{\alpha}I, t_0) - \hat{\alpha}_1 e_1 e_1^\top\right\|_F \\
&= O(\tilde{\alpha}_1^{P+1} + \hat{\alpha}) \\
&= O(\hat{\alpha}).
\end{aligned}$$

By Lemma K.9, $\|\phi_m(\hat{\alpha}'I, t_0)\|_2 \le \tilde{\alpha}_1$. Then by Lemma K.12, we have

$$\begin{aligned}
\left\|M_{\hat{\alpha}}(t) - M_{\hat{\alpha}_1}^G(t)\right\|_F &= \left\|\phi_m(\hat{\alpha}I, t_0 + t) - \phi_m(\hat{\alpha}_1 e_1 e_1^\top, t)\right\|_F \\
&\le \left(\frac{c}{\tilde{\alpha}_1}\right)^P e^{2\kappa c^P} \cdot O(\hat{\alpha}) = O(\hat{\alpha}^{\frac{1}{P+1}}).
\end{aligned}$$

Combining this with the convergence rate for $M_{\hat{\alpha}_1}^G(t)$ proves the bound (36).

For (37), by Lemma K.7, we have

$$\begin{aligned}
\lambda_k^{1-P}(M_{\hat{\alpha}}(\bar{T})) - \lambda_k^{1-P}(M_{\hat{\alpha}}(t_0)) &\le \int_{t_0}^{\bar{T}} 2(P-1)\left\|\nabla f(M_{\hat{\alpha}}^P(t))\right\|_2 dt \\
&\le \int_{t_0}^{\bar{T}} 2(P-1)(\mu_1 + \beta\|M_{\hat{\alpha}}(t)\|_2^P)dt \\
&\le -2(P-1)\left(\mu_1(t - T_1) + \frac{\kappa}{2} \cdot g_{\hat{\alpha},c}(t)^{\frac{1}{P-1}}\right).
\end{aligned} \tag{38}$$

By Lemma K.11, $\lambda_1(M_{\hat{\alpha}}(\bar{T})) = \left\|M_{\hat{\alpha}}(\bar{T})\right\|_2 = c + O(c^{P+1})$. For $k \ge 2$,

$$\begin{aligned}
\lambda_k(M_{\hat{\alpha}}(\bar{T}))^{-(P-1)} &\ge \Omega(\hat{\alpha}^{-(P-1)}) - 2(P-1)\left(\mu_1(\bar{T} - T_1) + \frac{\kappa}{2} \cdot c\right) \\
&\ge \Omega(\hat{\alpha}^{-(P-1)}) - \frac{\hat{\alpha}^{-\frac{P-1}{P+1}} - c^{-(P-1)}}{2\mu_1(P-1)} - O(c) \\
&\ge \Omega(\hat{\alpha}^{-(P-1)}).
\end{aligned}$$

Thus $\lambda_k(\boldsymbol{M}_{\hat{\alpha}}(\bar{T})) \le O(\hat{\alpha})$.

**Case 2.** For $t = \bar{T} + \tau$ with $\tau > 0$, $\phi_m(\boldsymbol{M}, \tau)$ is locally Lipschitz with respect to $\boldsymbol{M}$. So

$$\left\| \boldsymbol{M}_{\hat{\alpha}}(t) - \boldsymbol{M}_{\hat{\alpha}_1}^{\mathrm{G}}(t) \right\|_{\mathrm{F}} = \left\| \phi_m(\boldsymbol{M}_{\hat{\alpha}}(\bar{T}), \tau) - \phi_m(\boldsymbol{M}_{\hat{\alpha}_1}^{\mathrm{G}}(\bar{T}), \tau) \right\|_{\mathrm{F}}$$
$$= O\left( \left\| \boldsymbol{M}_{\hat{\alpha}}(\bar{T}) - \boldsymbol{M}_{\hat{\alpha}_1}^{\mathrm{G}}(\bar{T}) \right\|_{\mathrm{F}} \right) = O(\hat{\alpha}^{\frac{1}{P+1}}),$$

which proves the bound (36).

For (37), again by Lemma K.7, we have

$$\lambda_k^{1-P}(\boldsymbol{M}_{\hat{\alpha}}(\bar{T})) - \lambda_k^{1-P}(\boldsymbol{M}_{\hat{\alpha}}(\bar{T} + \tau))$$
$$\le \int_{\bar{T}}^{\bar{T}+\tau} 2(P-1) \left\| \nabla f(\boldsymbol{M}_{\hat{\alpha}}^P(t)) \right\|_2 \mathrm{d}t$$
$$\le \int_{\bar{T}}^{\bar{T}+\tau} 2(P-1) \left( \beta \left\| \boldsymbol{M}_{\hat{\alpha}}^P(t) - (\boldsymbol{M}^{\mathrm{G}})^P(t) \right\|_2 + \left\| \nabla f\left( (\boldsymbol{M}^{\mathrm{G}})^P(t) \right) \right\|_2 \right) \mathrm{d}t$$
$$\le \int_{\bar{T}}^{\bar{T}+\tau} 2(P-1)(O(\hat{\alpha}^{\frac{1}{1+P}}) + \beta \left\| \boldsymbol{M}^{\mathrm{G}}(t) \right\|_2^P) \mathrm{d}t$$
$$\le O(1).$$

Thus $\lambda_k^{1-P}(\boldsymbol{M}_{\hat{\alpha}}(\bar{T} + \tau)) = \Omega(\hat{\alpha}^{-(P-1)})$, that is, $\lambda_k(\boldsymbol{M}_{\hat{\alpha}}(\bar{T} + \tau)) = O(\hat{\alpha}), \forall k \ge 2$. $\qquad\square$

*Proof of Theorem 6.2.* Note that $\left(\overline{\boldsymbol{M}}(t)\right)^P = \overline{\boldsymbol{W}}(t)$ and

$$\left( \phi_m\left( \hat{\alpha}\boldsymbol{I}, \frac{\hat{\alpha}^{-(P-1)}}{2\mu_1(P-1)} + t \right) \right)^P = \phi\left( \alpha\boldsymbol{I}, \frac{\hat{\alpha}^{-(P-1)}}{2\mu_1(P-1)} + t \right).$$

By Theorem K.14, We have

$$\left\| \phi\left( \alpha\boldsymbol{I}, \frac{\alpha^{-(1-1/P)}}{2\mu_1(P-1)} + t \right) - \overline{\boldsymbol{W}}(t) \right\|_{\mathrm{F}}$$
$$\le \left\| \left( \phi_m\left( \hat{\alpha}\boldsymbol{I}, \frac{\hat{\alpha}^{-(P-1)}}{2\mu_1(P-1)} + t \right) \right)^P - \left( \overline{\boldsymbol{M}}(t) \right)^P \right\|_{\mathrm{F}}$$
$$\le P \left\| \phi_m\left( \hat{\alpha}\boldsymbol{I}, \frac{\hat{\alpha}^{-(P-1)}}{2\mu_1(P-1)} + t \right) - \overline{\boldsymbol{M}}(t) \right\|_{\mathrm{F}} \max\left( \left\| \phi_m\left( \hat{\alpha}\boldsymbol{I}, \frac{\hat{\alpha}^{-(P-1)}}{2\mu_1(P-1)} + t \right) \right\|_2, \left\| \overline{\boldsymbol{M}}(t) \right\|_2 \right)^{P-1}$$
$$= O(\hat{\alpha}^{\frac{1}{P+1}})O(1) = O(\alpha^{\frac{1}{P(P+1)}}),$$

and for $2 \le k \le d$,

$$\lambda_k\left( \phi\left( \alpha\boldsymbol{I}, \frac{\alpha^{-(1-1/P)}}{2\mu_1(P-1)} + t \right) \right) = \lambda_k\left( \phi_m\left( \hat{\alpha}\boldsymbol{I}, \frac{\hat{\alpha}^{-(P-1)}}{2\mu_1(P-1)} + t \right) \right) = O(\hat{\alpha}^P) = O(\alpha).$$

$$\square$$

## L  ESCAPING DIRECTION FOR DEEP MATRIX FACTORIZATION

For deep matrix factorization, recall that we only prove that GF with infinitesimal identity initialization escapes in the direction of the top eigenvector. The main burden for us to generalize this proof to general initialization is that we don't know how to analyze the early phase dynamics of (29), i.e., the analytical solution of (39) is difficult to compute, when $L \ge 3$. Intuitively, the direction that the infinitesimal initialization escapes $\boldsymbol{0}$ is exactly $\overline{\boldsymbol{M}} := \lim_{t \to \infty} \frac{\boldsymbol{M}(t)}{\|\boldsymbol{M}(t)\|_{\mathrm{F}}}$, where $\boldsymbol{M}(t)$ is the solution of (39). Showing $\overline{\boldsymbol{M}} = \boldsymbol{v}_1 \boldsymbol{v}_1^\top$ is a critical step in our analysis towards convergence to GLRL.

$$\frac{\mathrm{d}\boldsymbol{M}}{\mathrm{d}t} = -\nabla f(\boldsymbol{0})\boldsymbol{M}^{L/2} - \boldsymbol{M}^{L/2}\nabla f(\boldsymbol{0}). \tag{39}$$

However, unlike the depth-2 case, $\overline{M}$ can be different from $v_1 v_1^\top$ even if $v_1^\top M(0)v_1 > 0$. We here give an example for diagonal $M(0)$ and $\nabla f(\mathbf{0})$ at Appendix L.2. Nevertheless, we still conjecture that except for a zero measure set of $M(0)$, $\overline{M} = v_1 v_1^\top$, based on the following theoretical and experimental evidences:

- If $v_1^\top M(0)v_1 > 0$ and $\mathrm{rank}(M(0)) = 1$, we prove that $\overline{M} = v_1 v_1^\top$. (See Theorem L.1)
- For the counter-example, we show experimentally, even with perturbation of only magnitude $10^{-5}$, $\overline{M} = v_1 v_1^\top$. The results are shown at Figure 7. The $y$-axis indicates $\langle v_1, u_1(t) \rangle$ where $u_1(t)$ is the top eigenvector of $M(t)$. As $\|W(t)\|_\mathrm{F}$ becomes larger, $u_1(t)$ aligns better with $v_1$, which means the noise helps $M$ escaping from $v_1$. The larger the noise is, the faster $u_1(t)$ converges to $v_1$.

## L.1 RANK-ONE CASE

**Theorem L.1** (rank-1 initialization escapes along the top eigenvector). *When* $\mathrm{rank}(M(0)) = 1$, $\lim_{t \to \infty} \frac{M(t)}{\|M(t)\|_F} = v_1 v_1^\top$, *if* $v_1^\top M(0)v_1 > 0$.

*Proof.* Let $u(0)$ be the vector such that $M(0) = u(0)u(0)^\top$ and $u(t) \in \mathbb{R}^d$ be the solution of

$$\frac{\mathrm{d}u(t)}{\mathrm{d}t} = \|u(t)\|_2^{L-2} \nabla f(\mathbf{0})u(t).$$

It is easy to check that $M(t) = u(t)u(t)^\top$ is the solution of (39), because

$$\frac{\mathrm{d}M}{\mathrm{d}t} = \frac{\mathrm{d}u}{\mathrm{d}t}u^\top + u\frac{\mathrm{d}u}{\mathrm{d}t}^\top = -\nabla f(\mathbf{0})M(t)\|u(t)\|_2^{L-2} - M(t)\nabla f(\mathbf{0})\|u(t)\|_2^{L-2}$$

$$= -\nabla f(\mathbf{0})M^{L/2} - M^{L/2}\nabla f(\mathbf{0}).$$

Let $\tau(t) = \int_0^t \|u(s)\|_2^{L-2}\,\mathrm{d}s$. Then

$$\frac{\mathrm{d}u}{\mathrm{d}\tau} = \frac{\mathrm{d}u}{\mathrm{d}t}\frac{\mathrm{d}t}{\mathrm{d}\tau} = -\frac{1}{\frac{\mathrm{d}\tau}{\mathrm{d}t}}\|u\|_2^{L-2}\nabla f(\mathbf{0})u = -\nabla f(\mathbf{0})u.$$

That is, under time rescaling $t \to \tau(t)$, the trajectory of $u(t)$ still follows the power iteration, regardless of the depth $L$. □

## L.2 COUNTER-EXAMPLE FOR ESCAPING DIRECTION

Let $\nabla f(\mathbf{0}) = \mathrm{diag}(2, 0.9, 0.8, \ldots, 0.1) \in \mathbb{R}^{10 \times 10}$ be diagonal. Let $W(0)$ be also diagonal and $W(0)_{i,i} \sim \mathrm{Unif}[0.9, 1.1] \cdot \alpha$ for $i \in [10] \setminus \{2\}$, $W(0)_{2,2} = 16\alpha$, where $\alpha = 10^{-16}$ is a small constant. Let the depth be 4.

**Lemma L.2.** *With* $\nabla f(\mathbf{0})$ *and* $W(0)$ *constructed above,* $v_1 M(0)v_1^\top > 0$ *and* $\overline{M} \neq v_1 v_1^\top$.

*Proof.* It is easy to check that $v_1 = e_1$, so $v_1 M(0)v_1^\top > 0$. Now we prove that $\overline{M}(\infty) \neq v_1 v_1^\top$.

As both $W(0)$ and $\nabla f(\mathbf{0})$ are diagonal, $W(t)$ is always diagonal and has dynamics

$$\frac{\mathrm{d}M(t)_{i,i}}{\mathrm{d}t} = -2\nabla f(\mathbf{0})_{i,i}M(t)_{i,i}^2, \quad \forall i \in [10],$$

therefore we have closed form of $M(t)$:

$$M(t)_{i,i}^{-1} = M(0)_{i,i}^{-1} - 2\nabla f(\mathbf{0})_{i,i}t, \quad \forall i \in [10].$$

For $i \in [10]$, the time for $M(t)_{i,i}$ going to infinity is $(2M(0)_{i,i}\nabla f(\mathbf{0})_{i,i})^{-1}$. By simple calculation, $M(t)_{2,2}$ goes to infinity fastest, thus $\overline{M} = e_2 e_2^\top \neq v_1 v_1^\top$. □

We remark that the scales of $W(0)$ and $\nabla f(\mathbf{0})$ do not matter as in gradient flow, as scaling $\nabla f(\mathbf{0})$ is equivalent to scaling time (by Lemma L.3 below). And for this reason, the $x$-axis is the chosen as $\frac{\|W(t)\|_\mathrm{F}}{\|W(0)\|_\mathrm{F}}$, the relative growth rate.

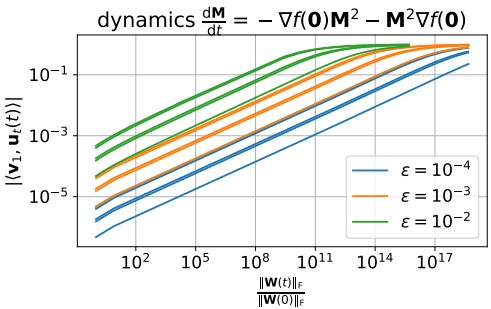

Figure 7: Dynamics of $\frac{\mathrm{d}M}{\mathrm{d}t} = -\nabla f(\mathbf{0})M^{L/2} - M^{L/2}\nabla f(\mathbf{0})$ plotted, where $L = 4$, $\boldsymbol{u}_1(t)$ is the top eigenvector of $\boldsymbol{W}(t)$ and $\epsilon$ is the relative magnitude of noise. The initialization we use in this experiment is $\boldsymbol{W}_{\mathrm{noise}}(0) = \boldsymbol{W}(0) + \frac{\alpha\epsilon}{2}(\boldsymbol{Z} + \boldsymbol{Z}^\top)$, where $\boldsymbol{W}(0)$ is what we construct at Appendix L.2, and $\boldsymbol{Z}$ is a matrix where entries are i.i.d. samples from the standard Gaussian distribution $\mathcal{N}(0, 1)$. We run 5 fixed random seeds (the noise matrix) for each $\epsilon$. The trajectory of $\boldsymbol{W}$ is calculated by simulating gradient flow on $M$ with small timestep and RMSprop (Tieleman and Hinton, 2012) for faster convergence.

**Lemma L.3.** *Suppose $\boldsymbol{g} : \mathbb{R}^d \to \mathbb{R}^d$ is a P-homogeneous function, that is, $\boldsymbol{g}(\alpha\boldsymbol{\theta}) = \lambda^P \boldsymbol{g}(\alpha)$ for any $\alpha > 0$, and $\frac{\mathrm{d}\boldsymbol{\theta}'(t)}{\mathrm{d}t} = \boldsymbol{g}(\boldsymbol{\theta}'(t))$. Then $\alpha\boldsymbol{\theta}'(\alpha^{P-1}t)$ is the solution of*

$$\frac{\mathrm{d}\boldsymbol{\theta}(t)}{\mathrm{d}t} = \boldsymbol{g}(\boldsymbol{\theta}(t)), \qquad \boldsymbol{\theta}(0) = \alpha\boldsymbol{\theta}'(0). \tag{40}$$

*Proof.* Simply plug in $\boldsymbol{\theta}(t) = \alpha\boldsymbol{\theta}'(\alpha^{P-1}t)$, then we have

$$\frac{\mathrm{d}\boldsymbol{\theta}(t)}{\mathrm{d}t} = \frac{\mathrm{d}\alpha\boldsymbol{\theta}'(\alpha^{P-1}t)}{\mathrm{d}t} = \alpha^P \frac{\mathrm{d}\boldsymbol{\theta}'(\alpha^{P-1}t)}{\mathrm{d}(\alpha^{P-1}t)} = \alpha^P \boldsymbol{g}(\boldsymbol{\theta}'(\alpha^{P-1}t)) = \boldsymbol{g}(\alpha\boldsymbol{\theta}'(\alpha^{P-1}t)) = \boldsymbol{g}(\boldsymbol{\theta}(t)).$$

$\square$

## M   PROOF OF LINEAR CONVERGENCE TO MINIMIZER

In this section, we will present the theorems that guarantee the linear convergence to a minimizer $\boldsymbol{W}_0$ of $f(\,\cdot\,)$ if the dynamics (41) is initialized sufficiently close to $\boldsymbol{W}_0$, i.e., $\|\boldsymbol{W}(0) - \boldsymbol{W}_0\|_{\mathrm{F}}$ is sufficiently small. In Appendix M.3, we will apply this result to prove Theorem 6.4.

$$\frac{\mathrm{d}\boldsymbol{W}}{\mathrm{d}t} = -\sum_{i=0}^{L-1} \boldsymbol{W}^{\frac{2i}{L}}\nabla f(\boldsymbol{W})\boldsymbol{W}^{2-\frac{2i+2}{L}} =: \boldsymbol{g}(\boldsymbol{W}). \tag{41}$$

Throughout this section, we assume $\mathrm{rank}(\boldsymbol{W}_0) = k$ and use $m := \lambda_k(\boldsymbol{W}_0)$ to denote the $k$-th smallest non-zero eigenvalue of $\boldsymbol{W}_0$. The tangent space of manifold of rank-$k$ symmetric matrices at $\boldsymbol{W}_0$ is $\mathcal{T} = \{\boldsymbol{V}\boldsymbol{W}_0^\top + \boldsymbol{W}_0\boldsymbol{V}^\top : \boldsymbol{V} \in \mathbb{R}^{d\times k}\}$. It can be shown that $\dim(\mathcal{T}) = k(d-k) + \frac{k(k+1)}{2} = \frac{k(2d-k+1)}{2}$.

Let $\boldsymbol{J}(\boldsymbol{W})$ be the Jacobian of $\boldsymbol{g}(\boldsymbol{W})$ in (41). For depth-2 case, we have shown that $\mathcal{T}$ is an invariant subspace of $\boldsymbol{J}(\boldsymbol{W}_0)$ in Theorem H.5, property 2. This can be generalize to the deep case where $L \geq 3$. Therefore, we can use $\boldsymbol{J}(\boldsymbol{W}_0)|_{\mathcal{T}} : \mathcal{T} \to \mathcal{T}$ to denote the linear operator $\boldsymbol{J}(\boldsymbol{W}_0)$ restricted on $\mathcal{T}$. We also define $\Pi_1^{d^2}(\boldsymbol{W})$ as the projection of $\boldsymbol{W} \in \mathbb{R}^{d\times d}$ on $\mathcal{T}$, and $\Pi_2^{d^2}(\boldsymbol{W}) := \boldsymbol{W} - \Pi_1^{d^2}(\boldsymbol{W})$.

Towards showing the main convergence result in the section, we make the following assumption.

**Assumption M.1.** Suppose $\boldsymbol{J}(\boldsymbol{W}_0)|_{\mathcal{T}}$ diagonalizable and all eigenvalues are negative real numbers.

$\boldsymbol{W}_0$ is a minimizer, so it is clear that $\boldsymbol{J}(\boldsymbol{W}_0)|_{\mathcal{T}}$ has no eigenvalues with positive real parts (otherwise there is a descending direction of $f(\,\cdot\,)$ from $\boldsymbol{W}_0$, since the loss $f(\,\cdot\,)$ strictly decreases along

the trajectory of (41)). If further Assumption M.1 holds, then we know $\boldsymbol{J}(\boldsymbol{W}_0)|_{\mathcal{T}}: \mathcal{T} \to \mathcal{T}$ can be diagonalized as $\boldsymbol{J}(\boldsymbol{W}_0)|_{\mathcal{T}}[\cdot] = \mathcal{V}(\Sigma \mathcal{V}^{-1}(\cdot))$, where $\Sigma_i = \text{diag}(-\mu_1, \ldots, -\mu_{\dim(\mathcal{T})})$, $\mathcal{V}: \mathbb{R}^{\dim(\mathcal{T})} \to \mathcal{T}$, $\mathcal{V}(\boldsymbol{x}) = \sum_{i=1}^{\dim(\mathcal{T})} x_i \boldsymbol{V}_i$, and $\boldsymbol{V}_i$ is the eigenvector associated with eigenvalue $-\mu_i$.

As shown in Theorem M.3 below, this assumption implies that if $\boldsymbol{W}(0)$ is rank-$k$ and is sufficiently close to $\boldsymbol{W}_0$, then $\|\boldsymbol{W}(t) - \boldsymbol{W}_0\|_F \leq Ce^{-\mu_1 t}$ for some constant $C$. For depth-2 case, the above assumption is equivalent to that $\mathcal{L}(\boldsymbol{U}_0)$ is "strongly convex" at $\boldsymbol{U}_0$, except those 0 eigenvalues due to symmetry, by property 2 of Theorem H.5). For the case where $L \geq 3$, because this dynamics is not gradient flow, in general it does not correspond to a loss function and strongly convexity does not make any sense. Nevertheless, in experiments we do observe linear convergence to $\boldsymbol{W}_0$, so this assumption is reasonable.

## M.1 RANK-$k$ INITIALIZATION

For convenience, we define for all $\boldsymbol{W} \in \mathbb{S}_d$,

$$\|\boldsymbol{W}\|_{\mathcal{V}} := \left\|\mathcal{V}^{-1}\left(\Pi_1^{d^2}(\boldsymbol{W})\right)\right\|_{\mathrm{F}}, \quad \|\boldsymbol{W}\|_{\mathrm{F},1} := \left\|\Pi_1^{d^2}(\boldsymbol{W})\right\|_{\mathrm{F}}, \quad \|\boldsymbol{W}\|_{\mathrm{F},2} := \left\|\Pi_2^{d^2}(\boldsymbol{W})\right\|_{\mathrm{F}}.$$

The reason for such definition of norms, as we will see later, is that the norm (or the difference) in the tangent space of the manifold of symmetric rank-$r$ matrices, $\|\boldsymbol{W} - \boldsymbol{W}'\|_{\mathrm{F},1}$, dominates that in the orthogonal complement of the tangent space, $\|\boldsymbol{W} - \boldsymbol{W}'\|_{\mathrm{F},2}$, when both $\boldsymbol{W}, \boldsymbol{W}'$ get very close to the $\boldsymbol{W}_0$ (see a more rigorous statement in Lemma M.2). WLOG, we can assume

$$\frac{\|\cdot\|_{\mathrm{F},1}}{K} \leq \|\cdot\|_{\mathcal{V}} \leq \|\cdot\|_{\mathrm{F},1},$$

for some constant $K$, which may depend on $f$ and $\boldsymbol{W}_0$. This also implies that $\|\cdot\|_{\mathcal{V}} \leq \|\cdot\|_{\mathrm{F}}$. Below we also assume for sufficiently small $R$, and any $\boldsymbol{W}$ such that $\|\boldsymbol{W} - \boldsymbol{W}_0\|_{\mathrm{F}} \leq R$, we have $\|\nabla f(\boldsymbol{W})\|_2 \leq \rho$ and $\|\boldsymbol{J}(\boldsymbol{W})[\boldsymbol{\Delta}]\|_{\mathrm{F}} \leq \beta \|\boldsymbol{\Delta}\|_{\mathrm{F}}$ for any $\boldsymbol{\Delta}$. In the proof below, we assume such properties hold as long as we can show the boundedness of $\boldsymbol{W}(t) - \boldsymbol{W}_0$.

**Lemma M.2.** *Let* $\max\{\|\boldsymbol{W} - \boldsymbol{W}_0\|_{\mathrm{F},1}, \|\boldsymbol{W}' - \boldsymbol{W}_0\|_{\mathrm{F},1}\} = r$, *when* $r \leq \frac{m}{2}$, *we have*

$$\|\boldsymbol{W} - \boldsymbol{W}'\|_{\mathrm{F},2} \leq \frac{5r}{m} \|\boldsymbol{W} - \boldsymbol{W}'\|_{\mathrm{F},1}.$$

*As a special case, we have*

$$\|\boldsymbol{W} - \boldsymbol{W}_0\|_{\mathrm{F},2} \leq \frac{5 \|\boldsymbol{W} - \boldsymbol{W}'\|_{\mathrm{F},1}^2}{m}.$$

*Proof.* WLOG we can assume $\boldsymbol{W}_0$ is only non-zero in the first $k$ dimension, i.e., $[\boldsymbol{W}_0]_{ij} = 0$, for all $i \geq k+1, j \geq k+1$. We further denote $\boldsymbol{W}$ and $\boldsymbol{W}'$ by

$$\boldsymbol{W} = \begin{bmatrix} \boldsymbol{A} & \boldsymbol{B}^\top \\ \boldsymbol{B} & \boldsymbol{C} \end{bmatrix} \quad \text{and} \quad \boldsymbol{W}' = \begin{bmatrix} \boldsymbol{A}' & \boldsymbol{B}'^\top \\ \boldsymbol{B}' & \boldsymbol{C}' \end{bmatrix},$$

where $\boldsymbol{A}, \boldsymbol{A}' \in \mathbb{R}^{k \times k}, \boldsymbol{B}, \boldsymbol{B}' \in \mathbb{R}^{(d-k) \times k}, \boldsymbol{C}, \boldsymbol{C}' \in \mathbb{R}^{(d-k) \times (d-k)}$. By definition, we have $\|\boldsymbol{A} - \boldsymbol{A}'\|_F, \|\boldsymbol{B} - \boldsymbol{B}'\|_F \leq \|\boldsymbol{W} - \boldsymbol{W}'\|_{\mathrm{F},1}$ and $\|\boldsymbol{W} - \boldsymbol{W}'\|_{\mathrm{F},2} = \|\boldsymbol{C} - \boldsymbol{C}'\|_F$. Moreover, we have $\lambda_{\min}(\boldsymbol{A}) \geq m - \|\boldsymbol{A} - \boldsymbol{W}_0\|_F \geq m - \|\boldsymbol{W} - \boldsymbol{W}_0\|_{\mathrm{F},1} \geq \frac{m}{2}$.

Since $\boldsymbol{W}, \boldsymbol{W}'$ is rank-$k$, we have $\boldsymbol{C} = \boldsymbol{B}\boldsymbol{A}^{-1}\boldsymbol{B}^\top, \boldsymbol{C}' = \boldsymbol{B}'\boldsymbol{A}'^{-1}\boldsymbol{B}'^\top$. Thus

$$\begin{aligned}
&\|\boldsymbol{W} - \boldsymbol{W}'\|_{\mathrm{F},2} \\
&= \|\boldsymbol{C} - \boldsymbol{C}'\|_F \\
&= \left\|\boldsymbol{B}\boldsymbol{A}^{-1}\boldsymbol{B}^\top - \boldsymbol{B}'\boldsymbol{A}'^{-1}\boldsymbol{B}'^\top\right\|_F \\
&\leq \|\boldsymbol{B} - \boldsymbol{B}'\|_{\mathrm{F}}\|\boldsymbol{A}^{-1}\boldsymbol{B}^\top\|_{\mathrm{F}} + \|\boldsymbol{B}\boldsymbol{A}^{-1}\|_{\mathrm{F}}\|\boldsymbol{A}' - \boldsymbol{A}\|_{\mathrm{F}}\|\boldsymbol{A}'^{-1}\boldsymbol{B}'^\top\|_{\mathrm{F}} + \|\boldsymbol{B}'\boldsymbol{A}'^{-1}\|_{\mathrm{F}}\|\boldsymbol{B}^\top - \boldsymbol{B}'^\top\|_{\mathrm{F}} \\
&\leq \|\boldsymbol{W} - \boldsymbol{W}'\|_{\mathrm{F},1}\frac{2r}{m} + \|\boldsymbol{W} - \boldsymbol{W}'\|_{\mathrm{F},1}\left(\frac{2r}{m}\right)^2 + \|\boldsymbol{W} - \boldsymbol{W}'\|_{\mathrm{F},1}\frac{2r}{m} \\
&\leq \|\boldsymbol{W} - \boldsymbol{W}'\|_{\mathrm{F},1}\frac{5r}{m}.
\end{aligned}$$

$\square$

**Theorem M.3** (Linear convergence of rank-$k$ matrices). *Suppose that* $\text{rank}(\boldsymbol{W}(0)) = \text{rank}(\boldsymbol{W}_0) = k$ *and*

$$\|\boldsymbol{W}(0) - \boldsymbol{W}_0\|_{\mathcal{V}} \leq R := \max\left\{\frac{m}{2K}, \frac{\mu_1}{K^2(29\beta + 10\rho/m)}\right\},$$

*we have* $\|\boldsymbol{W}(t) - \boldsymbol{W}_0\|_{\mathcal{V}} \leq Ce^{-\mu_1 t}\|\boldsymbol{W}(0) - \boldsymbol{W}_0\|_{\mathcal{V}}$ *for some constant $C$ depending on $\boldsymbol{W}_0$, where $\boldsymbol{W}(t)$ satisfies* (41).

*Proof.* For convenience, we define $\boldsymbol{W}_1(t) := \Pi_1^{d^2}\left(\boldsymbol{W}(t) - \boldsymbol{W}_0\right), \boldsymbol{W}_2(t) := \Pi_2^{d^2}\left(\boldsymbol{W}(t) - \boldsymbol{W}_0\right) = \Pi_2^{d^2}\left(\boldsymbol{W}(t)\right)$. We also use $\langle\cdot, \cdot\rangle_{\mathcal{V}^{-1}} = \left\langle\mathcal{V}^{-1}\left(\cdot\right), \mathcal{V}^{-1}\left(\cdot\right)\right\rangle$ for short.

$$\begin{aligned}
\frac{\mathrm{d}\|\boldsymbol{W}_1(t)\|_{\mathcal{V}}^2}{\mathrm{d}t} &= \frac{\mathrm{d}\|\boldsymbol{W}(t) - \boldsymbol{W}_0\|_{\mathcal{V}}^2}{\mathrm{d}t} \\
&= 2\left\langle\Pi_1^{d^2}\left(\frac{\mathrm{d}\boldsymbol{W}(t)}{\mathrm{d}t}\right), \Pi_1^{d^2}\left(\boldsymbol{W}(t) - \boldsymbol{W}_0\right)\right\rangle_{\mathcal{V}^{-1}} \\
&= 2\left\langle\Pi_1^{d^2}\left(\boldsymbol{g}(\boldsymbol{W}(t))\right), \boldsymbol{W}_1(t)\right\rangle_{\mathcal{V}^{-1}} \\
&\leq 2\left\langle\Pi_1^{d^2}\left(\boldsymbol{J}(\boldsymbol{W}_0)[\boldsymbol{W}(t) - \boldsymbol{W}_0]\right), \boldsymbol{W}_1(t)\right\rangle_{\mathcal{V}^{-1}} \\
&\quad + 2\|\boldsymbol{g}(\boldsymbol{W}(t) - \boldsymbol{W}_0) - \boldsymbol{J}(\boldsymbol{W}_0)[\boldsymbol{W}(t) - \boldsymbol{W}_0]\|_{\mathcal{V}}\|\boldsymbol{W}(t) - \boldsymbol{W}_0\|_{\mathcal{V}} \\
&= 2\left\langle\Pi_1^{d^2}\left(\boldsymbol{J}(\boldsymbol{W}_0)[\boldsymbol{W}_1(t) + \boldsymbol{W}_2(t)]\right), \boldsymbol{W}_1(t)\right\rangle_{\mathcal{V}^{-1}} \\
&\quad + 2\|\boldsymbol{g}(\boldsymbol{W}(t) - \boldsymbol{W}_0) - \boldsymbol{J}(\boldsymbol{W}_0)[\boldsymbol{W}(t) - \boldsymbol{W}_0]\|_{\mathcal{V}}\|\boldsymbol{W}_1(t)\|_{\mathcal{V}} \\
&= 2\left\langle\Pi_1^{d^2}\left(\boldsymbol{J}(\boldsymbol{W}_0)[\boldsymbol{W}_1(t)]\right), \boldsymbol{W}_1(t)\right\rangle_{\mathcal{V}^{-1}} + 2\|\boldsymbol{J}(\boldsymbol{W}_0)[\boldsymbol{W}_2(t)]\|_{\mathcal{V}}\|\boldsymbol{W}_1(t)\|_{\mathcal{V}} \\
&\quad + 2\|\boldsymbol{g}(\boldsymbol{W}(t) - \boldsymbol{W}_0) - \boldsymbol{J}(\boldsymbol{W}_0)[\boldsymbol{W}(t) - \boldsymbol{W}_0]\|_{\mathcal{V}}\|\boldsymbol{W}_1(t)\|_{\mathcal{V}}.
\end{aligned}$$

For the first term $\left\langle\Pi_1^{d^2}\left(\boldsymbol{J}(\boldsymbol{W}_0)[\boldsymbol{W}_1(t)]\right), \boldsymbol{W}_1(t)\right\rangle_{\mathcal{V}^{-1}}$, we know $\boldsymbol{W}_1(t) \in \mathcal{T}$, and $\mathcal{T}$ is an invariant space of $\boldsymbol{J}(\boldsymbol{W}_0)$. Recall $\boldsymbol{J}(\boldsymbol{W}_0)|_{\mathcal{T}}[\cdot] = \mathcal{V}\left(\Sigma\mathcal{V}^{-1}\left(\cdot\right)\right)$, we have

$$2\left\langle\Pi_1^{d^2}\left(\boldsymbol{J}(\boldsymbol{W}_0)[\boldsymbol{W}_1(t)]\right), \boldsymbol{W}_1(t)\right\rangle_{\mathcal{V}^{-1}} = 2\left\langle\Sigma\mathcal{V}^{-1}\left(\boldsymbol{W}_1(t)\right), \mathcal{V}^{-1}\left(\boldsymbol{W}_1(t)\right)\right\rangle \leq -2\mu_1\|\boldsymbol{W}_1(t)\|_{\mathrm{F},1}.$$

For the second term $2\beta\|\boldsymbol{J}(\boldsymbol{W}_0)[\boldsymbol{W}_2(t)]\|_{\mathcal{V}}\|\boldsymbol{W}_1(t)\|_{\mathcal{V}}$, we have

$$2\|\boldsymbol{J}(\boldsymbol{W}_0)[\boldsymbol{W}_2(t)]\|_{\mathcal{V}} \leq 2\|\boldsymbol{J}(\boldsymbol{W}_0)[\boldsymbol{W}_2(t)]\|_{\mathrm{F}} \leq 2\|\boldsymbol{J}(\boldsymbol{W}_0)\|_2\|\boldsymbol{W}_2(t)\|_{\mathrm{F}} = 2\rho\|\boldsymbol{W}_2(t)\|_{\mathrm{F}}.$$

For the third term $2\|\boldsymbol{g}(\boldsymbol{W}(t) - \boldsymbol{W}_0) - \boldsymbol{J}(\boldsymbol{W}_0)[\boldsymbol{W}(t) - \boldsymbol{W}_0]\|_{\mathcal{V}}\|\boldsymbol{W}_1(t)\|_{\mathcal{V}}$, we have

$$\begin{aligned}
2\|\boldsymbol{g}(\boldsymbol{W}(t) - \boldsymbol{W}_0) - \boldsymbol{J}(\boldsymbol{W}_0)[\boldsymbol{W}(t) - \boldsymbol{W}_0]\|_{\mathcal{V}} &\leq 2\beta\|\boldsymbol{W}(t) - \boldsymbol{W}_0\|_{\mathrm{F}}^2 \\
&\leq 4\beta(\|\boldsymbol{W}_1(t)\|_F^2 + \|\boldsymbol{W}_2(t)\|_{\mathrm{F}}^2) \\
&\leq 4\beta(K^2\|\boldsymbol{W}_1(t)\|_{\mathcal{V}}^2 + \|\boldsymbol{W}_2(t)\|_{\mathrm{F}}^2).
\end{aligned}$$

Thus we have shown the following. Note so far we have not used the assumption that $\boldsymbol{W}$ is rank-$k$.

$$\frac{\mathrm{d}\|\boldsymbol{W}_1(t)\|_{\mathcal{V}}^2}{\mathrm{d}t} \leq -2\mu_1\|\boldsymbol{W}_1(t)\|_{\mathcal{V}}^2 + 2\|\boldsymbol{W}_1(t)\|_{\mathcal{V}}\left(\rho\|\boldsymbol{W}_2(t)\|_{\mathrm{F}} + 2\beta K^2\|\boldsymbol{W}_1(t)\|_{\mathcal{V}}^2 + 2\beta\|\boldsymbol{W}_2(t)\|_{\mathrm{F}}^2\right),$$

that is,

$$\frac{\mathrm{d}\log\|\boldsymbol{W}_1(t)\|_{\mathcal{V}}^2}{\mathrm{d}t} \leq -2\mu_1 + 4\beta K^2\|\boldsymbol{W}_1(t)\|_{\mathcal{V}} + \frac{4\beta\|\boldsymbol{W}_2(t)\|_{\mathrm{F}}^2 + 2\rho\|\boldsymbol{W}_2(t)\|_{\mathrm{F}}}{\|\boldsymbol{W}_1(t)\|_{\mathcal{V}}}. \quad (42)$$

Let $T := \sup\{t \geq 0 : \|\boldsymbol{W}_1(t)\|_{\mathcal{V}} \leq \frac{m}{2K}\}$. Setting $\boldsymbol{W}' = \boldsymbol{W}_0$ in Lemma M.2, we have for $t < T$, $r = \|\boldsymbol{W}(t) - \boldsymbol{W}_0\|_{\mathrm{F},1} \leq \|\boldsymbol{W}(t) - \boldsymbol{W}_0\|_{\mathrm{F}} \leq K\|\boldsymbol{W}(t) - \boldsymbol{W}_0\|_{\mathcal{V}} \leq \frac{m}{2}$. Thus,

$$\|\boldsymbol{W}_2(t)\|_{\mathrm{F}} = \|\boldsymbol{W}_2(t)\|_{\mathrm{F},2} \leq \frac{5\|\boldsymbol{W}(t) - \boldsymbol{W}_0\|_{\mathrm{F},1}^2}{m} \leq \frac{5K^2\|\boldsymbol{W}(t) - \boldsymbol{W}_0\|_{\mathcal{V}}^2}{m} \leq \frac{5}{4}m.$$

Thus, from (42) we can derive that

$$\frac{\mathrm{d}\log\|\boldsymbol{W}_1(t)\|_{\mathcal{V}}^2}{\mathrm{d}t} \leq -2\mu_1 + K^2(29\beta + 10\rho/m))\|\boldsymbol{W}_1(t)\|_{\mathcal{V}} \leq -\mu_1. \tag{43}$$

Since $\mu_1 < 0$, $\|\boldsymbol{W}_1(t)\|_{\mathcal{V}}$ decreases for $[0, T)$. Thus $T$ must be $\infty$, otherwise $\|\boldsymbol{W}_1(T)\|_{\mathcal{V}} = \lim_{t \to T^-} \|\boldsymbol{W}_1(t)\|_{\mathcal{V}} < R_1$. Contradiction.

Therefore, for any $t \in [0, \infty)$, we have $\|\boldsymbol{W}_1(t)\|_{\mathcal{V}} \leq \|\boldsymbol{W}_1(0)\|_{\mathcal{V}} e^{-\frac{\mu_1}{2}t}$. That is,

$$\int_0^\infty \|\boldsymbol{W}_1(t)\|_{\mathcal{V}}\,\mathrm{d}t \leq \frac{2}{\mu_1}\|\boldsymbol{W}_1(0)\|_{\mathcal{V}} \leq \frac{2R}{\mu_1}.$$

Thus from (43), we have

$$\begin{aligned}
\|\boldsymbol{W}(t)\|_{\mathcal{V}} = \|\boldsymbol{W}_1(t)\|_{\mathcal{V}} &\leq \|\boldsymbol{W}_1(0)\|_{\mathcal{V}} \exp\left(-\mu_1 t + \frac{K^2}{2}(29\beta + 10\rho/m)\int_0^\infty \|\boldsymbol{W}_1(t)\|_{\mathcal{V}}\,\mathrm{d}t\right) \\
&\leq \|\boldsymbol{W}_1(0)\|_{\mathcal{V}} \exp\left(-\mu_1 t + \frac{K^2 R}{\mu_1}(29\beta + 10\rho/m)\right) \\
&=: C\|\boldsymbol{W}(0)\|_{\mathcal{V}} e^{-\mu_1 t},
\end{aligned}$$

which completes the proof. $\qquad\square$

## M.2 Almost Rank-$k$ Initialization

We use $\boldsymbol{M}(t)$ to denote the top-$k$ components of $\boldsymbol{W}(t)$ in SVD, and $\boldsymbol{N}(t)$ to denote the rest part, i.e., $\boldsymbol{W}(t) - \boldsymbol{M}(t)$. One can think $\boldsymbol{M}(t)$ as the main part and $\boldsymbol{N}(t)$ as the negligible part.

Below we show that for deep overparametrized matrix factorization, where $\boldsymbol{W}(t)$ satisfies (41), if the trajectory is initialized at some $\boldsymbol{W}(0)$ in a small neighborhood of the $k$-th critical point $\boldsymbol{W}_0$ of deep GLRL, and $\boldsymbol{W}(0)$ is approximately rank-$k$, in the sense that $\boldsymbol{N}(0)$ is very small, then $\inf_{t \geq 0}\|\boldsymbol{W}(t) - \boldsymbol{W}_0\|_{\mathcal{V}}$ is roughly at the same magnitude of $\boldsymbol{N}(0)$.

**Theorem M.4** (Linear convergence of almost rank-$k$ matrices, deep case)**.** *Suppose $\boldsymbol{W}_0$ is a critical point of rank $k$ and $\boldsymbol{W}_0$ satisfies Assumption M.1, there exists constants $C_0$ and $r$, such that if $C_0\|\boldsymbol{N}(0)\|_{\mathrm{F}} \leq \|\boldsymbol{W}_1(0)\|_{\mathcal{V}} \leq r$, then there exists a time $T$ and constants $C, C'$, such that*

*(1).* $\|\boldsymbol{W}(t) - \boldsymbol{W}_0\|_{\mathcal{V}} \leq Ce^{-\mu_1 t/2}\|\boldsymbol{W}(0) - \boldsymbol{W}_0\|_{\mathcal{V}}$*, for $t \leq T$.*

*(2).* $\|\boldsymbol{W}(T) - \boldsymbol{W}_0\|_{\mathrm{F}} \leq C'\|\boldsymbol{N}(0)\|_{\mathrm{F}}$*.*

*Proof.* When $\|\boldsymbol{W}(t) - \boldsymbol{W}_0\|_{\mathrm{F}} \leq \frac{\lambda_{\min}(\boldsymbol{W}_0)}{4}$, $\|\boldsymbol{N}(t)\|_{\mathrm{F}} \leq \frac{\lambda_{\min}(\boldsymbol{W}_0)}{4}$, thus we have

$$\|\boldsymbol{M}(t) - \boldsymbol{W}_0\|_{\mathrm{F},1} \leq \|\boldsymbol{W}(t) - \boldsymbol{W}_0\|_{\mathrm{F},1} + \|\boldsymbol{N}(t)\|_{\mathrm{F},1} \leq \frac{\lambda_{\min}(\boldsymbol{W}_0)}{2},$$

thus by Lemma M.2, we have

$$\begin{aligned}
\|\boldsymbol{W}_2(t)\|_{\mathrm{F},2} &\leq \|\boldsymbol{M}(t) - \boldsymbol{W}_0\|_{\mathrm{F},2} + \|\boldsymbol{N}(t)\|_{\mathrm{F},2} \\
&\leq \frac{5\|\boldsymbol{M}(t) - \boldsymbol{W}_0\|_{\mathrm{F},1}^2}{\lambda_{\min}(\boldsymbol{W}_0)} + \|\boldsymbol{N}(t)\|_{\mathrm{F},2} \\
&\leq \frac{10\|\boldsymbol{W}_1(t)\|_{\mathrm{F},1}^2 + 10\|\boldsymbol{N}(t)\|_{\mathrm{F}}^2}{\lambda_{\min}(\boldsymbol{W}_0)} + \|\boldsymbol{N}(t)\|_{\mathrm{F},2} \\
&\leq \frac{10K^2\|\boldsymbol{W}_1(t)\|_{\mathcal{V}}^2 + 10\|\boldsymbol{N}(t)\|_{\mathrm{F}}^2}{\lambda_{\min}(\boldsymbol{W}_0)} + \|\boldsymbol{N}(t)\|_{\mathrm{F},2}.
\end{aligned}$$

Thus we can pick constant $C_0$ large enough and $r$ small enough, such that for any $t \geq 0$, if $C_0\|\boldsymbol{N}(t)\|_{\mathrm{F}} \leq \|\boldsymbol{W}_1(t)\|_{\mathcal{V}} \leq r$, then it holds that:

- The "small terms" in the RHS of (42) satisfies that

$$4\beta K^2 \left\| \boldsymbol{W}_1(t) \right\|_{\mathcal{V}} + \frac{4\beta \left\| \boldsymbol{W}_2(t) \right\|_{\mathrm{F}}^2 + 2\rho \left\| \boldsymbol{W}_2(t) \right\|_{\mathrm{F}}}{\left\| \boldsymbol{W}_1(t) \right\|_{\mathcal{V}}} \leq C_1 \left\| \boldsymbol{W}_1(t) \right\|_{\mathcal{V}} + C_2 \left\| \boldsymbol{N}(t) \right\|_{\mathrm{F}} \leq \mu_1$$

for some $C_1$ and $C_2$ independent of $t$.

- The spectral norm $\frac{1}{2} \left\| \nabla f(\boldsymbol{W}(t)) \right\|_2 \leq \left\| \nabla f(\boldsymbol{W}_0) \right\|_2 =: \rho$ for all $t \geq 0$.

- $\forall x < r, \frac{\kappa_L x^{\frac{2}{L}-1}}{(L-2)\rho} > \frac{2}{\mu_1} \ln \frac{2r}{C_0 x}$, where $\kappa_L = 1 - 0.5^{\frac{L-2}{L}}$.

Note these conditions can always be satisfied by some $C_0$ and $r$ because we can first find 3 groups $(C_0, r)$ to satisfy each individual condition, and then take the maximal $C_0$ and minimal $r$, it's easy to check these conditions are still verified. And we let $T_{C_0,r}$ be the earliest time that such condition, i.e., $C_0 \left\| \boldsymbol{N}(t) \right\|_{\mathrm{F}} \leq \left\| \boldsymbol{W}_1(t) \right\|_{\mathcal{V}} \leq r$ fails. Thus by (42), for $t \in [0, T_{C_0,r})$, we have $\left\| \boldsymbol{W}(t) \right\|_{\mathcal{V}} = \left\| \boldsymbol{W}_1(t) \right\|_{\mathcal{V}} \leq \left\| \boldsymbol{W}_1(0) \right\|_{\mathcal{V}} e^{-\frac{\mu_1 t}{2}} = \left\| \boldsymbol{W}(0) \right\|_{\mathcal{V}} e^{-\frac{\mu_1 t}{2}}$. Thus (1) holds for any $T$ smaller than $T_{C_0,r}$. If $T_{C_0,r} = \infty$, then clearly we can pick a sufficiently large $T$, such that (2) holds. Therefore, below it suffices to consider the case where $T_{C_0,r}$ is finite. And we know the condition that fails must be $C_0 \left\| \boldsymbol{N}(t) \right\|_{\mathrm{F}} \leq \left\| \boldsymbol{W}_1(t) \right\|_{\mathcal{V}}$, i.e. $C_0 \left\| \boldsymbol{N}(T_{C_0,r}) \right\|_{\mathrm{F}} = \left\| \boldsymbol{W}_1(T_{C_0,r}) \right\|_{\mathcal{V}}$.

By (34) in Lemma K.7, we have

$$\left| \left\| \boldsymbol{N}(0) \right\|_2^{\frac{2}{L}-1} - \left\| \boldsymbol{N}(t) \right\|_2^{\frac{2}{L}-1} \right| \leq (L-2)\rho t.$$

Define $T' := \frac{\kappa_L \left\| \boldsymbol{N}(0) \right\|_2^{\frac{2}{L}-1}}{(L-2)\rho}$, we know for any $t < T'$, we have $\left| \left\| \boldsymbol{N}(0) \right\|_2^{\frac{2}{L}-1} - \left\| \boldsymbol{N}(t) \right\|_2^{\frac{2}{L}-1} \right| \leq \kappa_L \left\| \boldsymbol{N}(t) \right\|_2^{\frac{2}{L}-1}$. That is,

$$\frac{\left\| \boldsymbol{N}(t) \right\|_2^{\frac{2}{L}-1}}{\left\| \boldsymbol{N}(0) \right\|_2^{\frac{2}{L}-1}} \in \left[ 1 - \kappa_L, \frac{1}{1-\kappa_L} \right] = \left[ 0.5^{\frac{L-2}{L}}, 0.5^{-\frac{L-2}{L}} \right] \implies \frac{\left\| \boldsymbol{N}(t) \right\|_2}{\left\| \boldsymbol{N}(0) \right\|_2} \in [1/2, 2].$$

Now we claim it must hold that $T' \geq T_{C_0,r}$. Otherwise, we have

$$\frac{C_0}{2} \left\| \boldsymbol{N}(0) \right\|_2 \leq C_0 \left\| \boldsymbol{N}(T') \right\|_{\mathrm{F}} \leq \left\| \boldsymbol{W}_1(T') \right\|_{\mathcal{V}} \leq e^{-\mu_1 T'/2} \left\| \boldsymbol{W}_1(0) \right\|_{\mathcal{V}} \leq e^{-\mu_1 T'/2} r.$$

Therefore, $\frac{\kappa_L \left\| \boldsymbol{N}(0) \right\|_2^{\frac{2}{L}-1}}{(L-2)\rho} = T' \leq \frac{2}{\mu_1} \ln \frac{2r}{C_0 \left\| \boldsymbol{N}(0) \right\|_2}$, which contradicts to the definition of $C_0$ and $r$.

As a result, we have

$$2C_0 \sqrt{d} \left\| \boldsymbol{N}(0) \right\|_2 \geq 2C_0 \left\| \boldsymbol{N}(0) \right\|_{\mathrm{F}} \geq C_0 \left\| \boldsymbol{N}(T_{c_0,r}) \right\|_{\mathrm{F}} = \left\| \boldsymbol{W}_1(T_{C_0,r}) \right\|_{\mathcal{V}}$$
$$\geq \left\| \boldsymbol{W}_1(0) \right\|_{\mathcal{V}} e^{-\mu_1 T_{C_0,r}/2},$$

and therefore,

$$T_{C_0,r} \leq \frac{2}{\mu_1} \ln \frac{\left\| \boldsymbol{W}_1(0) \right\|_{\mathcal{V}}}{2\sqrt{d} C_0 \left\| \boldsymbol{N}(0) \right\|_{\mathrm{F}}}.$$

Thus by Lemma M.2, we know

$$\begin{aligned}
\left\| \boldsymbol{W}(T_{C_0,r}) - \boldsymbol{W}_0 \right\|_{\mathrm{F}} &\leq \left\| \boldsymbol{W}(T_{C_0,r}) - \boldsymbol{W}_0 \right\|_{\mathrm{F},1} + \left\| \boldsymbol{W}(T_{C_0,r}) - \boldsymbol{W}_0 \right\|_{\mathrm{F},2} \\
&\leq K \left\| \boldsymbol{W}_1(T_{C_0,r}) \right\|_{\mathcal{V}} + \left\| \boldsymbol{M}(T_{C_0,r}) - \boldsymbol{W}_0 \right\|_{\mathrm{F},2} + \left\| \boldsymbol{N}(T_{C_0,r}) \right\|_{\mathrm{F},2} \\
&\leq O(\left\| \boldsymbol{N}(0) \right\|_{\mathrm{F}}) + O(\left\| \boldsymbol{N}(0) \right\|_{\mathrm{F}}^2) + O(\left\| \boldsymbol{N}(0) \right\|_{\mathrm{F}}) \\
&= O(\left\| \boldsymbol{N}(0) \right\|_{\mathrm{F}}).
\end{aligned}$$

$\square$

## M.3   PROOF FOR THEOREM 6.4

*Proof for Theorem 6.4.* Let $C_0, r$ be the constants predicted by Theorem M.4 w.r.t. to $\overline{\boldsymbol{W}}(\infty)$. We claim that we can pick large enough constant $T$, and $\alpha_0$ sufficiently small, such that for all $\alpha \leq \alpha_0$, the initial condition in Theorem M.4 holds, i.e. $C_0 \left\| \boldsymbol{N}(0) \right\|_{\mathrm{F}} \leq \left\| \boldsymbol{W}_1(0) \right\|_{\mathcal{V}} \leq r$, where $\boldsymbol{W}(0) := \phi \left( \alpha \boldsymbol{I}, \frac{\alpha^{-(P-1)}}{2\mu_1^{-1}(P-1)} + T \right)$.

This is because we can first ensure $\left\| \overline{\boldsymbol{W}}(T) - \overline{\boldsymbol{W}}(\infty) \right\|_2$ is sufficiently small, i.e., smaller than $\frac{r}{2}$. By Theorem 6.2, we know when $\alpha \to 0$, $\left\| \overline{\boldsymbol{W}}(T) - \boldsymbol{W}(0) \right\|_{\mathcal{V}} \leq K \left\| \overline{\boldsymbol{W}}(T) - \boldsymbol{W}(0) \right\|_{\mathrm{F}} = o(1)$ and $\left\| \boldsymbol{N}(0) \right\|_{\mathrm{F}} = O(\alpha)$.

By Theorem M.4, we know there is a time $T$ (either $T_{C_0, r}$ or some sufficiently large number when $T_{C_0, r} = \infty$), such that $\left\| \boldsymbol{W}(T) - \boldsymbol{W}_0 \right\|_{\mathrm{F}} = O(\left\| \boldsymbol{N}(0) \right\|_{\mathrm{F}}) = O(\alpha)$. □

