# OpenReview forum: "Towards Resolving the Implicit Bias of Gradient Descent for Matrix Factorization: Greedy Low-Rank Learning"
_ICLR.cc/2021/Conference — ICLR 2021 Poster_

### Official Review · AnonReviewer2 · 2020-10-13
**Interesting paper; unfortunately there is a mathematical flaw**

**Rating:** 6
**Confidence:** 4

**Review:**

This paper analyses the implicit regularization in matrix factorization. The author suggest that for infinitesimal initialization the implicit regularization can be equivalently understood as iteratively solving matrix factorization with increasing rank until the rank is high enough such that the model can be fitted to zero loss. They provide certain theoretical evidence for their hypothesis. Furthermore, they analyze certain models with higher depth, claiming that in those models the described phenomenon appears already with "less infinitesimal" initialization.

As another contribution, the authors claim that their result refutes a conjecture by Gunasekar et al by giving a concrete counterexample. However, I think their counterexample is invalid: $M_{\text{rank}}$ does not have rank 1 as the authors claim; in particular, it is not even symmetric and, hence, it cannot be represented in the form $UU^T$. Also, I do not believe that there is a rank-one matrix, which can represent $M$. Hence, $M_{\text{nuc}}$ is the matrix of smaller norm, which fits to the observations.
This mistake is the main reason for my low score. I am willing to raise my score, if these issues have been addressed.

Further comments:
-Do I understand it correctly that Theorem 6.3 and its interpretation below do not predict any different behavior as long as the number of layers is chosen larger than $3$? Moreover, I am not sure whether I find this argument very convincing: The authors claim that one advantage of deeper layers lies in the fact that it keeps the low-rank component more separated from the learned component. I agree that Theorem 6.3 gives some hint, but there is much more evidence required and at its current state this is rather speculative.

Typos:
-p.4: sixth line from below: Do you mean  $ \mathcal{L} (U_{r-1}) \approx \mathcal{L} (U_r)- \epsilon \mu_1$ instead of $ \mathcal{L} (U_r) \approx \mathcal{L} (U_r)- \epsilon \mu_1 $?
-p.4 second line from below: $\phi$ is not really defined yet, if I make no mistake
-Theorem 5.3: $\tilde{\gamma}$ is not yet defined
-p.7: seventh line from below, $T_{(}' \alpha )$

-----------------------------
After having another look at the paper, I noticed that my complaint could be resolved easily in the following way: In $M_{rank}$ you replace 100 with $R^2$ and then the resulting matrix is symmetric and rank-one. Is this just a typo? I apologize that I missed this.
Furthermore, I wonder whether the proof in Appendix H could be clarified. I feel that the part where the assumptions 5.5 and 5.7 are verified is not particularly clear and more details would be helpful.
(Of course, I realize that I am late with this particular request, but I still believe that in this way the paper could be improved.)

--------------------------

Thank you for your response. I raised my score accordingly.

---

> ### Author Response · Authors · 2020-11-25
> **Response to Reviewer 2**
>
> 1. **Bug in the counterexample:** We sincerely apologize for the typo in the counterexample ($100$ should be $R^2$ in the original manuscript) which leads to the confusion. In the revision we elaborate the proof of the counter-example (including the verification of Assumptions 5.5 and 5.7) Thank you for going through the details of our proof carefully.
>
> 2. **Marginal Value of being deeper**: You are correct that on the impression that Theorem 6.3 and its interpretation doesn't predict any qualitative difference among $L\ge 3$. That's a great catch! We agree that theoretical evidence alone is limited. However, in all the experiments, we see the behavior gap between the behavior of $L=3$ vs $L=4$ is much closer than that of $L=2$ vs $L=3,4$. See Figure 1,2 and 6.
> 	Moreover, with the additional 1-page allowed for the revision, we elaborate on this point with more theoretical and experimental evidence which was originally only in the appendix. In short, we show we depth goes to infinity, the trajectory of GF converges both theoretically and experimentally to a limit, with a suitable rescaling of time (multiplied by depth $L$). See Section B.6 and Figure 4. The trend in Figure 4 is monotone and this suggests that we shouldn't expect any ''phase transition'' when the model becomes deeper and deeper.
>
> 3. **Typos**: We fixed the typos you mentioned. $\phi$ and $\tilde{\gamma}$ are indeed defined, right below Equation(2) and two lines above Definition 5.2, respectively. We repeat these definitions where they are first used.
>
> We thank you for your effort in reviewing our manuscript. We hope the revision and response have addressed all your concerns, especially on the correctness of the proof. In such a case, we would like to ask you to increase your rating.

---

### Official Review · AnonReviewer4 · 2020-10-16
**A good submission, with some points to be clarified**

**Rating:** 8
**Confidence:** 3

**Review:**

**Summary**

The paper studies the implicit bias of gradient descent in the problem of matrix factorization. The objective is to gain some intuition on the effective regularization performed by gradient descent in more involved problems, e.g. neural networks. Previous papers conjectured that gradient flow (i.e. the continuous-time limit of gradient descent) implicitly regularized by the nuclear norm, while more recent ones conjectured that the regularization was done purely in terms of rank. This work provides an answer to this question in an important setting, i.e. they show analytically and numerically that for the problem of factorizing two matrices without any rank constraint, the gradient flow (GF) algorithm effectively minimizes the rank rather than the nuclear norm, if starting from an infinitesimal initialization. This is done by showing that GF is equivalent to an algorithm called Greedy Low-Rank Learning (GLRL). They also provide evidence to show that this phenomenon transfers to the factorization of a larger number of matrices, and that in this context it is actually stronger as its dependency on the infinitesimal initialization weakens.

Note that given the length and available time, I did not check all calculations and experiments given in the supplementary material.

**Main comments and overall decision**

I found the paper well-written and pleasant to read. The results are quite clearly stated and explained. I think the paper would deserve some re-organization to better fit the 8-pages limit, as it feels a bit compressed, especially towards the end. I also believe that some numerics in the appendix should be included in the main text, clearing space by removing some technicalities. Moreover, the previous literature on greedy rank algorithms should be perhaps better discussed.
I believe the paper should ultimately be accepted at ICLR 2021, after the authors have taken into account the comments and criticisms I expressed.

**Strengths of the paper, by section**

- The paper is overall well-written and clear. In particular, the introduction and related works give a very clear overview of the previous literature (however I have to say that I am not extremely familiar with the previous works on the implicit regularization of gradient descent, so I could be forgetting important works).

- I genuinely enjoyed Section 4 which gives an intuition on the reasons behind this low-rank learning behavior of GF in very simple toy models. On a general note, the splitting of the paper into clear sections is very good for the readability of the paper.

- Section 5 is quite technical, and I believe some of these technicalities could be moved to appendix to improve readability and gain space. Having generalities in Section 5.1 makes the intuition quite clear, especially with Thm 5.3. I found the counterexample 5.9 to be particularly simple and appropriate. Section 5.3 could also perhaps be shortened, by making the results even more informal: the intuition behind the iterative rank growth of the solution of GF is very interesting and should be emphasized more than the technicalities.

- The treatment of depth in Lemma 6.1 and Theorem 6.2 is well presented. I noticed that in both Thm 6.2 and 5.3 the times considered in GF and GLRL are not the same, could the authors comment on this time difference, and its dependency on alpha ?

**Concerns and remarks**

- In general, the paper has many results and claims, however it feels like the 8-page limit is quite limiting the results, resulting in numerous references to the supplementary material. I think the paper should perhaps be shortened, removing some technicalities, for instance in Section 5. For instance the last paragraph of the paper (page 8) should either be removed, or the experiments should be included in the main text, but its current state makes it unclear. In this paragraph I also did not understand the sentence “For depth-2 GLRL, the low-rankness is raised to some power smaller than 1, which depends on the eigenvalue”, could the authors clarify it?

- On Greedy Low Rank algorithms, the paper (if I am not mistaken) does not compare this GLRL algorithm to previous algorithms developed in the literature, e.g. in “On Approximation Guarantees for Greedy Low Rank Optimization” , ICML 2017 or “Greedy Learning of Generalized Low-Rank Models” (IJCA 2016). I have to admit that I am not very familiar with this literature, but I believe that this type of algorithm is not entirely new, and the authors should perhaps discuss it more, or at least clarify this in their response.

- Figure 1 is the only numerical analysis presented in the main text (there are more in appendix). The numerics show that it requires really an extremely small initialization to see the closeness of GF and GLRL (it starts to be clear for initial norms around $10^{-12}$ !). This is counterbalanced with Figure 2 in the depth $L\geq 3$ setting, but this is only shown in appendix. In general, I believe that at least some part of Figure 2 should be included in the main text, perhaps merging it with Figure 1.

**Minor remarks**

- I was a bit confused when reading: after analyzing the simple linear case, the authors write “But what if $W(t)$ grows to be so large that the linear approximation is far from the actual $f(W)$?”, suggesting that the real behavior is not captured by the linear approximation, however to me it seems like Theorem 5.3 precisely shows under which conditions on the initialization one can use the linear approximation for the dynamics of GF.

- In Section 4, when considering Matrix Sensing, appealing to Thm~1 and 4 of (Gissin&al 20) seems completely overkill and does not read well, as the first result is simply writing the ODE in the diagonal basis, and the second is simply solving an ODE by separation of variables.

- What do the authors mean in Algorithm 1 by “if $W_r$ is close to a local minimizer of f among PSD matrices” ? Do they mean that going from rank r to (r+1) in GLRL does not change $W_r$ ?

- In Theorem 6.3, why do the authors consider $inf_t$ and not a limsup? Is this simply for technical reasons in the proof, or is this genuinely possible that the trajectory “escapes” the GLRL solution for long enough times ?

Finally, a list of typos :
- Introduction: “we show GF […] converges” → “we show that GF […] converges” ?-
- “before GLRL reaches first stationary point” → “before GLRL reaches its first stationary point”
- Related works sections: “which remains it unclear”
- Beginning of page 7: “Which is 40 when $R = 40$” → “Which is 40 when $R = 10$”]
- Thm 5.10: “the eigendecomposition of $W$” → “the eigendecomposition of $– \nabla f(W)$”
- Page 7 : A parenthesis is misplaced close to the end of the page.
- Page 8 (middle) : Gf → GF
- Page 8 : “depth-2 solution of have” → Something missing.
- Page 8 : a “but” should be removed.

---

> ### Author Response · Authors · 2020-11-25
> **Response to Reviewer 4**
>
> Thanks for your appreciation! We have adopted part of your advice on the organization of our paper, including moving Figure 2 to the main text and making Section 5.3 informal. We also fixed the typos you pointed out.
>
> 1. **Time difference between gradient flow and GLRL:** In Theorem 5.3 and 6.2, GLRL is compared with gradient flow at time $\frac{1}{2\mu_1} \log \frac{1}{\alpha} + t$ and $\frac{\alpha^{-(1-2/L)}}{2\mu_1(L/2-1)} + t$ respectively (here we take $W_{\alpha} = \alpha I$ in the two-layer case as an example). This difference is because gradient flow with small initialization needs some time to grow the weight from very small scale to a constant scale. As we want to compare gradient flow with different $\alpha$ to the fixed trajectory of GLRL that is independent of $\alpha$, there must be a time difference depending on $\alpha$. The dependency of $\alpha$ is related to the growth rate of $W$ from small initialization: $W$ grows to the constant scale in $O(\log \frac{1}{\alpha})$ time in the depth-2 case, while it needs $O(\alpha^{-(1-2/L)})$ in deeper cases. Also note GLRL itself is the limit of a sequence of trajectories whose initial scale and time are parametrized by $\alpha$.
> 2. **Explanation for "For depth-2 GLRL, the low-rankness is raised to some power smaller than 1, which depends on the eigenvalue":** We have added a formal definition of low-rankness in the revision and rephrased the paragraph of "So how does depth encourage ..." using the language of low-rankness.
> 3.  **Missing Related Works:** We have added two paragraphs for the similarity and difference between GLRL and other greedy learning algorithms in the literature at the top of Page 5, including the papers you mentioned.
> 4. **Why can linearization fail?** In the first warmup example, "linearization" refers to approximating $f$ by a linear function, and it only holds when $W$ is sufficiently small. It is true that this idea can be explored to obtain Theorem 5.3, but the real challenge is that gradient flow can escape from the rank-1 solution $\overline{W}_1$ and become rank-2, as we have illustrated in the second warmup example.
> 5. **References to (Gissin et al., 2020):**  We have removed the reference to (Gissin et al., 2020) when deriving for the simple formula for $\sigma_i(t)$.
> 6. **Ambiguity in the algorithm of GLRL:** We clarified the steps which were not implementable. See Algorithm 1 for the updated pseudocode and Appendix B.1 for the implementation details.
> 7. **inf in Lemma 6.1:** It is genuinely possible that the trajectory can escape from the rank-1 solution $\overline{W}(\infty)$ of GLRL for long enough time, if $\overline{W}(\infty)$ is not a local minimizer. In (Gissin et al., 2020), it is shown that deep matrix factorization in the full-observation case learns the solution with gradually increasing rank, just like what we have seen in the second warmup example. Thus, we have to consider $\inf_t$ in Lemma 6.1; otherwise, the lemma can be incorrect.
>
> We again thank you for your effort in reviewing our paper, especially for your detailed comments for each section. We hope the revision and response have addressed your criticisms.

---

### Official Review · AnonReviewer3 · 2020-10-28
**A good contribution for the understanding of the implicit bias of gradient descent.**

**Rating:** 7
**Confidence:** 3

**Review:**

Summary
This paper analyses the dynamics of gradient descent learning for matrix factorization. The authors show that when there are two factors, the gradient flow initialized close from 0 closely follows another algorithm, called greedy low rank learning (GLRL), which learns the factors starting from rank one approximations, and then greedily adds dimension. This shows that the gradient flow learns incrementally, and favors low rank solutions. This allows to invalidate a conjecture stating that gradient flow finds minimum nuclear norm solutions. The authors finally study deeper factorizations under the light of GLRL


Major comments
- The article is easy to read and well organised. Some results could be made more explicit (see minor comments).
I am not an expert in this domain, but it seems like the article provides an advance towards understanding the implicit bias of gradient descent.
- Algorithm 1 is not implementable, the authors should be clearer about what ‘keeps decreasing’, or ‘close to a local minimum’ mean. I think that it is rather a theoretical ‘oracle’ algorithm, where the gradient descent on $U$ should be a gradient flow and the outer look ends when the rank is matched. The practical approximation to this algorithm can then be discussed in appendix.
Thm.5.3 seems like an important and fairly general result for the dynamics of the gradient flow.


Minor comments
- In eq.5, the matrices are always going to be symmetric, so why is there a transpose?
- Beginning of page 4, why start from $U(0) = \sqrt{\alpha} I$ ? I think that the general case ($U(0)$ non diagonal), is also tractable in closed form.
- The references to Gissin 2020 in the same paragraph also seem unnecessary to invoke for such a simple result, and feel a bit frustrating.
- In 5.1, the authors assume that $J(0)$ is diagonalizable, and write $J = \sum_{i=1}^d\mu_i \mathbf{v}_i \mathbf{v}_i^{\top}$. It should be specified exactly what the vectors $\mathbf{v}_i$ and $\mathbf{u}_i$ verify: to me, this looks more like a singular value decomposition; a diagonalization would imply that $U = V^{-1}$ (but I think that the authors really need diagonalization here).
- Page 5: “One can easily show that for the other case that the initialization has positive alignment with...” it would valuable to explicitly write down the expression for the “other $z(t)$” in question.
- Theorem 5.3: similarly, it would be valuable to explicitly write down the expression for $z(t)$.
- Lemma 5.4: it would be insightful to write down the (simple) explicit expression for $J(0)$.
- Lemma 6.1: It should be specified before that $W$ is symmetric, otherwise it makes the definition of $W^{2/L}$ awkward.
- The reasoning in the proof of the lemma 6.1. is also slightly misleading: it should be made more explicit that the authors do not assume that $R= W^{1/L}$, but rather that $R$ is defined by the initial value problem $R(0) = W(0)^{1 / L}$ and the ODE (25). Then, the authors verify that $R^L$ satisfies the same ODE as $W$, hereby showing $R = W^{1/L}$.


Misc

- In the intro: “which remains it unclear whether the approximation error can be bounded until convergence”: this should be rephrased
- Avoid undefined acronyms
- Page 5: “, e.g., GF around a saddle point lies in this case” this sentence is unclear to me.
- Page 5: “A special case is that the direction of$\theta_{\alpha}  -\theta_0$ converges.”: I don’t understand this sentence, does it mean something like “this definition implies that $\theta_{\alpha}  -\theta_0$ converges”?
- Theorem 5.6, first equation: there is a missing $\varepsilon$ subscript missing after the first missing on the $W$

---

> ### Author Response · Authors · 2020-11-25
> **Response to Reviewer 3**
>
> Thanks for your careful reading and appreciation! We have fixed all the typos and clarified the unclear points in the rebuttal revision.
>
> 1. **Ambiguity in the algorithm of GLRL:** We clarified the steps which were not implementable. See Algorithm 1 for the updated pseudocode and Appendix B.1 for the implementation details.
> 2. **Initialization in the second warmup example:** In the second warmup example, we start GF from $U(0)=\sqrt{\alpha}I$ because the general case does not seem to admit a closed-form solution (to the best of the authors' knowledge).
> 3. **References to (Gissin et al., 2020):** We have removed the references to (Gissin et al., 2020) when deriving the simple formulas for $\sigma_i(t)$. Thanks for the suggestion.
> 4. **Eigendecomposition of $J(0)$:** In Section 5.1, $J(0) = \sum _ {i} \tilde{\mu} _ i \tilde{v} _ i \tilde{u} _ i^{\top}$ is indeed the eigendecomposition. Let $J(0) := \tilde{V} \tilde{D} \tilde{V}^{-1}$ be the eigendecomposition of $J(0)$ in the matrix form, where $\tilde{V}$ is an invertible matrix and $\tilde{D} = \mathrm{diag}(\tilde{\mu} _ 1, \dots, \tilde{\mu} _ d)$ is the diagonal matrix consisting of the eigenvalues $\tilde{\mu} _ 1 \ge \tilde{\mu} _ 2 \ge \cdots \ge \tilde{\mu} _ d$. Let $\tilde{V} = (\tilde{v} _ 1, \dots, \tilde{v} _ d)$ and $\tilde{V}^{-1} = (\tilde{u} _ 1, \dots, \tilde{u} _ d)^{\top}$, then $\tilde{u} _ i$, $\tilde{v} _ i$ are left and right eigenvectors associated with $\tilde{\mu} _ i$ and we can rewrite the eigendecomposition as $J(0) = \sum_{i=1}^{d} \tilde{\mu} _ i \tilde{v} _ i\tilde{u} _ i^{\top}$. We have clarify this point in the revision.
> 5. **Other unclear points:** The transpose is unnecessary in eq.5, so we have removed this. We have given the explicit expressions for "other $z(t)$" and $J(0)$ in the revision. We have added the condition that $W(t)$ is symmetric in Lemma 6.1 and rewrite the proof to clarify that we do not assume that $R = W^{1/L}$.
>
> We believe the clarity of our paper is much improved in the revised version. We hope the revision and the response have addressed all your concerns.

---

### Official Review · AnonReviewer1 · 2020-10-28
**Interesting results for rank-1 while the extension are harder to understand**

**Rating:** 6
**Confidence:** 3

**Review:**

## Summary

This paper study the solution of matrix factorization that can be found using gradient descent. In particular, the authors show that gradient descent converges toward low rank solution obtained with GLRL when the initialization magnitude is small enough. The proof are based on the analysis of Gradient Flow and its link to power iteration for linear system. The authors show that for rank-1 solution, trajectories with any small enough initialization have 2 behaviors:
- a power iteration one where the trajectory gets close to $\epsilon v_1v_1^\top$
- the same trajectory as the one starting from this $\epsilon v_1v_1^\top$ point, which correspond to the GLRL algorithm.

Thus the limit of the trajectory are independent of the initialization for the init is small enough. Then these results are extended to higher rank solution and to more layers. Some small scale experiments highlights those results.

## Overall assessment

- I find the results and the methodology of for the results in the rank-1 case very interesting. Part of the discussion of the results can be improved -- in particular the tradeoff between power iterations and the trajectory and the effect of the $\log \frac{1}{\langle W_alpha, u_1u_1^\top\rangle}$ on the convergence speed.
- The results in the general case (5.3) and deep MF (section.6) are harder to grasp and understand in the current form. I would advise to move section.6 in appendix to make more room for section5.3 and experiments. In particular, the setting of Figure.1 is not detailled enough. We don't know the data generation process/ the ratio of missing entries/... which are important to understand the figure.
- One big question to me is how to ensure the assumption that one can find $T_\alpha$ such that $\phi(W_\alpha, T_\alpha)$ converges to $\widebar W$ when it is not rank 1. In particular, we know that there exists a $T_alpha$ such that it will be very close of a rank-1 saddle point. The question is how to make sure this can be escaped as for 1st order differential equation, once 2 trajectories cross, they are the same I think. I would like the authors to clarify this point.
- Figure.1, instead of the mean, it would be more interesting to report the max as the bound in (7) is uniform on the $\|W_\alpha\|_F$ ball. Also, using 20 repetitions would be more convincing (these are small scale experiments so averaging over many realisation is possible).
- I would find it interesting to explore the effect of using $\epsilon v_1v_1^\top$ as an initialisation compared with random init. As a significant part of the optim is used to perform power iteration to find the right direction and the results does not depend on it, it should give better results.
- There is a series of work very related to this one on global optimality of matrix factorization that also advocate for a greedy resolution. It would be interesting to mention this in the related work. See:

```bibtex
@article{Haeffele2019,
  title = {Structured {{Low}}-{{Rank Matrix Factorization}}: {{Global Optimality}}, {{Algorithms}}, and {{Applications}}},
  author = {Haeffele, Benjamin D. and Vidal, Ren{\'e}},
  year = {2019},
  volume = {42},
  pages = {1468--1482},
  archivePrefix = {arXiv},
  eprint = {1708.07850},
  eprinttype = {arxiv},
  journal = {IEEE Transactions on Pattern Analysis and Machine Intelligence (PAMI)},
  number = {6}
}
```

## Minor comments, nitpicks and typos

Some minor comments and grammatical comment. I am not an english native speaker so it might not be real mistakes.

- p.1: `(see (Chi et al. 2019) for` - use `citealt` to avoid duplicated parenthesis?
- p.1: `however, even in the case that the rank` -> `in the case where`?
- p.1: `but its behind mechanism` -> `but its mechanism`?
- p.2: `one single` -> `a single`?
- p.2: `and matrix factorization -- the focus of our paper.` references are needed here.
- p.3: `there is an unknown rank-$r$ matrix` -> rank $r^*$.
- p.4: `a unit top eigenvector` -> isn't it the normalized top eigenvector? The part on the `arbitrary` top eigenvector is unclear. This should be clarified.
- p.6: `Thm5.6` There is a missing $\epsilon$ in the definition of $W_1^G$.
- p.8: `for distance between Gf` -> `GF`.
- Algorithm.2: I think the update run for `perform on step` is incorrect. It should be something like $U_{r,i} \leftarrow U_{r,i} - \eta \prod_{j=1}^{i-1}U_{r,j}\nabla f(\prod_{j=1}^LU_{r, i})\prod_{j=i+1}^{L}U_{r,j}$.

---

> ### Author Response · Authors · 2020-11-25
> **Response to Reviewer 1**
>
> Thank you for your detailed reviews and constructive suggestions. We are glad that you find our rank-1 result interesting. We have incorporated your specific remarks and minor comments into the revision. Below we will respond to your comments point by point.
>
> 1. **On the convergence rate in Theorem 5.6:** In the revision, we give a recap of the definition of convergence with positive alignment in Theorem 5.6, which essentially says that there is a constant $q > 0$, such that for sufficiently small $W_{\alpha}$, $q\| W_{\alpha} \| _ F \le \langle W_{\alpha}, u_1u_1^\top\rangle $. In this way one can replace the RHS of (8) by $C \cdot \left(\frac{1}{q}\langle W_{\alpha}, u_1u_1^\top\rangle\right)^{\tilde{\gamma} / (2\mu_1 + \tilde{\gamma})}$.
> 2. **Presentation of Section 5.3 and 6:** We utilized the one additional page for a better presentation of section 5.3 and 6. We would really appreciate it if you could give another chance for these sections, especially section 6. In the revised section 6, we gave theoretical justification for why the performance gap between $L=2$ and $L\ge 3$ is much larger than the differences among $L\ge 3$, e.g., $L=3$ vs $L=4$. (due to page limit, those results were buried in the appendix) Also see the corresponding experimental verification in Figures 4 and 6 in appendix B.
> 3. **Regarding the assumption that there's $T_{\alpha}$ that $\phi(W_\alpha,T_\alpha)$ converges to $\overline{W}$:** In the updated section 5.3, we prove that for random initialization, GF almost surely only finds the global minimizers (Theorem 5.10). Thus if the ground truth is not low rank, no matter how close the trajectory of GF (or GD) gets to the rank-1 saddle, it will escape that saddle eventually. This theoretical result is a continuous analog of the main theorem in (Lee et al, 2016), (Lee et al., 2017), which shows that Gradient Descent (or other discrete first-order methods) only finds minimizers.
>
> 	Not only this assumption is theoretically reasonable, but it's also consistent with the experimental observation that overparametrized GD with full rank initialization always achieves 0 training loss for matrix completion.
>
> 4. **Presentation of Figure 1:** We added the details of the model generation and observation ratio into the caption of Figure 1. Indeed in the original manuscript, we plotted each repetition separately but most of them just overlap with each other so you might think we were reporting the mean. We increased the number of repetitions to 20 in the revision and noted clearly in the caption that they are plotted separately, though most of the distance curves still tend to be almost the same regardless of the randomness of the initialization.
> 5. **Suggestion for exploring the effect of using $\varepsilon v_1v_1^\top$ compared to random initialization**: Thanks for your insightful suggestion! We compared their effects experimentally in Figure 5, where we measure the distance of trajectories with each initialization to a fixed reference point on the trajectory of GLRL. The conclusion is that $\varepsilon v_1v_1^\top$ is closer to GLRL by magnitudes ($\sim10^4$ times closer). Thus the take-home message here is that GLRL is in general a more computational efficient method to simulate the trajectory of GF (GD) with infinitesimal initialization, as one can start GLRL with a much larger initialization, while still maintaining high precision.
>
> 	This matches with our theoretical prediction. Suppose $\|W_\alpha\|_F =\Theta(\alpha)$, the convergence rate of GF with random initialization to a fixed reference point on GLRL is given in Theorem 5.6, which is $\alpha^{(\lambda_1-\lambda_2) / (3\lambda_1-\lambda_2)}$ (the most general form is in Theorem 5.3). As a comparison, the convergence rate for initialization being $\alpha v_1v_1^\top$ is $O(\alpha)$, as given in Lemma C.5.
>
> 6. **Missing Related Works:** We added two paragraphs for related works on greedy algorithms at the top of Page 5, including the paper you mentioned. In Figure 3 we show our methods outperforms R1MP (Wang et al., 2014) in our small scale example, where ground truth is rank-3. (Note Figure 3 was also in the original manuscript)
>
> 7. **Typos:** We fixed all the typos pointed out by the reviewer.
>
> We again thank you for your great care and effort in reviewing our manuscript, which helps us improve the manuscript by a lot. We hope the revision and response have addressed all your concerns.

---

### Author Response · Authors · 2020-11-25
**An Overview of Paper Update**

We thank all the reviewers sincerely for their constructive and positive feedback. We have incorporated all the suggestions in our updated manuscript, and we hope that the reviewers will reconsider their ratings to reflect the revisions. Below we provide an overview of the revisions. Please also see our individual responses to each reviewer.

Major Updates:

1. Adding comparison to existing greedy algorithms for rank-constrained optimization (Section 5). (R1)
2. Algorithm 1 and 2 updated ---  steps with ambiguity are clarified and now the algorithm is implementable. (R1,R3,R4)
3. Section 5.3 updated --- A new theorem showing that GF with random initialization only converges to global minimizers, thus will not get stuck at rank-1 stationary point, unless the ground truth is rank-1. (R1)
4. Section 6 updated --- we formally define the notion of "Low-Rankness" and improve readability. (R4) We brought up the discussion on the marginal value of being deeper from the appendix to the end of Section 6. (R2)
5. A new figure (Figure 5) is added, which compares the effect between using $\varepsilon v_1v_1^\top$ as initialization and using random initialization. (R1)
6. Figure 2 moved from the appendix to the main text. (R4)
7. Elaboration on the proof of the counterexample (Appendix I). (R2)
8. We fixed the typos in the expression of $M_{rank}$. The correct nuclear norm should be $2R^2+2$ instead of $R^3+2R+R^{-1}$. The measurements and objective remain unchanged. This doesn't change any of our conclusion.

---

### Decision · Program_Chairs · 2021-01-07
**Final Decision**

**Decision:**

Accept (Poster)

**Comment:**

The article is easy to read, of interest for the community, and provide some advance towards understanding the implicit bias of gradient descent.
The results and the methodology for the rank-1 case are very interesting and convincing.
Yet, some results could be made more explicit and the comments by the reviewers should be addressed for the camera ready paper, in particular the one on the organization.